



# BAERLIN2014 - The influence of land surface types on and the horizontal heterogeneity of air pollutant levels in Berlin

B. Bonn[1*], E. von Schneidemesser[1], D. Andrich[1**], J. Quedenau[1], H. Gerwig[2], A. Lüdecke[2], J. Kura[2], A. Pietsch[2], C. Ehlers[3], D. Klemp[3], C. Kofahl[3], R. Nothard[4], A. Kerschbaumer[4], W. Junkermann[5], R. Grote[5], T. Pohl[6], K. Weber[6], B. Lode[1], P. Schönberger[1], G. Churkina[1], T. M. Butler[1] and M. G. Lawrence[1]

[1]{Institute for Advanced Sustainability Studies (IASS), D-14467 Potsdam, Germany}

[2]{Division Environmental Health and Protection of Ecosystems, D-06844 Dessau-Roßlau, Germany}

[3]{IEK-8, Research Centre Jülich, D-52425 Jülich, Germany}

[4]{Senate Department for Urban Development and the Environment, 10179 Berlin, Germany}

[5]{Institute of Meteorology and Climate Research, Atmospheric Environmental Research (IMK-IFU), Campus Alpin, D-82467 Garmisch-Partenkirchen, Germany}

[6]{Environmental Measurement Techniques, University of Applied Sciences, D-40474 Düsseldorf, Germany}

[*]{now at: Institute for Forest Sciences, Albert-Ludwig University, D-79110 Freiburg, Germany}

[**]{now at: Andritz AG, Graz, Austria}

*Correspondence to: Erika von Schneidemesser (evs@iass-potsdam.de)*

**Abstract.** Urban air quality and human health are among the key aspects of future urban planning. In order to address pollutants such as ozone and particulate matter, efforts need to be made to quantify and reduce their concentrations. One important aspect in understanding urban air quality is the influence of urban vegetation which may act as both, emitter and sink for trace gases and aerosol particles. In this context, the "Berlin Air quality and Ecosystem Research: Local and long-range Impact of anthropogenic and Natural hydrocarbons 2014" (BAERLIN2014) campaign was conducted between the June 2nd and August 29th in the metropolitan area of Berlin-Brandenburg, Germany. The predominant goals of the campaign were (1) the characterization of urban gaseous and particulate pollution and its attribution to anthropogenic and natural sources in the region of interest, especially considering the connection between biogenic volatile organic compounds and particulates and ozone; (2) the quantification of the impact of urban vegetation on organic trace gas levels and the presence of oxidants such as ozone; and (3) to explain the local heterogeneity of pollutants by defining the distribution of sources and sinks relevant for the interpretation of model simulations. In order to do so, the campaign included stationary measurements at an urban background station and mobile observations carried out from bicycle, van and airborne platforms. This paper provides an overview of the mobile measurements (Mobile BAERLIN2014) and general conclusions drawn from the analysis. Bicycle measurements showed micro-scale variations of temperature and particulate matter, displaying a substantial reduction of temperature and particulates in the proximity of vegetated areas compared to typical urban residential area (background) measurements. Van measurements extended the area covered by bicycle observations and included continuous measurements of $O_3$, $NO_x$, CO, $CO_2$, and pointwise volatile organic compounds (VOCs) identification. The quantification displayed notable horizontal heterogeneity of the short lived gases and particle number concentrations. E.g. concentrations





of the traffic related chemical species CO and NO varied by more than ±20 % and ±60 % on the scale of one
hundred meters, respectively. Airborne observations revealed the dominant source of elevated urban particulate
number and mass concentrations being local, i.e. not being caused by long range transport. Surface based
observations related these two parameters predominantly to traffic sources. Vegetated areas lowered the
pollutant concentrations substantially with ozone being reduced most by coniferous forests, which is most likely
caused by their reactive biogenic VOC emissions. With respect to the overall potential to reduce air pollutant
levels forests were found to result in the largest decrease, followed by parks and facilities for sports and leisure.
Surface temperature was generally 0.6-2.1°C lower in vegetated regions, which in turn will have an impact on
tropospheric chemical processes. Based on our findings effective future mitigation activities to provide a more
sustainable and healthier urban environment would focus predominantly on reducing fossil-fuel emissions from
traffic as well as on increasing vegetated areas.

## 12    1    Introduction

Today 54% of the Earth's population lives in urban areas (United Nations, 2015). This number is expected to
increase beyond 60% within the next fifteen to twenty years. Due to the highly concentrated resource use, air
pollution levels are closely related to population density, despite some success in reducing emissions (Lamsal et
al., 2013). Numerous epidemiologic studies show that highly polluted conditions, such as experienced in many
cities, are already causing major adverse health effects (e.g. Chen and Kann, 2008; Heinrich et al., 2013; WHO,
2013) expected to worsen with increasing urban areas. Therefore it is crucial to find means for improving air
quality even under increased urbanization and traffic occurrence, which, however, requires a thorough
understanding of sources and sinks of air pollutants.
Poor air quality has been documented in many metropolitan areas such as Bejing (Huang et al., 2015; Huo et al.,
2015; Sua et al., 2015; Zhang et al., 2015), Los Angeles (Chen et al., 2013; Ensberg et al., 2014; McDonald et
al., 2015), Paris (von der Weiden-Reinmüller et al., 2014) and for Europe in general (Henschel et al., 2015).
Elevated levels of gaseous pollutants such as ozone ($O_3$), nitrogen oxides ($NO_x = NO+NO_2$), sulphur dioxide
($SO_2$), toxic agents such as aromatic hydrocarbons and of particulate matter (PM) have been attributed to
anthropogenic emissions from urban sources, especially traffic and energy production (Downey et al., 2015;
Hong et al., 2015; Huo et al., 2015; Padilla et al., 2014). Since atmospheric pollutants can affect the human
respiratory system (e.g. oxygen capacity) and significantly reduce a person's working capacity and life
expectancy (chronic obstructive pulmonary disease, acute lower respiratory illness, cerebrovascular disease,
ischaemic heart disease and lung cancer) (Dockery et al., 1993; Peng et al., 2005; Pope et al., 2009, Lelieveld et
al., 2015) with an intensity of effects depending on time scale, limits of daily and annual averages of pollutant
concentrations have been proposed by national and international authorities (European Union, 2008; WHO,

33    2006).

In this context the European Union introduced legally binding limit values applying to all Member States in the
Air Quality Framework Directive (Directive 2008/50/EC, European Union, 2008). If cities fail to meet these
health related limit values, they are obliged to develop air quality programs capable of reducing the pollution
concentration and the duration of elevated concentrations. As held by the European Court of Justice (ECJ), the





establishment of such air quality programs is a subjective right of any person directly concerned and can thus be
claimed by citizens in court (*Janecek v. Bayern*, ECJ, 2008).
In Germany, the EU-limits for $NO_2$ and $PM_{10}$ continue to be exceeded in many cities (including Berlin).As a
result, in drawing up their air quality programs, the Federal Administrative Court ruled that authorities must
implement all measures available to keep the time of exceedance as short as possible
(Bundesverwaltungsgericht, 2012). Otherwise citizens and environmental associations can sue for an adjustment
of the program, as has already happened in Darmstadt, Hamburg, Limburg, Mainz, Offenbach, Reutlingen and
Wiesbaden.
In consequence Berlin, like every European city, has the legal obligation to provide air quality programs that are
capable of substantially reducing nitrogen oxides and particulate matter. The Senate of Berlin adopted a
respective program for 2011-2017 (Berlin Senate, 2013b). However, given that limit values continues to be
exceeded, it is questionable whether the measures contained herein are sufficient to enable Berlin to comply with
this obligation. An exceedance of these values is only permissible, when all necessary and appropriate measures
at disposal are exhausted. So far Berlin has established an environment protection zone (German "Umweltzone",
$2^{nd}$ step, green level; Berlin Senate, 2011) in the city centre. This measure was set-up to lower traffic related
emissions and the number of critical threshold exceedances according to EU law for $NO_x$ and PM (see Table 1)
in Berlin per year and resulted in an emission reduction by 20% for $NO_x$ and 58% for soot by diesel engines
(Berlin Senate, 2011). A substantial contribution to $NO_x$ and particulate matter (PM) has been claimed to
originate by long range transport from Polish industrialized areas (Kerschbaumer, 2007). Other PM sources can
be attributed to nearby emission or gas-phase (secondary) PM production. As the city of Berlin is surrounded by
and contains extensive forested regions, enclosed by three rivers (Havel, Spree and Dahme) and a couple of lakes
(6% by area), the concentration of trace gases and particles will be influenced from both, i.e. local anthropogenic
and biogenic (vegetation) sources (see e.g. Becker et al., 1999; Beekmann et al., 2007). Due to their provision of
multiple ecosystem services, increasing green areas such as parks and forests are often considered as measures to
counteract urban heat island effects (Fallmann et al., 2014; Grewe et al., 2013; Schubert and Grossman-Clake,
2013) and air pollution problems (Irga et al., 2015; Janhäll et al., 2015). Emission of biogenic BVOCs can affect
chemical ozone production and destruction (Seinfeld and Pandis, 2006; Klemp, 2012) as well as secondary
organic aerosol mass production (Hallquist et al., 2009) if higher terpenes are emitted. A high impact of reactive
BVOCs on $O_3$ concentrations and vice versa has been observed during warm seasons in highly polluted
temperate and semi-arid areas (Papiez, et al. 2009; Bourtsoukidis et al., 2012; Calfapietra et al., 2013; Situ, et al.,
2013), while the influence in northern countries has been found generally smaller (Setälä et al., 2013, von
Schneidemesser, et al. 2011). The reducing effect of vegetation on $NO_x$ concentrations was described earlier by
Velikova et al. (2005). The effects of vegetation and especially the emission of biogenic VOCs (BVOCs,
Guenther et al., 1995; 2006; Ghirardo et al., 2015) have been neglected so far but are expected intensify in a
warmer climate (e.g. Bonn, 2014; Churkina et al. 2015).
Given this background, the presented study tries to support city authorities by improving the knowledge about
small-scale pollution sources and sinks with a focus on the role of vegetation, and thereby supporting authorities
to meet target values and limits based on providing pertinent scientific information to support decision-making.





**2    Focus of the campaign and of this study**
This study focusses on the Berlin-Brandenburg Metropolitan area (BBMA) with about four million inhabitants,
and a hub for major transport routes through Europe. Both cities in this area, Berlin with approximately 3.3
million citizens and Potsdam, the capital of Brandenburg, with about 0.2 million inhabitants, are extraordinary
among European metropolitan areas because of the large proportion of water and vegetated areas making up
about 40% of the total land surface area in the cities (Berlin Senate, 2010; 2013b)(Table 2). Because of its large
area vegetation is expected to have a notable impact on pollution levels (trace gas and aerosol particle
concentrations), and thereby on pollution levels as it was found for other locations (Cowling and Furiness, 2004;
Zaveri et al., 2012). Ambient air pollution levels generally meet the EU limit values with the exception of
nitrogen oxides ($NO_x = NO + NO_2$) and of particulate matter ($PM_{10}$ and $PM_{2.5}$). Based on previous studies in
urban areas, and a limited number of studies in Berlin, the predominant sources of both pollutants are expected
to be traffic, residential heating, industry and transport of primary and secondary particulate matter (Seinfeld and
Pandis, 2006; Berlin Senate, 2013a; 2015). Earlier studies have indicated substantial deviations between
observed and simulated $NO_2$ (mean: -20%) and PM values (mean: -10%) (see e.g. Tullius and Lutz, 2003),
which both influence health (Fischer et al., 2015; Liu et al., 2016) and ozone production (Atkinson et al., 2004;
Seinfeld and Pandis, 2006). *The deviations of PM are linked to secondary and semivolatile organic substances*
*contributing to particulate mass. These contributions vary depending on ambient mixing ratios of VOC*
*precursors as well as on temperature as the precursors' saturation vapour pressure and the total organic*
*particle mass change, aspects, which are not or poorly represented in air quality models due to their complexity.*
Here we present the project "Berlin Air quality and Ecosytem Research: Local and long-range Impact of
anthropogenic and Natural hydrocarbons 2014" (BAERLIN2014). Considering the context outlined in the
paragraph above, it focused on the following questions:
(1)    What is the spatial and temporal heterogeneity of pollutants in the BBMA area with a focus on Berlin

24         and Potsdam?

(2)    How do different vegetation types influence the levels of ozone, $NO_x$ and VOCs in Berlin?
(3)    What is the impact of different types of vegetated areas on urban environmental conditions i.e.

27         temperature, humidity and particulate pollutants (number and mass)?

(4)    And finally what is the contribution of anthropogenic and biogenic organic compounds to secondary

29         organic aerosol and the total particulate mass in the Berlin and Potsdam area affecting health, both

30         directly and indirectly through ozone production?

The present article provides an overview on the mobile measurements tackling points (1), (2) and (3) using
observations in different environments, classifying the data by different vegetated and urban surface types,
comparing to local observations (von Schneidemesser et al., in prep.) and their heterogeneity with respect to
important pollution parameters (CO, $NO_x$, $O_3$ and particle number and mass). Especially aspect (4) is of interest
for simulation studies when comparing model simulation results with measurements to draw conclusions about
PM sources as well as on ozone sources and sinks. The aim of this study was to identify hotspots of pollution,
the variability of basic air pollution trace gases, to quantify the impact of green areas and to exemplarily identify
dominant VOC sources to support action plans such as made by the Berlin Senate (2013b).





The mobile measurements described in this paper were conducted as part of the larger BAERLIN campaign,
which included extensive stationary measurements at an air quality monitoring station in Berlin-Neukölln. The
stationary measurements are described elsewhere (von Schneidemesser et al., in preparation). Both measurement
types contribute to the identification of local sources and sinks as well as their effects on the urban background
concentration of air pollutants. Further studies using atmospheric transport models are planned for assessing
different mitigation options.
**3    Methods**
Mobile measurements were performed using different observation platforms, i.e. bicycles, a van and aircrafts
(Table 3), with tracks throughout and around the BBMA area (Fig. 1). To allow for the comparison of different
measurement types at different times of day and under different conditions a relative parameter method has been
used, which is described in section 3.4. In order to characterize the spatial variation of parameters of interest a
set of instruments and methods on different platforms were linked to form a complimentary set of observations.
The different ranges and scales of observations were
1.    Microscale (3.1 Bicycle measurements), ground-based, real-time and highly spatially-resolved

15          observations: bicycle measurements covering a variety of routes during the three month period;

2.    Mesoscale (3.2 Van measurements), ground-based, including source profiling (traffic and vegetation)

17          and VOC source classification: van (Mercedes VITO) measurements (RC Jülich, Germany) for the first

18          week in August; and

3.    Mesoscale (3.3 Airborne measurements), airborne, in- and outflow of BBMA area: (1) ultralight aircraft

20          (KIT, IMK-IFU, Garmisch-Partenkirchen, Germany) for outflow characteristics of BBMA including

21          altitude information and (2) Diamond (DA42) twin-engine small airplane observations (TU Düsseldorf,

22          Germany) circling around Berlin.

The parameters quantified are listed in Table 3 by measurement platform. All the instruments were calibrated a
priori. Further details as to the timeframe of measurements, instrument information, and parameters measured
can be found in Tables 3 and Table A1 in the Appendix). Finally the observations were classified according the
predominant land use type (see section 3.5 Classification of observed data by land use types).
**3.1    Bicycle measurements**
Bicycles provide a level of flexibility and access to certain areas that cars cannot, in addition to their travel
speed, which allows for well-resolved horizontal resolution of measurement points. Moreover this measurement
type addresses best the conditions where humans are exposed to pollutants. Because of this, they were used as
the basic mobile method for the majority of the time period (10th of June – 29th of August 2014). The instruments
applied for quantifying meteorological and particulate values are listed with their characteristics in Table A1. In
brief, a DiSCmini from Matter Aerosol (CH) was applied for detecting particle number concentrations using a
charged equilibrium in the aerodynamic diameter size range of 10-500 nm. More technical information to the
instrument can be found in Kaminski et al. (2013). The corresponding software supplied an algorithm estimating





the lung deposited surface area (LDSA), a metric linked primarily to smaller particles and their size distribution
providing a measure of potential health effects. Applied as well, was the optical particle counter GRIMM 1.108
(Airing, Germany) for detecting particles in the aerodynamic size range of 0.3–20 μm. The GRIMM 1.108
instrument measured accumulation and coarse mode aerosol particles with a time resolution of 6 s. It included an
additional sensor for air temperature. Both instruments were transported in a backpack or pannier.
A detailed logbook was carried with the instruments and filled out by each cyclist. A Garmin Virb Elite HD
action camera with GPS and WiFi mounted on the handlebar of the bicycle was used to record the exact time and
location of the mobile measurement route and facilitate identification of sources. Please find more details on the
measurements in Appendix A.
The measurement routes covered large parts from south west to the centre of the BBMA area with several
repetitions of a number of the routes (see Fig. 1), such as between the IASS in Potsdam and Berlin-
Charlottenburg. The majority of the routes followed commuter paths to and/or from the IASS. In total 80 routes,
covering 1850 km were obtained during the three month campaign period. The mobile measurements are
viewable online at http://baerlin.iass-potsdam.de.
It should be noted that the mobile measurements represent snapshots for a specific location at a certain point in
time with substantial influence of local sources and sinks. In addition, scaling to daily and annual time periods is
difficult due to the preferred measurement periods in the morning and afternoons, while the sampling frequency
in Berlin-Neukölln was relatively high. Thus, in order to understand the daily pattern of the measured values and
all the contributions in detail microscale simulations would be required. This is beyond the scope of the present
study.
**3.2 Mobile van measurements**
Van measurements were carried out in a one week intensive period between the 31st of July and the 6th of
August using the Research Centre of Jülich mobile laboratory MOBILAB. It consists of a Mercedes Vito van
fitted with an isokinetic particle inlet and gas-phase inlets just above the van roof at ca. 2 m above ground level
(Ehlers, 2013). The following quantities were measured: temperature, relative humidity, ozone, NO, $NO_2$, CO,
$CO_2$, methane, total particle number concentration (2.5 nm - 3 μm), and size distribution of particles between 7
nm and 20 μm in diameter (ELPI). Location data was collected via GPS. A list of the instrumentation is provided
in Tables 3 and A1 (Appendix).
In addition, "baseline" values were derived for CO as well as for the total number and mass of aerosol particles
on the local scale measured in real-time. These "baseline" values were running mean values of the lowest 5% in
a running time period of 180s for minimizing the effect of measurements affected directly by emissions for
instance of cars right in front of the van (Ehlers, 2013).
Each day of the intensive period a specific track was carried out that lasted several hours. The measurement
routes started at the IASS institute in Potsdam and followed cross-sections throughout Berlin and its
surroundings (see Fig. 1 for more details). Some of the focus areas were industrial areas such as Siemensstadt
and Rummelsburg, the Tiergarten tunnel (ca. 50,000 cars/day, Senate Berlin, 2011) and AVUS (ca. 50,000
cars/day at Grunewald and > 80,000 at Berlin West) for traffic emissions, and various urban green spaces,





Grunewald (surrounding streets: <1,000 to 50,000 cars/day, 1 sample), Treptower Park (surrounding streets:
20,000 cars/day, 12 samples) and Pfaueninsel (>1,000 cars/day, 1 sample). In addition to the continuous
measurements canister samples were carried out and analysed for volatile organic compounds (VOCs) by GC-
MS right after return to Jülich (Ehlers, 2013). Further details of the set-up of the van and the analysis methods
can be found elsewhere (Ehlers, 2013; Ehlers et al., 2014; 2015; Barker et al., 2006).
**3.3    Airborne measurements**
Due to technical limitations and restrictions of flight permission over Berlin, air-borne measurements were
carried out at the borders of the investigated region and used to characterize the in- and outflow of particles and
trace gases. Two different platforms were applied, each during a separate period. Both measurement set-ups are
based on long-term experience and included a number of measurements further described below.
The first set of observations was recorded by the KIT ultralight aircraft (Junkermann, 2005; Junkermann et al.,
2011) on the 12th of June (11:53 am – 2:30 pm CEST) during the first days of the campaign. The flight
originated in Schönhagen (EDAZ) south east of Potsdam and followed an eastbound trajectory to Eggersdorf
(EDCE) near Fürstenwalde, from which it returned towards Schönhagen for a repetition of the track further to
the south (see Figure 1). Due to the prevalent weather type on that particular day the outflow of Berlin was
characterised. The aircraft was equipped with a set of instruments for aerosol number and size distributions,
meteorological variables and trace gases (see Table A1; Junkermann, 2005; 2011). The aerosol size distribution
instrumentation, consisted of a WRAS system, GRIMM (Ainring, Germany), measuring the ultrafine fraction
with an Scanning Mobility Particle Spectrometer (SMPS + C, GRIMM, Model 5.403) in the size range from 4.5
to 350 nm and the fine fraction from 300 nm to 20 μm with an optical particle spectrometer (OPS, GRIMM,
Model 1.108). The total number of ultrafine particles was measured with a separate fast (1 sec) condensation
particle counter (GRIMM, CPC4).
The second flight took place on the 10th of October 2014 (9:30 – 10:45 am CEST) a month after mobile ground
measurements had been finished. It was executed by the University of Applied Sciences in Düsseldorf in the
context of a measurement campaign at Melpitz, close to Leipzig, organized by TROPOS (Leipzig, Germany).
Particle size distributions, particle number concentrations, black carbon (BC), sulphur dioxide ($SO_2$) as well as
temperature and relative humidity were measured from a Diamond (DA42) twin-engine small airplane. Air was
sampled using an isokinetic inlet just below the pilots' right window. For details see Weber et al. (2012). Wind
conditions on that particular day were as follows: ground level wind speed was 13 km/h from the south west and
varied between 11 and 33 km/h on the flight level (see Figure S5.1 in supporting online information). The flight
entered the Berlin area in the southeast and continued at the edge of the inner flight control zone making a
clockwise circle around Berlin (see Fig. 1). Temperature and humidity data loggers (VOLTCRAFT, DL-121TH),
unipolar charger and electrometer (GRIMM, NanoCheck 1320, ultrafine particle number concentration, 25 nm <
$D_p$ < 300 nm), optical particle counter (GRIMM, 1.109, accumulation and coarse mode particles, $D_p$ > 0.25 μm),
aethalometer (MAGEE, AE 33 Avio, BC), and an sulphur dioxide instrument (Horiba, APSA-370) measured
continuously with a time resolution of 15 s ($SO_2$) or higher. The prevailing wind direction during the flight





period was from southwest; both inflow and outflow were measured. A complete list of instruments and their
time resolution can be found in Table A1.
**3.4        Method of relative parameters**
Over the course of the three month campaign, measurements were taken by different platforms, at different
locations, under different meteorological conditions, and with different time resolution. To make all the data
acquired comparable and to facilitate comparison independent of meteorological conditions such as daily
maximum temperature, all mobile measurement values were related to the background value of the
corresponding parameters at the reference site in Berlin-Neukölln (von Schneidemesser et al., in prep.) at the
same time. Previous work on analysing mobile measurements (e.g. Van Poppel et al., 2013) has required an
average background value (reference site). For comparison Van Poppel et al. (2013) subtracted this background
value from the measured value (Van Poppel et al., 2013). The result is a direct marker of local changes with
respect to the background site without any possibility for changes by time. Other approaches (e.g. Van den
Bossche et al., 2015) subtract the current pollution level at the background site at identical time in addition to the
method applied by Van Poppel et al. (2013). Except for temperature measurements, for which we applied the
Van den Bossche et al. (2013) approach, we applied the "relative" approach for surface bound observations. The
approach was as follows:
Calculation of the individual relative value by dividing the calibrated mobile measurement by the observation of
the same parameter at the reference site at the corresponding time. For temperature (in degrees Celsius) this was
done by subtraction instead of division as the difference is more representative than the ratio.

20         $\Delta X_{rel} = X(mobile)/X(reference, MC042 \text{ or } MW088)$                    (1)

This method yields not the absolute difference, which varies for different meteorological conditions but the
decreasing or increasing percentage compared to the background site (normalization).
The representative reference site was chosen as a permanent urban background measurement station (Shahraiyni
et al., 2015a; 2015b) of the Berlin Senate, i.e. the aforementioned Nansenstraße monitoring network site in
Berlin-Neukölln (MC042; Berlin Senate, 2015). The long-term measurements from this station (container
MC042) provided reference data for $O_3$, CO, NO and $NO_2$. Additionally, further instruments for the observation
of particle properties (mass, number and size) as well as for quantification of selected VOCs were placed in a
measurement van (MW088, Berlin Senate) parked at a distance of about 5m next to the container MC042 in the
street at the curb. In this way, a reference was provided against which the mobile measurements could be related
to facilitate comparison over space and time. While the mobile measurements were taken on the order of seconds
(see also Table A1) the reference site values were 30 min averages to reduce fluctuations and to minimize short
term effects of the urban background.
The resulting dataset allowed for the assessment of the van and bicycle measurements at different times and
locations to support the identification of different sources and the corresponding regions of impact. All data
'relativized' to the Nansenstraße urban background site in Berlin-Neukölln will be referred to as the 'relative
values' of the urban background reference station.



### 3.5 Classification of observed data by land use types

The mobile measured data were classified according to the CORINE land use map (Bossard et al., 2000; Waser and Schwarz, 2006; European Environment Agency, 2012). CORINE classifies several tenths of different categories of which 15 land use types representative for the area of interest were extracted and partially lumped. The categories relevant are listed in Table 4. The surface classification had a moderate resolution (100m x 100m) and referred to conditions in 2006 (European Environment Agency, 2012). A data point was associated with the predominant land use type for the grid in which it was located. There were three categories of forested areas (coniferous, deciduous and mixed forests), and two categories for urban residential areas (block arrangements (continuous buildings) and single houses (discontinuous buildings)) reflecting the effect of dilution and mixing of pollutants. Once mobile measurement values had been classified, the values were divided by values of identical parameters observed at the reference site in Neukölln at the same time. Results are displayed for classification types with sufficient data (>100 data points, Wilcoxon test) for analysis. Other classification types with partially sufficient data are displayed in shaded colours to indicate tendencies but were not used for detailed discussion.

### 4 Results and discussion

The measurement and analysis results and their discussion will be structured as follows: Trace gas measurements will be presented in Section 4.1, particle measurements in Section 4.2, and temperature measurements in Section 4.3. However there is a strong connection between these parameters, which will be taken into account for the discussion.

### 4.1 Trace gases: VOCs, CO, NOx, and ozone

#### 4.1.1 Volatile organic compounds (VOCs)

Canister air samples were taken at different hotspots of traffic (anthropogenic) and vegetation (biogenic) dominated emission related sites in Berlin. A list of all the species quantified for six selected locations is provided in Table 5. These include two locations dominated by traffic emission (AVUS motorway and "Tiergarten tunnel"), three locations dominated by biogenic emissions ("Grunewald", "Treptower Park" and "Pfaueninsel") and one location for the representative urban background condition in Berlin-Neukölln with both, trees and minor amounts of traffic within the next 150m. A tentative sample was taken in the vicinity of a leaf blower being used, which is a common method for cleaning the pavements. This will be used for interpretation of observations made in residential areas, where a running leaf blower was turned on and may have affected the measurements. It is provided in the supporting online information document (Table S2.1). All compounds were considered to be representative for conditions at background level, where no direct emission sources were expected, e.g. toluene mixing ratios in vegetation dominated areas and isoprene and monoterpenes in traffic dominated areas. If the monitored concentrations exceeded the background level i.e. the level of vegetated areas





unaffected by direct emissions of the corresponding compound (average of the two locations with the lowest
mixing ratios + 2·standard deviation), they were marked in bold. Therefore, all compounds marked in bold
colour represent substantial influence by anthropogenic emissions on the vegetation. The biogenic VOCs and
oxidation products exceeding the average value plus two standard deviations in the traffic related areas are
underlined, indicating substantial impact of BVOCs on the traffic dominated locations. In general, the mixing
ratios of AVOCs observed at the AVUS (motorway in the western part of Berlin) were substantially higher than
for all the other sites e.g. within the Tiergarten tunnel (city centre), Nansenstraße (reference site) or Grunewald.
A compound concentration specific ratio of selected location/reference site larger than unity (= enhancement)
was found between 2 and 27 for non-biogenic species, depending on the individual species. The sample results
show substantially elevated (significance level of ±5%) levels of smaller alkanes, alkenes and alkynes such as
ethane, butane, propene, ethyne and propyne (Table 5). As expected from previous studies (e.g. Caplain et al.,
2006; Stojic et al., 2015, Valach et al., 2015), typical aromatic compounds  like benzene, toluene,
trimethylbenzenes (TMBs), ethylbenzene, and xylenes, as well as several alkanes and alkenes, methyl butene
and ethanol were present in high quantities. Those compounds are related to fossil fuel consumption and are
released either by incomplete combustion or by volatilisation from fuel tanks (Jedynska et al., 2015; Schmitz et
al., 2000). Ethanol can be related to the increased usage of bioethanol in E10-fuel (10% of ethanol). The
situation is similar within the Tiergarten tunnel, although the AVOCs were on average only 38±29% of the
concentration levels at the AVUS. The ratio VOC(Tiergarten tunnel)/VOC(AVUS) is lowest for the most
reactive species (alkenes such as butane and TMBs, 14-17%) and highest for general oxidation products of
tropospheric chemistry (e.g. methanol 91%). Two exceptions were butanol and cyclopentane with +130%,
indicating different sources or a different car fleet within the centre of Berlin controlled by the "Umweltzone",
while independent investigations on vehicle identification numbers did not show a significant change in car types
(Berlin Senate, 2011). Further information about the effect of the Berlin "Umweltzone" can be found elsewhere
(Berlin Senate, 2011).
The "Grunewald" sample was typical for a forested area partially influenced by anthropogenic pollution, i.e.
cross-cut by the motorway: Most VOC concentration levels stayed at quite low values. However, certain
aromatic compounds like p-ethyl toluene, xylenes and ethyl benzene) displayed substantially enriched
concentrations similar to octane. BVOCs displayed a mixture of coniferous and deciduous tree emissions with
isoprene and monoterpenes and their corresponding primary oxidation products (methacrolein, methyl vinyl
ketone and acetone). Different observations have been made for Treptower Park, with even elevated aromatic
compounds levels, i.e. benzene with more than 200 $ppt_v$ and toluene with 275 $ppt_v$. Values have been found
significantly enhanced for smaller alkanes and alkenes including ethane, various derivatives of propane and
general organic oxidations products. Ethyl toluene was significantly enhanced indicating a substantial influence
of the nearby traffic on the vegetation of the park. Its primary biogenic emission was monoterpene with α-pinene
highest among the vegetated sites. This location can be assessed as an exemplary case for vegetation impacted
by anthropogenic emissions. The situation changed with respect to the Pfaueninsel location. Aromatic compound
concentrations were fairly low and biogenic compounds such as isoprene and its oxidation products elevated.
Methanol and acetone were highest among all the samples achieved. It is of interest however, that cyclopentane
and 1-hexene displayed remarkably enriched levels as found in the Tiergarten tunnel samples pointing to present



traffic sources potentially of smaller boats and the ferry close by. While benzene was low, toluene and dimethyl-
pentane were highest for all the vegetated sites. TMBs and n-decane were enhanced too.
Different effects and impacts of anthropogenic and biogenic sources combined at the urban background site at
Nansenstraße. While anthropogenic VOC concentrations at Nansenstrasse were found remarkably smaller than
measured within the Tiergarten tunnel and at the AVUS their amounts were substantially larger than for the
vegetated areas. At the same time BVOCS were elevated because of the nearby trees and plants, similar to
Treptower Park with some exceptions. The BVOC composition changed somewhat because of a different
vegetation composition and structure. α-pinene concentrations were smaller but isoprene and β-pinene
concentrations were comparable to the conditions at Treptower Park. The Nansenstraße samples displayed
higher levels of smaller alkanes and especially high levels of methanol and acetone, i.e. the highest sampled in
this study.
For all the samples performed the individual contribution of isoprene and monoterpenes depend on the
vegetation types, i.e. deciduous or coniferous, and the individual tree types close by (Guenther et al., 1995).
While coniferous trees with elevated monoterpene emissions dominate in the Grunewald area, Treptower Park
and the vegetation close to the Pfaueninsel consists primarily of deciduous trees, thus isoprene emitting ones
(Berlin Senate, 2010; 2013b). The concentration difference reduces for the oxidation products such as methyl
vinyl ketone and methacrolein, which have longer ambient lifetimes and are transported to a larger extent.
The high local levels of methanol and acetone found may be caused by different processes: Methanol is either of
biogenic origin (McDonald and Fall, 1993; Folkers et al., 2008, Holst et al., 2009) or an oxidation product of a
variety of organic compounds e.g. methane and toluene (Atkinson et al., 2006). Acetone is primarily a product of
tropospheric oxidation chemistry of most organic species, including methane too (Atkinson et al., 2006). Direct
emissions of methanol, acetone and acetaldehyde have been reported for forests elsewhere (Gordon et al., 2014,
Rantala et al., 2015), which is expected to be growth related (Hüve et al., 2007). A potential further source are
marshy type (Berlin Senate, 2010; 2013b) such as the flat water-soaked shoreline along the river Havel between
the measurement sites and the Pfaueninsel yielding emissions from decaying organic matter (Warneke et al.,
1999) and water based plant processes (Kreuzwieser et al., 2000). Notable amounts of acetaldehyde and acetone
are expected to have formed by photochemical sources such as ethane and n-butane oxidation (Atkinson et al.,
2006). In order to summarize, the canister samples provided snapshots of the change of the presence of VOCs
with the urban area of Berlin. Depending on the distance of the closest anthropogenic or biogenic emission
sources the corresponding VOCs were detected in significant amounts. Smaller alkanes and alkenes plus
aromatic compounds were found as systematic markers for anthropogenic influence on the different types of
vegetated urban areas, i.e. parks and forests. Biogenic VOCs were found in substantial concentrations in all of
the locations and provided a significant contribution to urban background VOCs.
**4.1.2  Carbon monoxide (CO)**
The mobile observations displayed a similar pattern for all the traffic related gases (NO, $NO_2$, AVOCs and CO)
and particulates. In addition to its major urban source from traffic (incomplete combustion of fossil fuels; Klemp
et al., 2012) CO may originate to a smaller extent from photochemistry (Finlayson-Pitts and Pitts, 2000;





Atkinson et al., 2006; 2006). Because of its longer atmospheric lifetime (1-4 months, Seinfeld and Pandis, 2006)
compared to standard reactive gases such as ozone and $NO_x$ emissions and atmospheric transport are the
dominant processes influencing the spatial distribution of CO. The elevated CO mixing ratios due to high traffic
intensity can be seen in the horizontal distribution of CO (all the measurements, CO all) and dampened in plots
with the lowermost 5% of a running mean (180 s) (named 'CO baseline' in the following) (not shown). These
running mean ("baseline") values excluded values affected by direct emissions from for instance nearby cars
(Ehlers, 2013). The mobile measured values ranged between 100 $ppb_v$ and 43.8 $ppm_v$ for CO all and between
100 $ppb_v$ and 3.8 $ppm_v$ for CO baseline. The corresponding relative values of CO varied between 0.01 and about
230 times the value at the reference site in Neukölln. The spatial heterogeneity of the relative CO mixing ratio is
displayed in Figure 2 on the top as general overview and at the bottom zoomed over a smaller area including the
reference site in Berlin-Neukölln. Higher relative mixing ratios, likely owing to vehicle emissions in close
proximity and/or substantial traffic emissions, are indicated by reddish colours and were found at hotspots at
major traffic routes and crossings such as Zoological Garden, AVUS, Frankfurter Allee, Urbanstraße and
Mehringdamm (Fig. 2). Unity identifies points with identical volume mixing ratios measured from the van and
measured at the Neukölln station (BLUME station). For illustration purposes, the ratio of 1 (identity) ±10% is
depicted in blue, smaller values are shown in yellow and higher ones in red.

### 4.1.3    Nitrogen oxides

The situation is similar for nitrogen oxides ($NO_x$). In the BBMA $NO_x$ are emitted primarily from vehicles,
specifically fossil fuel based internal combustion engines (Tullius and Lutz, 2003). Figure B1 of the appendix
displays the mixing ratios of NO and $NO_2$ (top), as well as the relative values (bottom). Several locations had
elevated mixing ratios and relative values: (i) the Tiergarten tunnel with accumulation of pollutants and
substantial amounts of traffic, (ii) the "Straße des 17. Juni" across the Tiergarten and its continuation as "Unter
den Linden" with a significant number of public transport and tourist busses and older vehicles, the major traffic
routes such as (iii) "Frankfurter Allee" (East), (iv) "Mehringdamm" (South), (v) "Westkreuz" and (vi) AVUS
(West) as well as (vii) around the Central station. The individual locations are indicated in Figure B1 as far as
they are included in the plots.
The observed median mixing ratios ranged between 5.6 $ppb_v$ for NO and 0.7 $ppb_v$ for $NO_2$ in more remote
locations with little traffic, and 2.1 $ppm_v$ NO and 2.9 $ppm_v$ $NO_2$ in locations characterized by significant traffic,
including in some cases traffic hubs at the intersection of major roads coinciding with bus terminals and other
public transport infrastructure, e.g. "Hardenbergplatz" near "Zoologischer Garten". The relative values ranged
from 0.5 to 4000 for NO and 0.2 to 500 for $NO_2$. The median relative values were 15.1 for NO and 1.3 for $NO_2$
for all measurements, whereas the median of the relative values for NO and $NO_2$ for areas classified as urban
(continuous and discontinuous buildings) were 23.9 and 19.2, and 15.7 and 12.2 in commercial areas &
transport, respectively (Figure 3). Therefore we can state that for most of the road related area analysed the
mixing ratios were found to be higher than in Neukölln except the remote background sites such as forest or
agricultural areas out of Berlin.
The emitted NO reacts with ambient ozone to be converted into $NO_2$. The $NO_2$ can subsequently be rapidly
photolyzed back to NO and $O(^3P)$, which subsequently re-forms ozone, in the case of sufficiently strong solar



radiation. Through these reactions, ozone is rapidly consumed by reaction with NO, if and only if NO is present
in substantial amounts, and re-formed by the subsequent photolysis of $NO_2$. In a photostationary steady state
(PSS), with no additional sources of $NO_2$, the relative mixing ratios of NO, $NO_2$, and ozone are determined by
the photolysis rate of $NO_2$. NO measured on the road was often found to be approximately ≥15 times higher than
the values recorded at the Neukölln urban background site and therefore converted to $NO_2$ by ozone if it was
sufficiently available. However, the low ozone mixing ratios observed at the road sites, where high levels of both
$NO_x$ species were measured cannot be explained by the PSS chemistry alone, suggesting the presence of an
additional source of $NO_2$. In this case the change in median $NO_2$ mixing ratio between urban residential areas
with block buildings such as at the reference site at Nansenstraße is +39.6 $ppb_v$, while the decline in median
ozone volume mixing ratio is only -7.2 $ppb_v$, i.e. an excess of 32.4 $ppb_v$ with respect to $NO_2$ assuming a constant
Ox value throughout the city (absence of substantial $NO_x$ emissions). Thus, the expected $NO_2$ would be 23.9
$ppb_v$ and the measured median one is 63.5 $ppb_v$. It is well known that substantial amounts of $NO_2$ are produced
and released by oxidation catalytic converters of Diesel cars (Li et al., 2007). The car fleet of Berlin residents
and companies consists of 29.9±3.5% diesel driven passenger cars and 93.1±0.7% of diesel consuming light-
duty commercial vehicles (Berlin Senate, pers. comm.). Based on the study of Tullius and Lutz (2003) it is
expected that this source type contributes significantly (33%) to the measured nitrogen oxides mixing ratios
especially in urban areas with notable traffic and transport. Other sources like the energy industries, non-energy
combustion, non-road transportation and industry provide the remaining 67% of $NO_x$ production excluding ship
emissions for which Tullius and Lutz (2003) did not have information. However a detailed calculation of the
$NO_x$ budget was out of the scope of this campaign.
**4.1.4    Ozone ($O_3$) and Ox**
The observed volume mixing ratios of ozone are a product of the different compounds discussed earlier on:
VOCs (4.1.1), $NO_x$ (4.1.3) and CO (4.1.2). Their photochemical reactions (Finlayson-Pitts and Pitts, 2000;
Atkinson et al., 2004; 2006) depend on different intensities of local surface sources and sinks as well as on the
transport of ozone and precursor species from other areas. As ozone is not directly emitted and forms exclusively
in the gas-phase, it is secondary in nature. As described in Section 4.1.3, the urban ozone mixing ratio is closely
related to the mixing ratios of NO and $NO_2$, as well as the photolysis rate of $NO_2$. When the PSS holds, the
mixing ratios of ozone and $NO_2$ vary inversely with each other, with their sum being constant. This sum is
referred to as "Ox", which is generally an invariant background quantity in the area of observation, as long as the
PSS holds. Because of this ozone and Ox were expected to vary in space to a smaller amount than for instance
CO and NO.
Mobile measured ozone mixing ratios were quantified between negligible values and 62 $ppb_v$ (Fig. 4). Median
values for urban residential areas were situated around 28 $ppb_v$ for ozone and 52 $ppb_v$ for Ox. The highest
median ozone values were found in agricultural areas 46.5 $ppb_v$ and the lowest in the vicinity of parks 17 $ppb_v$.
Sport and leisure facilities displayed notably enhanced concentrations, while much lower concentrations were
observed in forests (23-27 $ppb_v$) with the exact values depending on the forest tree type (Figure 5). Nevertheless
notable variations of -70% to +30% of the mixing ratio of $O_3$ relative to the $O_3$ measured at the reference site in
Neukölln (MC042) were identified within the surrounding 500 m (Fig. 5).





### 4.1.5    Land use type effect on relative changes for gaseous pollutants

Trace gas mixing ratios displayed clear differences between different land use types. As frequency plots of measured gaseous mixing-ratios (not shown) provided evidence of non-normally distributed values, we provide median and $25^{th}$ as well as $75^{th}$ percentile values throughout this study. Figure 3 shows median values and interquartile ranges for CO, NO, and $NO_2$. Both, the measured values and the relative values are shown. As noted above, this relative approach allows for comparison of different conditions by reducing meteorological and large scale effects. In this context we assume meteorological conditions to be of minor relevance for the relative value approach used.

Several aspects are evident: Carbon monoxide and nitrogen oxides displayed quite similar land use type effects and therefore similar conclusions can be drawn. CO baseline and CO all mixing ratios (Fig. 3, top) were lowest in areas covered with vegetation, i.e. agricultural fields (-21% compared to urban values) and forests (-6% with respect to urban conditions), where emissions by traffic sources are expected to be smallest and dilution during the transport strongest. Baseline CO values were clearly elevated in industrial (+10%), commercial and transportation (+78%) affected areas. The effect on nitrogen oxides (Fig. B1, NO (middle) and $NO_2$ (bottom)) was even more pronounced and lower in areas covered with vegetation such as agricultural fields and forests (ca. -75%), and were highest in industrial (+86%) and commercial and transportation (+190%) affected areas. The relative $NO_x$ values for residential urban areas were located between the higher values in the industrial, commercial and transportation affected areas and lower values in forests with increasing ventilation reducing the $NO_x$ concentrations notably. Nearby parks nitrogen oxides were found at elevated levels (+100%) likely owing to nearby traffic sources and accumulation (less ventilation, reduced photolysis rates near the surface). Finally, median $NO_x$ levels in- and outside of Berlin were investigated to indicate the predominant local source: Total $NO_x$ in urban residential areas and the measured roadside $NO_x$ showed increases of 293 % and 906 % relative to forested areas depending on the type and the route, respectively. This presents evidence that dominant summertime $NO_x$ sources in Berlin are local and these are likely to originate primarily from traffic.

Measurements of Ox (sum of $NO_2$ and $O_3$) were constant, i.e. within the range of the notches, over most surface types, at about 51 $ppb_v$ (Fig. 5). Exceptions to this were forests, where median Ox varied between 42 (deciduous and coniferous) and 52 $ppb_v$ (mixed), and areas classified as "commercial and transport", where median values reached about 84 $ppb_v$. Within and near forests, observed ozone mixing ratios declined along with Ox mixing ratios, consistent with a sink of ozone due to reaction with biogenic VOCs (see below). Over "commercial and transport" areas, observed ozone mixing ratios increased together with Ox mixing ratios (median increase by about 40 $ppb_v$, see section 4.1.3 for details), which together with elevated levels of $NO_x$, is consistent with a local primary source of $NO_2$ due to diesel vehicles resulting in a 40 $ppb_v$ higher $NO_2$ value than expected from the PSS.

Ozone mixing ratios over land surface types other than forests and "commercial and transport", were quite different from each other, despite having similar levels of Ox. Ozone substantially increased (+30%) in agricultural areas and decreased in industrial (-27%) areas, while $NO_2$ decreased (-28%) in agricultural areas and increased by +100% in industrial areas. Agricultural areas are predominantly located in the outskirts of the city,



and are characterised by low mixing ratios of $NO_x$. Industrial areas are generally closer to the centre, and display
higher $NO_x$ mixing ratios (Fig. 3). For a certain group of surface usage types (urban residential areas, industry,
airport and parks) the Ox value stayed constant within 5% confidence interval based on-road measurements, for
which an inverse relationship between the mixing ratio of $NO_x$ and the mixing ratio of ozone was found. This is
consistent with the assumption of $NO_x$-saturated chemistry over an urban area, in which increased emissions of
NO lead to "titration" of ozone. A different situation was found for sport and leisure facilities such as Tempelhof
Air Field and Olympic area in Ruhleben, where Ox was significantly smaller (relative Ox = 0.88, i.e. -12%) with
$NO_2$ reduction by 5.1% and ozone reduction by 36.9%. Either the situation is non-$NO_x$-saturated or the PSS was
not achieved.
Ozone mixing ratios and nitrogen dioxide in forests are lower than those in the urban background (exception
mixed forests), while the relative Ox change (-24.6%) that is smaller than the reduction in $NO_2$ (-26.7%) and in
$O_3$ (-31.4%) (Figs. 3 and 5). The van based observations indicate a $NO_x$-limited ozone production scheme, which
is associated with notable emissions of terpene and of isoprene. This may be caused by the vegetation stress
response to ozone pollution (Bonn, 2014) with an enhanced release of VOCs (Guenther et al., 1995) reacting
with ozone and thus acting as a local sink for ozone, and thus Ox. According to Bourtsoukidis et al. (2012)
coniferous spruces start emitting notably more amounts of terpene species above 35 $ppb_v$ of ozone, with the
emission controlled more and more by ozone the higher the stressor gets. Subsequent to emission, those
particularly high reactive terpenoid species destroy ozone (Atkinson et al., 2006) and their oxidation products
form new aerosol particles in number and secondary organic aerosol mass (Bonn and Moortgat, 2003; Griffin et
al., 1999; Sakulyanontvittaya et al., 2008). Total $NO_x$ is also lower in forests (Fig. 3), likely due to lower
emissions from vehicles. Combined with the sink of Ox, this leads to lower mixing ratios of $NO_2$ in forests. A
similar reduction in Ox over 'parks' such as the 'Tiergarten' area is not seen in the BAERLIN results (Fig. 5).
Generally, ozone concentrations in parks were found to be significantly smaller (-36%) than in Neukölln, while
NO and $NO_2$ were enhanced by +98 and +86%. Parks seem to represent a mixture of urban residential areas and
the nearby traffic sources, with the corresponding median Ox value is within the ±5% range of 'background' Ox
values.
Our results indicate that the main impact of vegetated areas on gaseous processes and concentrations in Berlin
may be the chemical deposition effect that is particularly high over monoterpene emitting coniferous forests
within the urban area. The relatively small ozone reduction found in the vicinity of isoprene emitting deciduous
trees may indicate that ozone formation partly compensates ozone deposition at these sites.
**4.2    Particulate pollution**
Aerosol particles were potentially among the most challenging aspects of this campaign, as they can originate
from long-range transport and local primary particle sources or gaseous precursors, depending on individual
sources and are influenced by the presence of clouds and precipitation and their loss depend on size and
hygroscopicity too. They may possess an atmospheric lifetime of up to ten days and secondary (semi-volatile
gaseous) constituents adjust their phase distribution according to temperature and their vapour pressure related
gaseous mixing ratios. Thus, for a detailed analysis of interactions between secondary organic particular mass
and ozone for example a box model approach is required to track the related semi-volatile compounds in both





phases. This box model analysis is beyond the scope of this article, which will serve as characterising the
conditions for a follow up publication (Bonn et al., in prep.). The different mobile approaches were implemented
to look at small scale variation across the city and so to help distinguish local traffic from large scale sources.
The airborne measurements were used for identifying the transport of pollutants and precursors into the entire
area and to gain insight into the dominant sources at elevated levels.
### 4.2.1    Airborne measurements: City sources vs. transported particulate pollution
Upper boundary layer values of total particle number concentrations (PNC) above 4.5 nm in diameter ($D_p$) in an
altitude of about 300 to 500 m above ground were about 2,500 particles cm$^{-3}$ (PNC$_{4.5}$ displayed as UFP, Fig. 6)
at the edges of the city of Berlin and increased to 9,000 – 12,000 cm$^{-3}$ when passing through the Berlin city
plume. The city plume values were in agreement with the ones found at urban background conditions in
Neukölln (PNC$_{10}$, 8,800±5,000 cm$^{-3}$, $D_p$ > 10 nm), indicating a very similar atmospheric composition and a
minor contribution of particles between 4 and 10 nm in aerodynamic particle diameter. The mixed layer height
on the 12$^{th}$ of June (ultralight aircraft flight) was about 1,500 m a.g.l. (derived from HYSPLIT (Draxler and
Rolph, 2013) and observed cloud base temperature). Additionally on top of this city plume two well defined
plumes were observed on both flight sections W→ E (ca. 12:45-13:00) and returning E→W (ca. 13:15-13:30)
with a PNC$^{4.5}$ of  35,00-55,000 cm$^{-3}$ and a PNC$_{10}$ of 35,000 cm$^{-3}$ with a geometric mean diameter (GMD) of 12
nm and 45,000 cm$^{-3}$ at a GMD of 15 nm. These two plumes were nearly exactly downwind of the two coal fired
power stations located inside the city limits, Reutter West (600 MW), North of the Olympic Stadium in Berlin-
Charlottenburg (West, N 52° 32' 6.25"  E 13° 14' 30.59"), and Klingenberg (680 MW), in Berlin-Rummelsburg
(East, 52° 29′ 24″ N, 13° 29′ 42″ O). This was concluded from back trajectory analysis (HYSPLIT; Draxler and
Rolph, 2013) obtained for different positions of the flight. The observed small geometric mean particle diameters
clearly indicate recent particle number formation events, which are likely to have occurred either within the
exhaust chimney or within the plume of the individual power plant. The same applied for the airplane
measurements in October. When crossing the corresponding areal sector of a power plant with south westerly
winds (Fig. S1), number and mass concentration increased significantly with the sulphur dioxide (SO$_2$)
concentration displaying small jumps as the ultrafine particle number concentration (UFP, 25nm < $D_p$ < 300 nm)
increased (Fig. 7). However, SO$_2$ was higher at smaller UFP and so this was apparently not the limiting quantity
for nucleation. At the times of highest UFP the smallest concentration of BC was found on that particular flight,
indicating smallest total particulate mass burden and coagulation sink potentially allowing the newly formed
particles to survive longer and to grow before being captured by pre-existing larger particles. Up to 4000
ultrafine particles cm$^{-3}$ were detected at around 1,500 m altitude, i.e. around the uppermost mixing layer height
or slightly above. The flight around Berlin on the 10$^{th}$ of October 2014 (see Figs. 2 and S1) displayed a strong
gradient throughout the BBMA area (Figs. 7 and B1), from rather clean conditions around 1,000 cm$^{-3}$ downwind
of BBMA to 4,000 cm$^{-3}$ in UFP upwind of BBMA, i.e. behaving the opposite to major PM sources. These
sources would enhance the condensation sink, i.e. reduce the lifetime, of condensable species and a new particle
formation would occur either in a much smaller intensity or would be prevented (Kulmala et al., 2001; Lehntinen
et al., 2003). This clearly indicates that the majority of sources have to be found within the city boundaries at





both times of measurements (Figs. 6, 7 and B1). This is confirmed and even highlighted by the ultralight aircraft
flight on the 12th of June at a lower altitude, with total $PNC_{4.5}$ reaching up to 45,000 $cm^{-3}$, a value that is similar
to urban concentrations above a particle diameter of 10 nm within continuous traffic situations.
In contrast to $PNC_{4.5}$, $PM_{10}$ and $PM_{2.5}$ flight level mass concentrations were substantially lower ($PM_{10}$: ca. 8
$\mu g/m^3$, $PM_{2.5}$: ca. 6 $\mu g/m^3$) compared to the value detected at the surface at urban BLUME monitoring sites in the
city centre ($PM_{10}$: 20-25 $\mu g/m^3$). Particle mass concentrations were similar to concentrations observed at the city
border on the flight day at measurement stations in Grunewald (west) and in Friedrichshagen (southeast) with
$PM_{10}$ values between 9 and 10 $\mu g/m^3$ (BLUME, von Stülpnagel et al., 2015). Only in the Northeast of Berlin
(downwind of the city), close to Buch and Bernau, were moderately elevated mean concentrations of 16 $\mu g/m^3$
observed at the surface measurement stations (BLUME, von Stülpnagel et al., 2015) and at flight level (15
$\mu g/m^3$, this study). As a conclusion of this part, it can be stated clearly that there was a moderate background
concentration of particle number and particular mass, which increased substantially inside the Berlin area
because of city based emissions.
**4.2.2     Van measurements: The regional and local situation**
Next we focus on surface bound measurements on the regional and local scale. As the elevated particulate
number and mass concentrations tend to originate locally, the van measurement facility of RC Jülich quantified
particulate number and mass concentrations for cross-sections of the BBMA area. Simultaneous gas-phase
detection supported the attribution of sources and sinks. An overview of median particle number and mass
($PM_{10}$) concentrations and their variations is given in Tables 6 and 7. Again the parameters displayed a non-
normal distributed frequency plot. Details for further parameters such as PM1 can be obtained from the
supporting online information. The local sources and the heterogeneity of particle concentrations in the city
become obvious in Figs. 8, C2 and C3. As done for the standard gases measured, all measurements have also
been divided ("normalized") by the measurements at the reference site at the same time for comparison. Relative
particle number concentration ratios (relPNC (2.5 nm < $D_p$ < 7 $\mu$m (NanoCPC) vs. 4 nm < $D_p$ < 3 $\mu$m
(GRIMM5.416)) were gained from different instruments with different cut off sizes. However with respect to
particle number concentration, particles above a diameter of several hundred nm have been found negligible for
most conditions when compared to particle numbers at smaller sizes (Seinfeld and Pandis, 2006; Friedlander,
2000). Due to intense emissions in the urban area and the subsequent coagulation of smaller partially unstable
particles the detection of sizes between 2.5 and 4 nm in particle diameter is usually scarce and the vast majority
of particle number is located between 50 and 100 nm. Comparisons of both types at the reference site displayed
no significant difference between both observations used for comparison, i.e. the NanoCPC by RC Jülich and the
GRIMM 5.416 by UBA.
The focus is set on the baseline concentrations (lowest 5% in a moving 180s time interval) as for CO, as particle
number concentrations e.g. directly behind an emitting passenger car provide short term peaks that last for
seconds only and which are hard to compare with measurements on the flight level (always off from the emitter,
thus baseline) or at the urban reference site. Nevertheless, ranges for peak values will be noted as available.
RelPNC values found for the van measurements ranged from about 30% of the urban reference value outside of




the area of Berlin, to the 85fold in areas with substantial traffic density and in street canyons with less
ventilation. Peak values exceeded the 200fold concentration of the reference site. Highest values appeared at
motorways and the primary entering routes into Berlin, i.e. Hohenzollerndamm, Hasenheide, Karl-Marx-Straße
and the neighbouring streets in Kreuzberg and major crossings such as the Hardenbergplatz (Zoologischer
Garten).
Similar patterns but much more moderate increases have been seen for particulate masses. This can be explained
as follows: As remarkable fractions of particle mass are of secondary organic origin (mass closure at reference
site in Berlin-Neukölln: 38.2±9.4%; Kofahl et al., 2012; von Schneidemesser et al., in prep.), new particle
formation and particle mass production require different process times and sink strengths. The formation process
of particles is rapid and occurs in seconds to minutes depending on the precursor concentration and is suppressed
by high coagulation sink values i.e. substantial particulate matter already present. Mass production requires
gaseous and multiphase oxidation with a timescale of minutes to days and present particular mass to condense on
(inorganic species) or partition to (organic compounds). Consequently, depending on source strength the
observed relationship between source and PM may result in a smeared picture in the vicinity (tens of metres) of
sources, with greater enhancement for particle numbers.
### 4.2.3  Bicycle measurements: Quantifying the microscale on different timescales
As different areas have been probed by the van measurements two aspects could not be addressed: (i) Pollution
levels in different distances to major sources such as cars, (i.e. roads, cycling paths, bus stops, walking path and
parks) and (ii) the temporal variation at a particular site of interest. The bicycle measurements were able to
contribute to answering those points. For comparison with van measurements, two routes were done with both
platforms, i.e. bicycle and van, of which the 4[th] of August is shown in Fig. 9. As the bicycle measurements
focussed primarily on particle classification, two graphs are shown, the left one for total particle number
concentration (bicycle: DISCmini (10nm $< D_p <$ 500nm) + GRIMM 1.108 (500nm $< D_p <$ 20μm), van: (a)
nanoCPC (2.5nm $< D_p <$ 3μm) and (b) ELPI (30nm $< D_p <$ 10μm), see Tab. 3) and the right one for particulate
mass below 10 μm in diameter (GRIMM1.108 ($D_p >$ 270 nm) vs. ELPI, Tab. 3). The van measurements of
particulate mass were considered twice, i.e. all the measurements and the lowest 5% (bg = baseline) in a moving
3 min period to exclude peak values. In this case the PM$_{10}$ comparison is displayed.
Please take into account that the measurements were conducted with the van following the cyclist at street level
in order to exclude the vans exhaust. The different heights of the inlets for van around 2 m a.g.l. (Ehlers, 2013)
and cyclist measurements at about 1m a.g.l. had impact on results very close to the sources. While the baseline
values i.e. NanoCPC and DISCmini for number concentrations and GRIMM1.108 and ELPI for total particulate
matter measured by the different platforms agreed well, peak values showed only a moderate agreement. As
noted above the causes for these differences were most likely small and short term pollution drops, i.e. strong
horizontal and vertical changes, as measurements were performed next to the location of particle number
formation with rapid particle dynamics and growth processes associated with. Moreover both platforms were not
always able to drive right next to each other because of traffic density and changing lanes. As can be seen in Fig.
9 on the left the comparison of both total number measurements of the van i.e. NanoCPC and ELPI disagreed in



magnitude because of the different cut-off limits of both instruments. While the upper limit was less critical for
total number concentration, the major effect was caused by the difference in lower detection limit with 3nm for
NanoCPC, 10 nm for the DiSCmini and 30 nm for the ELPI with respect to the lowest particle diameter
detectable. As freshly formed new particles from traffic are expected to appear at sizes below 30 to 40 nm in
diameter the notable gap between DiSCmini and ELPI instruments became important (Fig. 9, left plot). With
respect to total aerosol mass, displayed here as PM10, the van results (ELPI) are slightly higher than the bicycle
observations (GRIMM 1.108). Baseline values were enriched by $16.4\pm0.1\%$ and all values measured by
$58.1\pm0.2\%$. This can be traced back to the different detection range of both instruments with the ELPI including
particle masses between 0.03 and 0.3 µm and excluding the size range between 10 and 20 µm, and the different
time resolution, $\Delta t(ELPI) = 1s$ and $\Delta t(GRIMM1.108) = 6s$. Because of this, smaller disturbances and emissions
of passing vehicles caused a larger variation in van (ELPI) data than for the bicycle data (GRIMM 1.108) and the
latter agreed better with the baseline ELPI values as stated above.
Two benefits of combining bicycle and van measurements were as follows: (1) the different speed at usual
conditions and thus an improved horizontal resolution and higher local variation of data with respect to particle
number concentration of bicycle measurements at identical time resolution and potential parallel observations at
identical time at different distance to the particle source; (2) the impact of the particles lifetime on the agreement
of number (relevant for smaller sizes) and mass (relevant for larger sizes) concentrations observed by the two
platforms. With respect to (1) the exhaust of for instance a car gathering speed and therefore contributing to new
particle formation was found to enhance the $PNC_{2.5}$ of the van in a more intense way than for $PNC_{10}$ for bicycles,
while hardly any change between both platforms was seen with respect to the mass (2).
While in general the complete area was of interest, several routes were frequently sampled by bicycle: (1)
Potsdam – IASS to Berlin-Charlottenburg (26 times), (2) a central round track in Berlin including Schöneberg,
Kreuzberg, Neukölln and Wilmersdorf (8 times) as well as (3) a cross-section of Berlin from Berlin-
Charlottenburg via Tiergarten to Friedrichshain (11 times). In the following we concentrated on those tracks with
the entire datasets (i.e. GPS and particle properties) with no gap in between were obtained and track (1)
especially for demonstration purposes (Fig. 10).
During route (1) the focus was set on the differences between residential and forested areas as well as motorways
(next to), while during track (2) the surrounding of the measurement site in Neukölln and the multitude of parks
were of particular interest. Route (3) was an exemplary study of changing the full range of conditions from the
west to the east driving along the major roads such as "Straße des 17. Juni" and "Unter den Linden". It started at
"Heerstraße" - one of the most frequented roads-, continued through the Tiergarten area with urban green space
and terminated east of the touristic hotspots of Brandenburg Gate and Alexanderplatz.
As an exemplary track route (1) is shown (Fig. 10) and discussed. Forested and vegetated areas evidently
displayed reduced pollution levels because of reduced sources and enlarged sinks. Relating distances to
locations, the first maximum was seen (upper plot) close to Berlin-Charlottenburg at its western edge, crossing
major transport ways of railway and cars (motorway AVUS). Subsequently, the cyclist rode across the
Grunewald – a forest of substantial size (see above) – that is crossed by the motorway to Potsdam in its south
eastern part. The cycling track was always west of the motorway inside the forest. During the transfer from
forest paths to streets influenced by notable traffic (30,000-40,000 cars/day) near Wannsee, pollution levels





increased remarkably with intermittent peaks due to passing busses and cars until reaching the Glienicker Bridge
when entering Potsdam. Once wind was opposite to the predominant direction from the North West (300°, Fig.
10 lower graph) the pollution was transported from the nearby motorway to the bicycle track through the
Grunewald forest more efficiently (south easterly winds, 140°). At one of two times with exceptionally high
particle numbers in Grunewald (morning of the 14th of August, track 61 at Tempelhof) no wind direction was
available from the record by the German Weather Service. However, before the gap starting at 4 am, i.e. four
hours before the measurements the wind direction was around 160-180° (south easterly), agreeing with the other
case of increase $PNC_{10}$.
### 4.2.4   Aerosol particle pollution characteristics and effect of land surface types
The aerosol particle burden quantified matched fairly well with pollution levels measured in a range of Central
European cities (e.g. project UFIREG; UFIREG 2014). Our measurements provide evidence that both mobile
measurement types can be combined for characterizing different land use types and for checking the reliability of
the results from each method.
As done in the case of the gaseous parameters the observations were classified by the land use type based on the
CORINE land cover (number: Fig. B2, mass: Fig. 11). Several aspects are particularly obvious: The urban areas
with block buildings (continuous buildings) show a slightly enhanced particle level compared to areas with
single houses (discontinuous buildings). Particle number concentration also tends to increase with decreasing
street canyon width. This is apparent in the city centre comparing different street positions with a changing
width, e.g. starting at Fehrbelliner Platz (widest) along the Hohenzollerndamm via Lietzenburger Straße (+26%
in PNC compared to Fehrbelliner Platz) until Urania (denser, +96% in PNC compared to Fehrbelliner Platz).
This indicates that ventilation effects do play an important role and thus need to be considered for urban
planning. Second, changes in particle concentrations were largest for particle number concentrations. They were
found to be highest in industrial and commercial areas with notable transport and production of goods (e.g.
Kurfürstendamm, Kantstraße, Greifswalder Straße and Frankfurter Allee): +47±11% for baseline concentrations
and +63±10% for all the observations (van). Observations in sport and leisure facilities did not display
significant changes, but the ones in parks resulted in a different outcome for bicycle based measurements (-
15±7%) and for van based measurements (baseline: no significant change, all: +55±28%), which evidently feeds
back to the coarse surface classification of CORINE (Bossard et al., 2000) with 100m x 100m. While the van
was unable to enter the parks and measured at the edges on the road, the cyclist passed through. In this case the
bicycle seems the more appropriate platform, while for industrial areas and transport the van was optimum. For
forests the particle number effect was identical within the confidence limit (bicycle: 0.67±0.04, van(baseline):
0.72±0.04; van(all): 0.66±0.10). It can be concluded that in vegetated areas PNC declined substantially except
for sport and leisure facilities and that PNC was enhanced substantially in industrial, commercial and transport
affected parts of the city.
The situation was less clear for particulate masses as e.g. PM10. While no significant change was observed for
van(all) values between any of the types, van(all) was significantly smaller in parks (0.91±0.02) but not in
forests (1.02±0.09) or sport and leisure facilities (1.01±0.01). Only the bicycle measurements displayed some
expected changes: PM10 increased in industrial, commercial and transport affected areas by 44±9% and





decreased for parks by 18±8% and for forests by 35±8%. A tendency but no significance for decrease was found
in sport and leisure facilities (-17±14%). Thus we can conclude that vegetation affected surface areas reduced
substantially the burden of particulate mass and particle number displayed best by bicycle measurements.
Differences between the bicycle and van platform data likely stem from the circumstance that different tracks
had to be used and the coarse CORINE classification treating sometimes even major roads next to a park as park
surface.
This was for example the case for "forests". Rather large motorways crossings exist in some forests such as the
Grunewald, and the van was either passing the forest on the western edge at the river Havel or on the motorway
in the centre, while the cyclists always rode through the forest on a bicycle only route. At a moderate resolution
(100m x 100m) this difference is not visible on the surface map used for classification (CORINE). For this
reason, we consider the particle measurement values for forests to those obtained by the cyclists as a better
proxy. Those are in line with what would be expected for a low-pollution land use type with concentrations
between 55% and maximum 70% of the values found in Berlin-Neukölln.
Initial analyses of individual bicycle videos have indicated primarily traffic related sources for occurrences of
high concentrations of particles, such as old double decker busses, mopeds and single ships when crossing
bridges. Detailed results from this investigation will be published elsewhere.
**4.3     Temperature**
Bicycle and van measurements were combined to investigate the presence of the urban heat island effect
(Collier, 2006; Seinfeld and Pandis, 2006). For most of the land use type classifications the differences between
the van and bicycle measurements agree within the associated uncertainty. While the urban heat island effect was
present, it was not as pronounced as in other metropolitan areas such as Paris or London (Dousset et al., 2011;
Jones and Lister, 2009) with about +4 °C (Paris) and +2 °C (London, Jones et al., 2009) and ca. +1.3 °C in
Berlin (von Stülpnagel, 2015), possibly attributable to the excellent ventilation and substantial green areas in
BBMA. The change in temperature classified by surface types can be seen in Fig. C1. Please note that the
observations have been made at daytime, which resulted in larger temperature differences between different
surface usage types than for the average day within the Berlin metropolitan area (Fenner et al., 2015). A map
(not shown) with the differences of measured temperature by mobile platforms and in Neukölln displayed only
moderate changes as wind tends to equilibrate and changes were limited to tenths of degrees C on the local scale.
However, certain land use types and wind circulation conditions enhanced cooling and mixing of air in the case
of forests, larger park areas and areas with nearby water bodies. For instance forests are found to be cooler by ca.
2 °C while commercial and transport affected areas are found to be warmest (+0.3 °C). This is in agreement with
the study of Dousset et al. (2011) stating that an increase of vegetation by 1% would result in a cooling effect of
0.2 °C in Paris. For some land use types such as for parks and commercial areas the results of both measurement
platforms deviate substantially from each other. This is likely caused by a) the different time scale of
measurements and time of day, as well as b) partially by using different pathways, e.g. by proceeding faster on
the bicycle route, while the van is stuck in dense traffic, with the asphalt street heating up more in the sunlight.
There is a difference in the surface temperature measured in urban areas with continuous buildings measured by
the mobile devices and the reference station of -0.3 to -0.4 °C, i.e. the reference site displays warmer



temperatures. This is caused by the shielding container set-up with only gently air movement present due to the
surrounding vegetation, while at the street level in the shadow the surface may cool down easier, as the same
temperature difference was observed between the temperature observed on top of the permanently parked van of
Berlin Senate and on top of the container at Neukölln.
**5    Conclusions**
The mobile measurements with bicycle, van and air plane/glider as part of the BAERLIN2014 measurement
campaign has demonstrated the ability of integrated measurement platforms to characterise air quality on
multiple scales. Van-based measurements were used to cover a large geographical area in and around the city of
Berlin, while bicycles, covered a range of main streets, but also penetrated to areas inaccessible for cars
(pedestrian areas, parks and forested areas). Bicycles were found to be a cheap, flexible and reliable platform for
characterising the spatiotemporal variations in pollutant concentrations and meteorological conditions over the
three month campaign period.  Comparison of van and bicycle measurements (particulate properties and
temperature only) agreed within the uncertainty level when measured under identical conditions. The "relative
value" approach used for individual parameters to compare different measurements of trace gases and aerosol
particle by different platforms in different conditions was found to be very applicable to our observations.
During the period of investigation the elevated air pollutant concentrations found in Berlin were most likely
produced in the vicinity of the observation and originated from local pollutant sources. Air plane measurements
displayed additional substantially elevated particle number concentrations in air masses from both coal-fired
power plants and within the flight corridor of Tegel airport. Van and bicycle based measurements observed
elevated particle number and particulate mass concentrations as well as NO, $NO_2$, CO and VOC mixing ratios in
traffic affected areas. As a consequence, emissions within the urban area of Berlin were responsible for the
elevated particle levels between June and August 2014. Canister samples displayed the presence of remarkably
elevated AVOCs between 19 times and 50 times higher values than the corresponding values at the urban
background site. These observations are in good agreement with other studies (e.g. von der Weiden-Reinmüller
et al., 2014).
During the campaign period, a significant influence of vegetation on pollutant concentrations was observed too
with quite substantial amounts of isoprene and terpenes. Differences in effects were noted between three broadly
different types of vegetation: agricultural areas, urban parks and urban forests.  While agricultural areas showed
similar particle number and mass concentrations relative to the urban background, significantly reduced particle
concentrations (number and mass) were observed in both forests (-33%) and parks (-15%) indicating a reduced
production of and/or a substantial sink for particles.
This vegetation effect however was dependent on vegetation dimensions. Urban parks with a much smaller
extension than urban forested areas were shown to not have significantly lower but rather elevated NO or $NO_2$
concentrations than the urban background station in Neukölln (NO: >+45% and $NO_2$: >100%). This is in contrast
with both agricultural areas and urban forests, which both showed significantly lower mixing ratios of $NO_2$
compared with the urban background. Ox was substantially reduced in forests (-17.9±7.7%) compared to urban



residential areas (reference site) with a VOC sink strength only about 50% of the corresponding air at the urban
residential area in Neukölln (see sections 4.1.1 and 4.1.4). Ozone mixing ratios were significantly reduced in
forests, and higher in agricultural areas. Similar effects on ozone and $NO_2$ were not observed in urban park areas,
perhaps due to the smaller park size and since the measurements conducted on the street near-by. Bicycle based
measurements would be needed for an improved classification.
These road based observations are consistent with a $NO_x$-saturated chemical regime throughout the majority of
the urban campaign area. However, while the additional biogenic VOCs from forests acted as an additional sink
for ozone, this did not compensate the strength of the AVOC sink within the areas affected by urban transport
and within residential areas. As both $NO_2$ and Ox decreased substantially towards forests, a PSS cannot be
assumed for the entire area of Berlin, rather only for residential areas, industrial, commercial and transport
affected areas and parks. These results suggest that increased urban green spaces would be a viable method to
reduce particulate pollution if substantial in dimension, however not necessarily for ozone or $NO_2$. Reduction of
NO mixing ratios would require reduction in emissions from traffic, which would be expected to lead to an
increase in the mixing ratio of ozone. The intensity of this increase would be dependent on the biogenic feedback
processes involving the emission on BVOCs and the formation of secondary organic aerosol mass.
The new approach using bicycles in addition to van measurements for a detailed microscale investigation yielded
important additional information for areas not accessible for road-based mobile platforms such as vans and for
regions most relevant for pedestrians and cyclists. For instance particle number concentrations varied by orders
of magnitude when shifting from the centre of the road to the walkway and when approaching bus stops or traffic
lights from the pedestrian point of view. A further development of miniature observation instruments for other
pollutants such as nitrogen oxides, CO and black carbon would be highly recommended to address not only the
street centre but the area most relevant for the health of local citizens. The BAERLIN2014 campaign was
conducted for summer time conditions (June to August 2014) in a selected region representing various
environments present in of the whole overall area. Clearly planned annual observations of different urban
conditions (in different environments) by a multitude of cheap bicycle observation methods, including making
use of volunteers cyclists, would improve the basis for an observation based pollution map of the city. We
underline the importance of a resolution  improved and updated surface coverage map (compared to the current
CORINE land cover) with more surface information such as vegetation type, street or buildings for any
stratification approach based on surface-types too.
To explore the effects and sensitivities of different vegetated land cover types, we recommend investigating the
data further with detailed atmosphere-biosphere-chemistry-transport models and box-model simulations, which
can then be used to test mitigation scenarios.





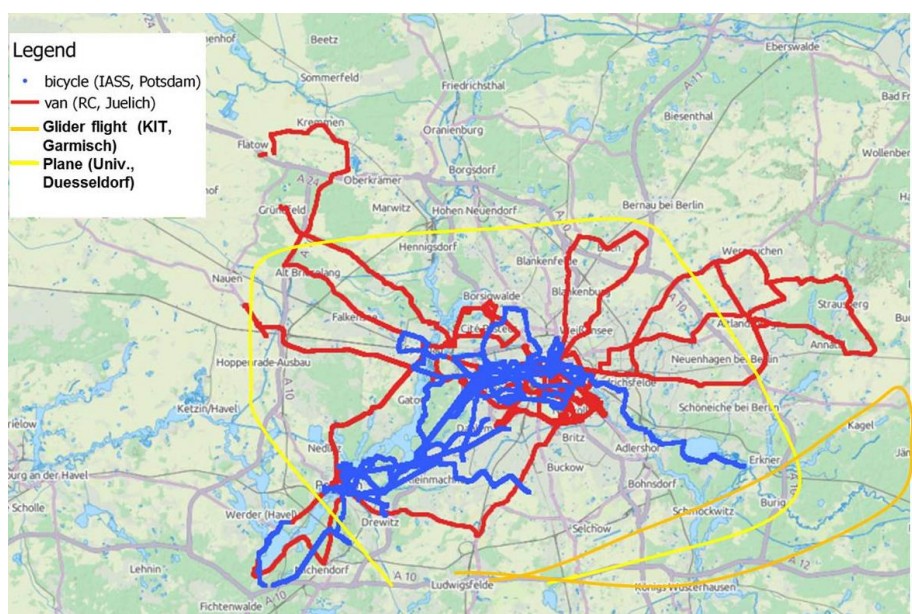

2   **Figure 1.** Mobile measurement routes in the BBMA area: bicycle routes in blue, van routes in red and airborne

3   tracks in yellow (air plane) and orange (glider). Berlin is located in the centre and Potsdam at the south western

4   concurrence of different bicycle and van tracks.





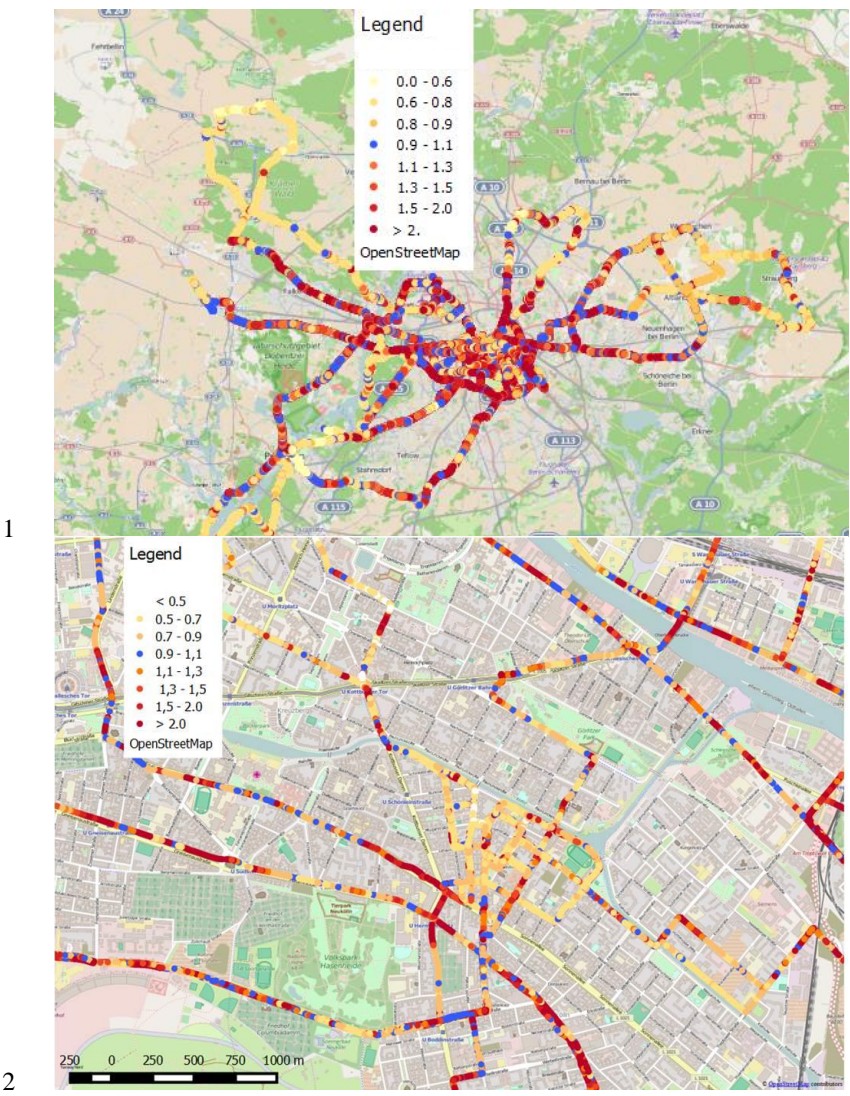

**Figure 2.** Relative values observed for carbon monoxide in the entire area of study (top) and for a zoom in
Neukölln (bottom). Colours indicate the horizontal heterogeneity and the deviation to the reference in Neukölln.
Blue indicates values ±10%.





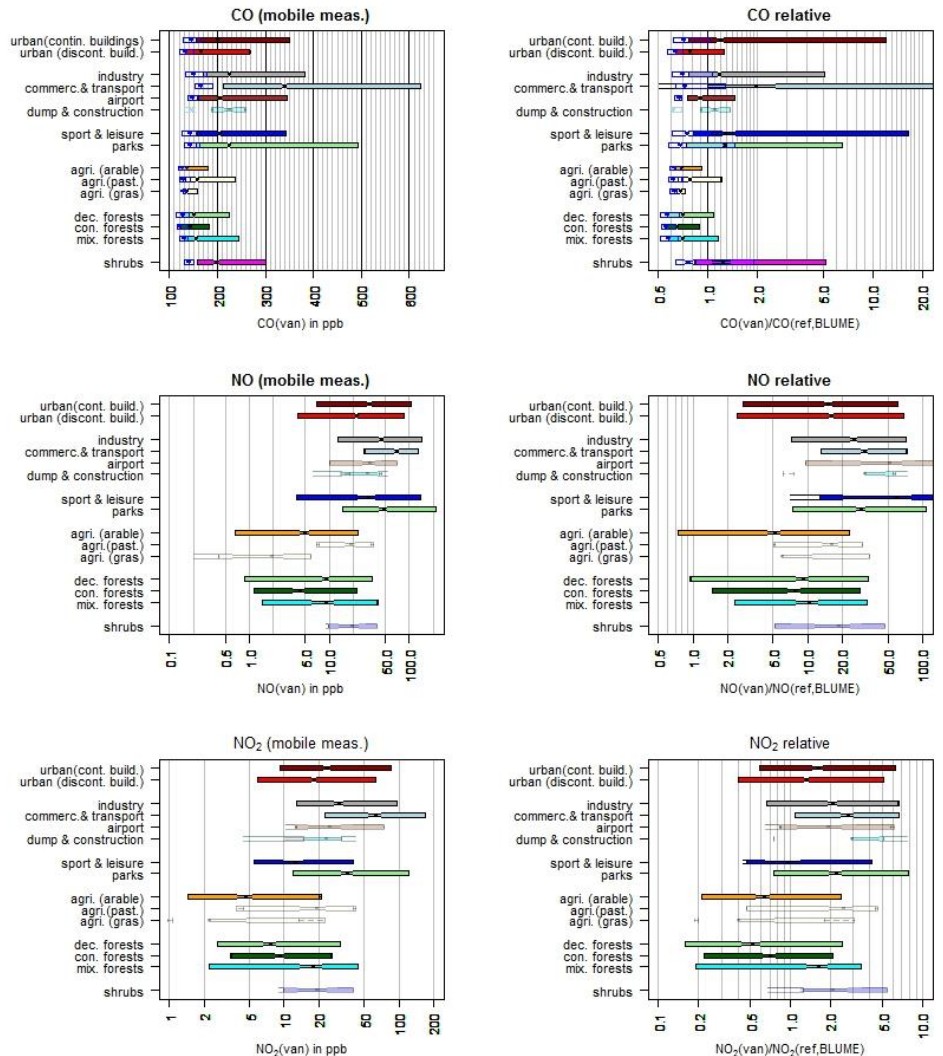

**Figure 3.** Boxplots of CO, NO and NO$_2$ mobile measurement values (upper graphs) and ratios of mobile measurements relative to Neukölln (lower graphs) in areas of different land use (CORINE). The boxplots start and end at the 25th and 75th percentile with a notch between the 45th and 55th percentiles. Blue surrounded transparent bars in the CO graphs refer to the so-called baseline values while the coloured bars represent all the observations. Shaded bars indicate an insufficient number of data points. Values and number of corresponding values are given in the supporting online information (SOI).



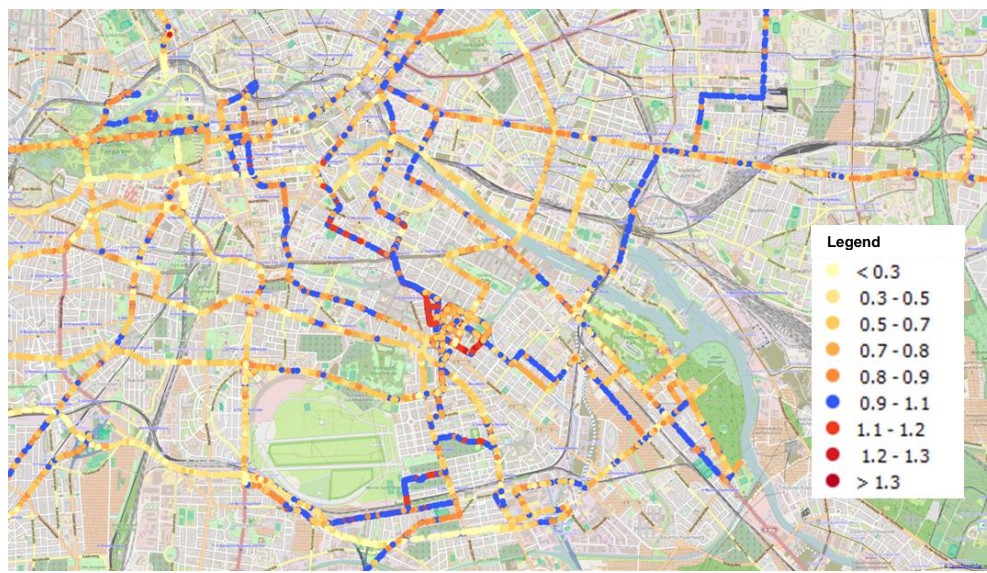

2  **Figure 4.** Horizontal variation of relative ozone mixing ratios, i.e. measured values relative to the ones at the

3  same time in Berlin-Neukölln. As before, blue colour indicates a 10% difference to the reference site.





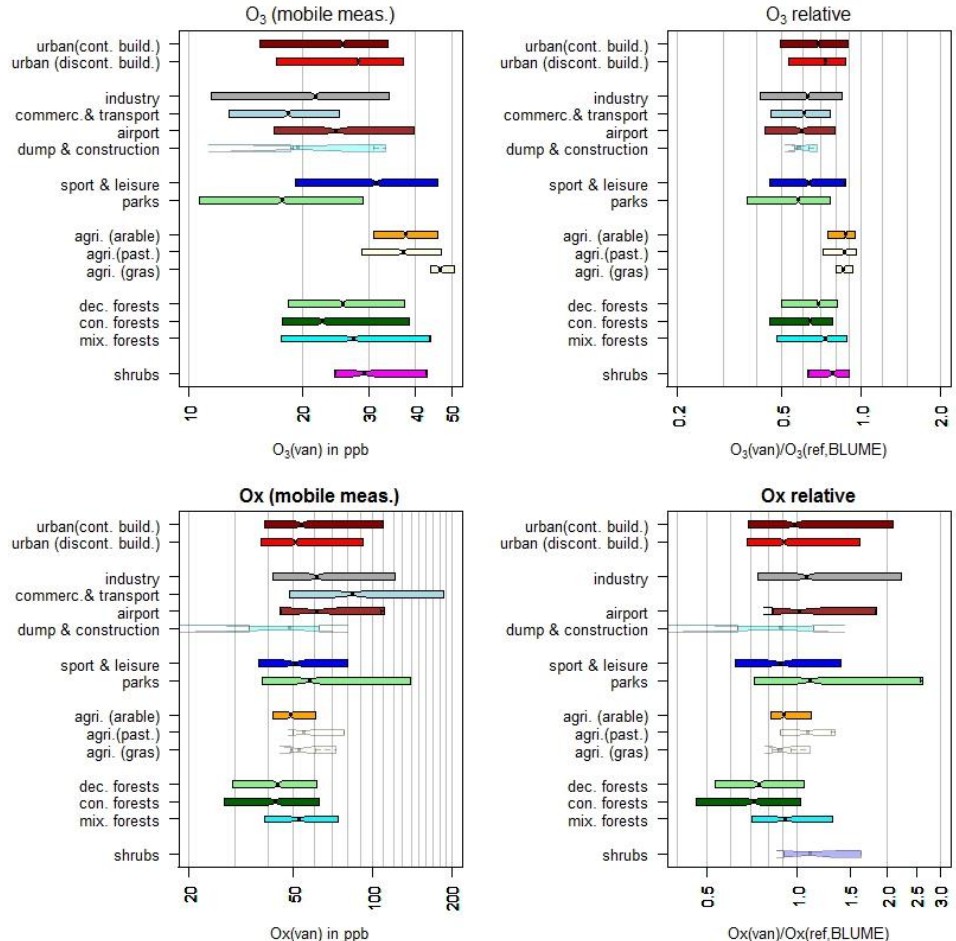

**Figure 5.** Boxplots of mobile measurements (left) and relative (right) values boxplots for ozone (top) and Ox
(bottom) with respect to different surface type usage based on CORINE. Boxplots range from the 25th to the 75th
percentile each with notches from the 45th to the 55th percentile centred round the median. Shaded bars indicate
an insufficient number of data points. Values and the corresponding numbers of available data are provided in
SOI.





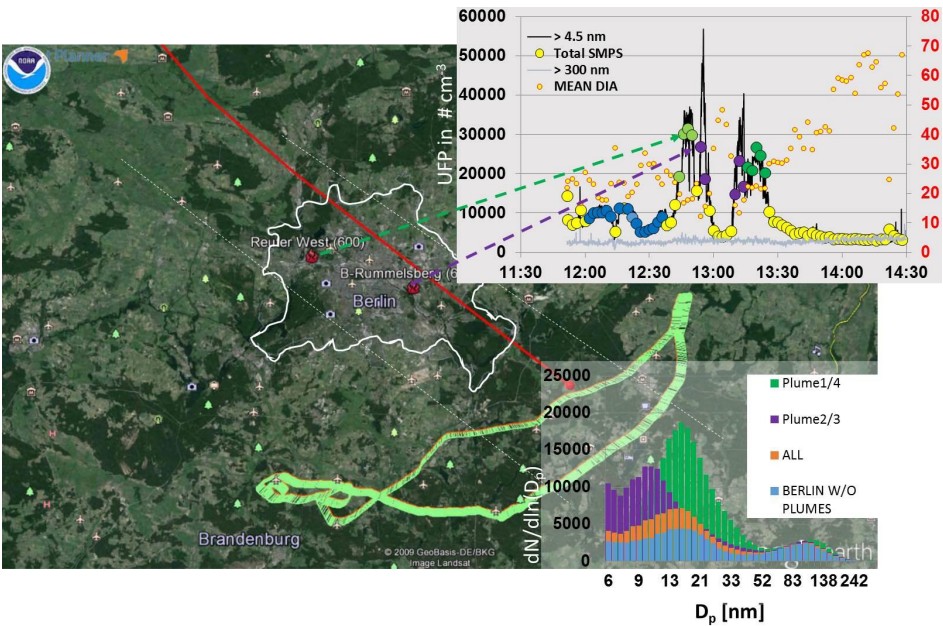

**Figure 6.** Particle number concentration (particle diameter $D_p > 4.5$nm) and mean particle diameters (right
vertical axis in the upper right plot) on the 12th of June 2014 during the glider measurements (W. Junkermann,
KIT, Garmisch-Partenkirchen). Maxima in UFP concentration and minima in mean particle diameter were found
in the pollution plumes of the two power plants located in Rummelsburg and in Reutter-West (Plume 2/3,
magenta size distribution).





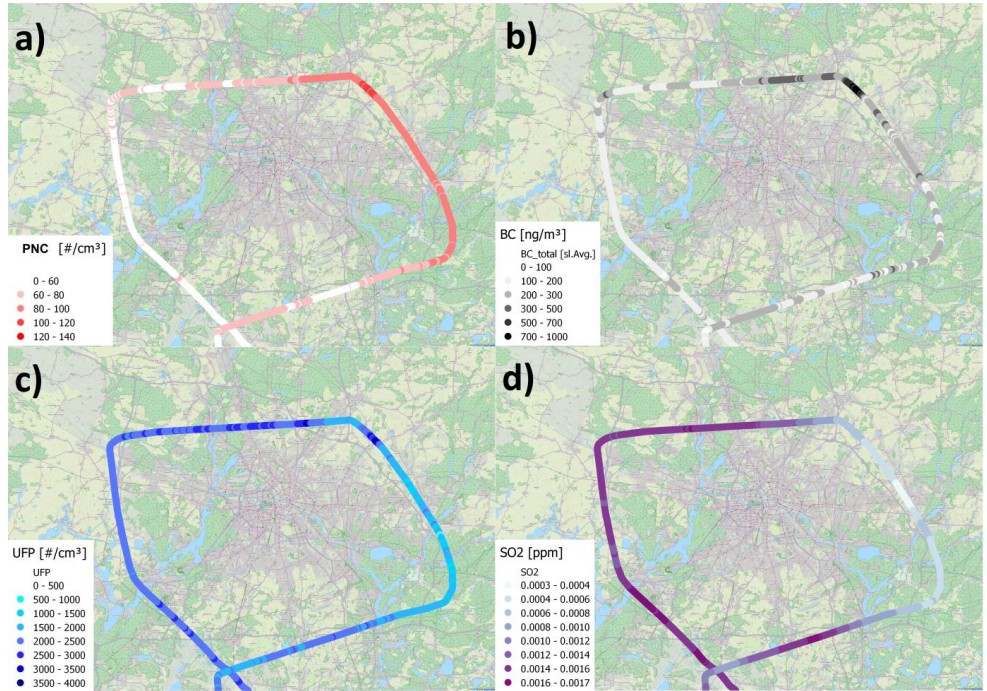

2 **Figure 7.** Spatial distribution of the air plane measurements on the 10th of October 2014: a) Coarse particle

3 number concentration, b) BC, c) ultrafine particle number concentration, d) sulphur dioxide.





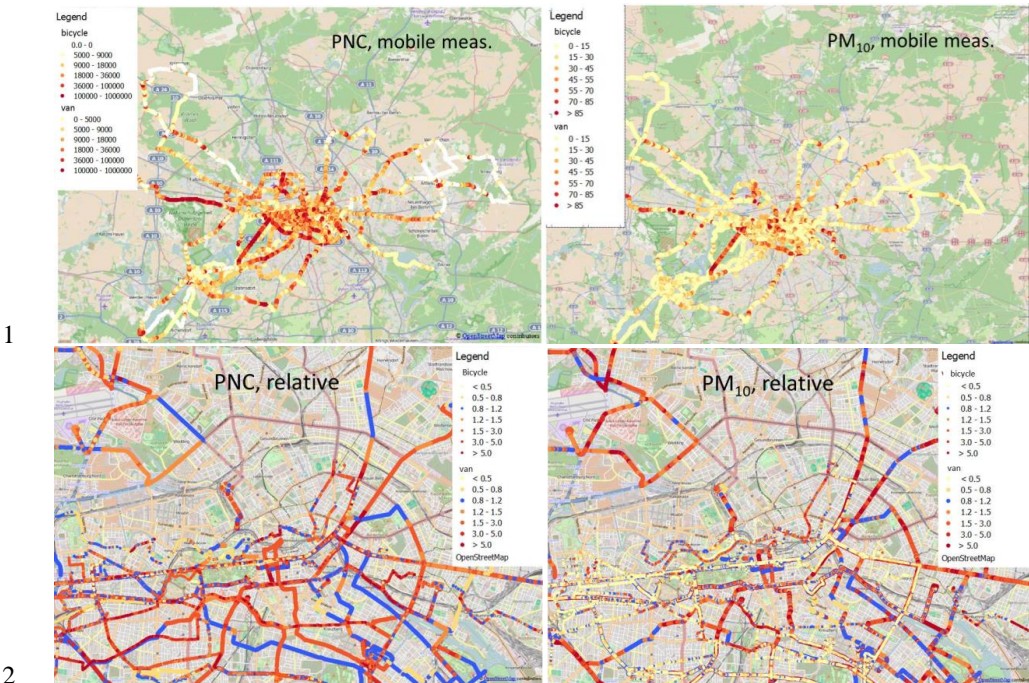

**Figure 8.** Heterogeneity of particle number (left) and mass ($PM_{10}$, right) concentrations in and around Berlin
detected by bicycle and van sensors. The upper line displays the total area and the bottom line provides the
relative values for number and $PM_{10}$ concentrations.



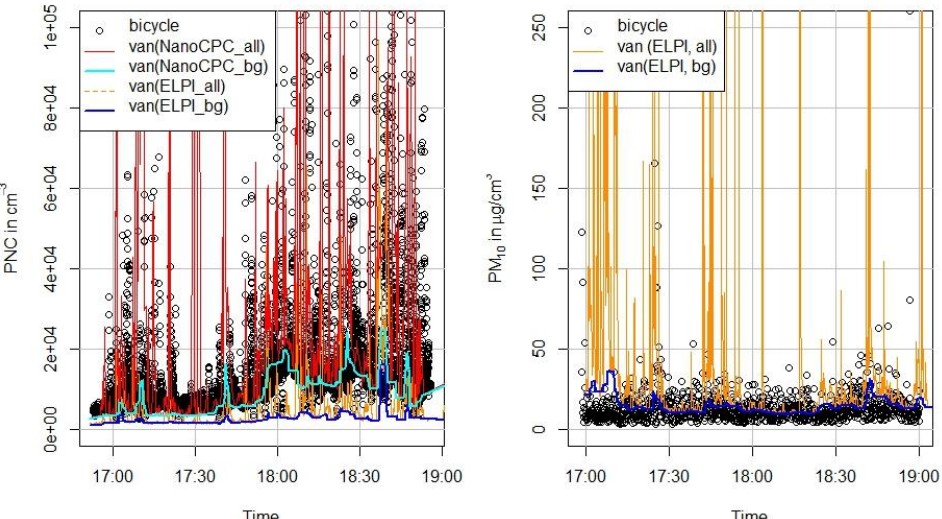

**Figure 9.** Comparison of bicycle and van based particle measurements: (left) total particle number concentration, (right) PM$_{10}$ mass, on the 4$^{th}$ of August 2014. Van measurements are shown by the two colored lines, with the red line representing all measurements and the blue line the calculated background concentrations (10$^{th}$ percentile of 3 min running mean). The time is provided in CEST.



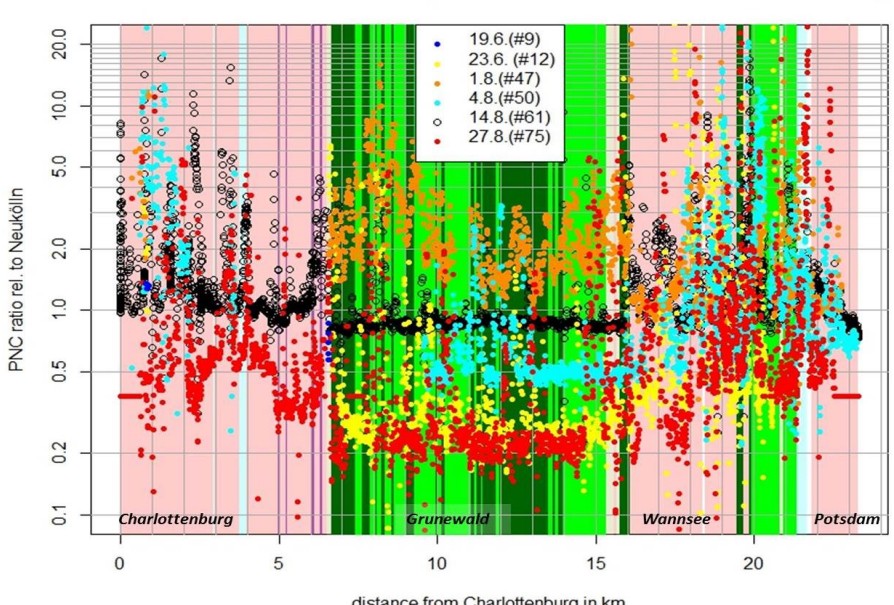

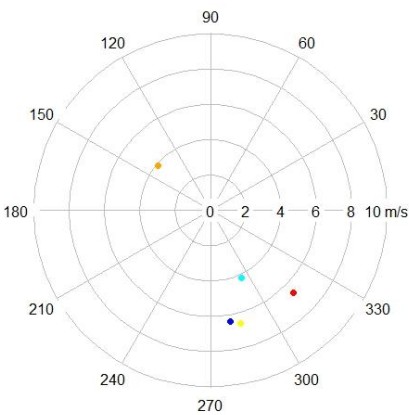

**Figure 10.** Top: Measurements by bicycle, following the same route, Berlin-Charlottenburg to Potsdam-IASS,
on different days and during different times. The dots plotted are particle number concentration ratios relative to
the stationary site in Neukölln with a time resolution of 10s. Green shaded areas are vegetated areas; pink shaded
areas are anthropogenically dominated areas. Bottom: Wind rose and speed at Tempelhof (DWD) measured for
the times of the individual tracks. The colour coding is identical with the one in the upper graph. Note, the
corresponding wind data for track 61 is not available.





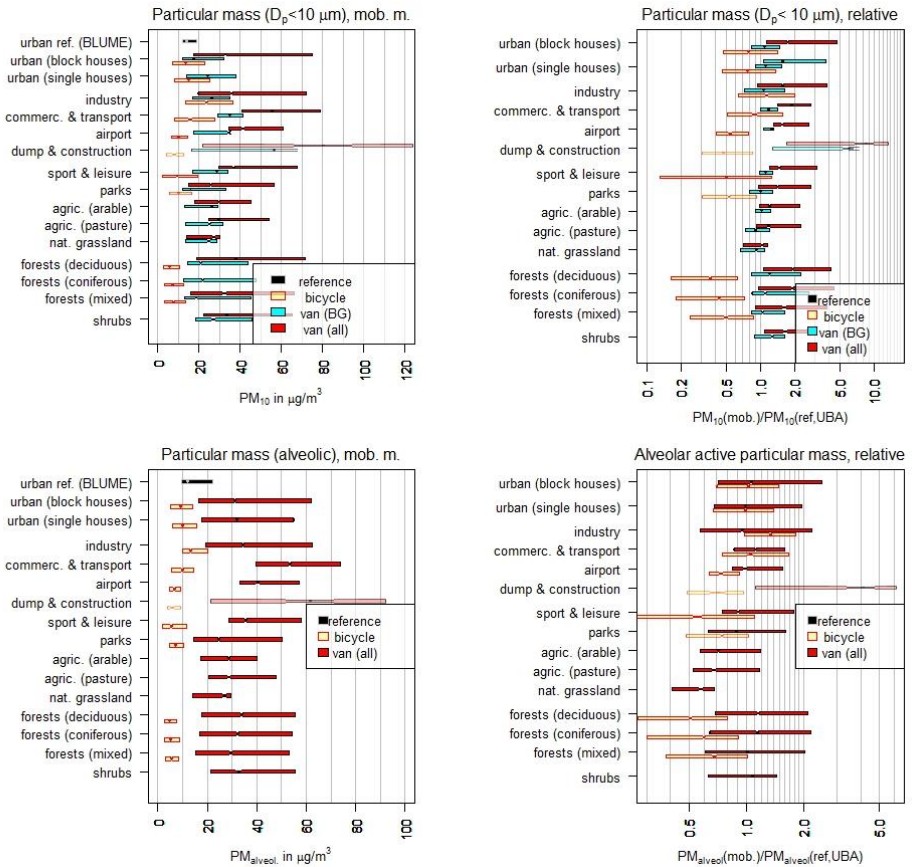

**Figure 11.** Particulate mass concentrations (left) and concentration ratios (right) for different land surface types and different observation platforms compared to the measurements in Berlin-Neukölln: top – $PM_{10}$, bottom – PM(alveolar).





1    **Appendix**

3    **A1. Information on instrument and methods applied**

4    **Table A1.** List of applied instruments, time resolutions, uncertainties and detection limits/ranges, respectively.

| Meas. platform (resp. inst.) | Parameter | Instrument | Time resolution | Uncertainty | Detection limit or range |
|---|---|---|---|---|---|
| Bicycle (IASS) | PNC, mod. mean size, LDSA | DiSCmini, Matter Aerosol (Wohlen, CH) | 1 s | 15% (500cm$^{-3}$), 30%, 15% | 10$^3$-10$^6$ cm$^{-3}$ μm$^2$/cm$^3$ (D$_p$: 10-500nm) |
| Bicycle (IASS) | PM$_{10}$, PM$_{2.5}$, PM$_1$, PM$_{health}$, PSD, MSD | Model 1.108, GRIMM (Ainring, D) | 12 s | 3% 3% | 0.1-10$^5$μg/m$^3$ 1-2x10$^6$ cm$^{-3}$ (D$_p$: 0.3-20 μm) |
| Bicycle (IASS) | Temperature & RH | Model 1.154, GRIMM (Ainring, D) | 12 s | ±0.1°C/ 1% rH | 0-80°C/ 10-95% |
| Van (RC Jülich) | particle number | ELPI, Dekati (Kangasala, FI) | 1s | 10% | 0.1-10$^7$ cm$^{-3}$ (f(size)) (D$_p$:0.007–10μm) |
| Van (RC Jülich) | particle number | NanoCPC 3788, TSI (Aachen, D) | 1s | 10% | 0-4x10$^5$ cm$^{-3}$ (D$_p$: 0.003-3μm) |
| Van (RC Jülich) | NO, NO$_2$, O$_3$ | CLD 770, Chemilumines-cence | 5 s | 5% (NO&NO$_2$) 10% (O$_3$) | 40ppt(NO&O$_3$) 80 ppt (NO$_2$) |
| Van (RC Jülich) | CO | UV-Resonance-Fluorescence | 1s | 1.3 ppb$_v$ | 1 ppb$_v$ |
| Van (RC Jülich) | CO$_2$, CH$_4$ | Cavity-ringdown Spectrometer | 0.1 s | ≤ 200ppb$_v$ (CO$_2$) ≤ 3 ppb$_v$ (CH$_4$) | ≈200ppb$_v$ (CO$_2$) ≈ 3ppb$_v$ (CH$_4$) |
| Van (RC Jülich) | temperature & RH | HMT 330, Vaisala (Helsinki, FI) | 1 s | 0.2°C 1% rH | -60 - +160°C 0-100% |
| Van (RC Jülich) | wind-direction & -speed | WMT 50 Vaisala (Helsinki, FI) | 1 s | 5% | 0-60 m/s |
| Van (RC Jülich) | Position | WBT202, Wintec | 1 s | ±5 m | - |





| | | (Milpitas, USA) | | | | |
|---|---|---|---|---|---|---|
| ultralight (KIT) | T, dew point | TP3-S, Meteolabor (Baiersdorf, D) | 1 s | ±0.25K | ±0.25K | -30 - +50°C  -80 - +60°C |
| ultralight (KIT) | $N_{total}$ ($D_p$ > 4.5 nm) | 5.410 SKY OPC, GRIMM (Ainring, D) | 1 s | 10% | | 0.1-$10^7$ cm$^{-3}$ |
| ultralight (KIT) | PSD: low. sizes, 4.5-350nm upp. sizes, 0.3-20μm, PM | SMPS 5.403, GRIMM (Ainring, D) OPC 1.108, GRIMM (Ainring, D) | 2 min | 3-15% (f(size)) | 3% | 0.1-$10^7$ cm$^{-3}$ 0.1-$10^5$μg/m$^3$ |
| ultralight (KIT) | Soot/BC | AE33 AVIO, AEROSOL d.o.o., (Ljubljana, SLO) | 1 min | 10% | | 0.03–100 μg/m$^3$ (1 min), 5 LPM. |
| DA42 (HSD) | T, rH | Voltcraft, DL-121 TH | 2s | 1°C, 3%rH | | -40-(+70) °C |
| DA42 (HSD) | UF-$N_{total}$ ($D_p$: 25-300 nm) | NanoCheck 1320, GRIMM (Ainring, D) | 10 s | 30% | | $5×10^2$-$5×10^5$ cm$^{-3}$ |
| DA42 (HSD) | PSD (0.25 – 32 μm), PM | 1.109, Grimm (Ainring, D) | 6 s | 3% | | 1-$10^6$ cm$^{-3}$ |
| DA42 (HSD) | Soot/BC | AE 33 Avio, Magee, Ljubliana, SLO | 1 s | 10% | | 0.03–100 μg/m$^3$ (1 min), 5 LPM. |
| DA42 (HSD) | $SO_2$ | APSA-370, Horiba | 15 s | 1% | | 0-10pm |



**A2. Additional information with respect to the bicycle measurements**
As stated in section 3.1 of the study, both particle instruments, i.e. the GRIMM1.108 and the DiSCmini, were
located in a backpack or a pannier which sampled ambient air by conductive inlet tubes. These inlet tubes (black
silicone for the GRIMM, Tygon for DiSCmini, both ca. 50 cm in length) and the temperature sensor were fixed
on the outside of backpack or pannier. Losses from inlets and tubing were accounted for with correction factors
provided in Table A2.
Based on the particle measurements of the GRIMM instrument its software calculated six particulate mass values
corresponding to different size ranges and corresponding to potential health effects: $PM_{10}$, $PM_{2.5}$ and $PM_1$ as well
as PM(inhalable), PM(thoracic) and PM(alveolar). The final three health-related quantities estimate the particle
number concentration for those size fractions making it to the throat/upper respiratory system, lung, and blood
system, respectively (EN 481; European Committee for Standardization, 1993).
All particle instruments except the instrument were calibrated a month prior to the campaign in a controlled
comparison experiment at TROPOS in Leipzig. Both instruments used on the bicycle measurement platform -
DiSCmini and GRIMM 1.108 – were repeatedly operated in parallel with the suite of calibrated particle
instruments (GRIMM 1.108, 5.403 and 5.416, and a TSI NSAM provided by the Federal Environmental Agency,
Berlin) set up at the reference site in Neukölln. This was used for both instruments to obtain the calibration
factors including the inlet losses listed in Table A2.
**Table A2.** Correction factors and mean losses for different parameters of both bicycle instruments.

| Instrument and parameter | Correction factor f | Mean loss |
|---|---|---|
| DiSCmini, tot. part. num. conc. | 1.22±0.20 | 18.8±3.1% |
| DiSCmini, lung depos. surface area | 1.15±0.13 | 13.0±9.0% |
| Grimm 1.108, PM10 | 1.24±0.46 | 19.3±7.1% |
| Grimm 1.108, PM2.5 | 1.24±0.29 | 19.6±4.5% |
| Grimm 1.108, PM1 | 1.29±0.12 | 22.6±2.1% |
| Grimm 1.108, PM(inhalable) | 1.28±0.64 | 21.7±10.8% |
| Grimm 1.108, PM(thoracic) | 1.25±0.47 | 19.7±7.5% |
| Grimm 1.108, PM(alveolic) | 1.21±0.28 | 17.5±4.0% |



**B. Additional gas-phase related results**

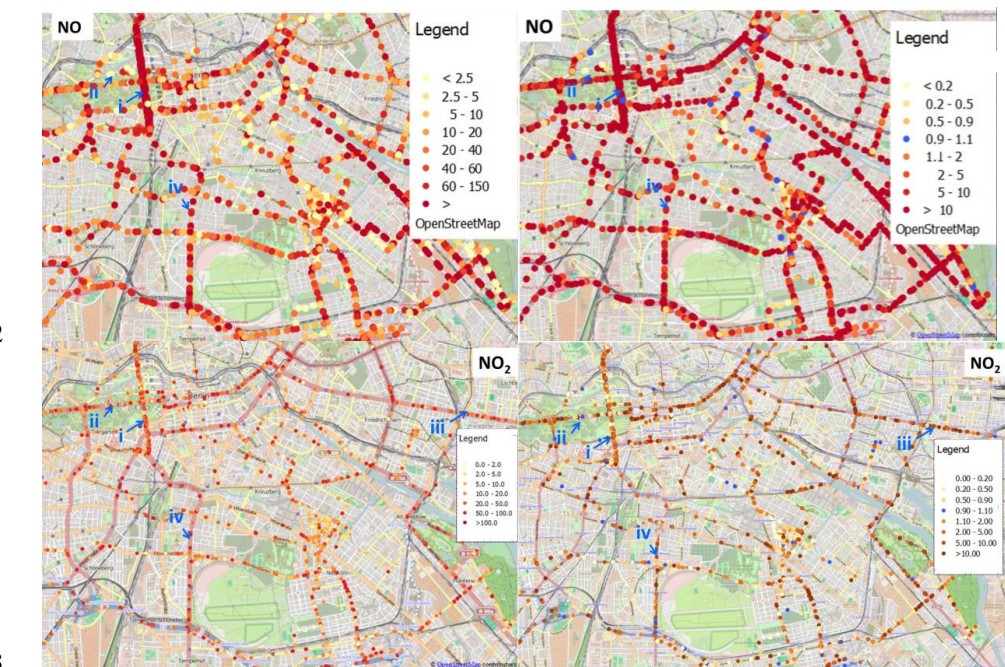

**Figure B1.** Mobile measurement data (left) and relative (right) graphs of nitrogen monoxide (NO, top) and
dioxide (NO$_2$, bottom) observed by the measurement van. Mobile measured values are displayed in ppbv.
Colours indicate the heterogeneity of the parameters, of the range (mobile measured values) and the variation
with respect to reference value.





**C. Additional aerosol particle related results**
**Table C1.** Particle mass ($PM_{10}$) burden characteristics (bicycle/van background (van all) meas.) at different land
use types in $\mu g/m^3$. "-" indicates areas, which have not been tested by the method. This table provides the 25th,
50th and 75th percentiles as well as the mean and the number of available data points.

| surface type | 25th | median | 75th | mean | no. of data |
|---|---|---|---|---|---|
| Urban - block build. | 6.9/12.2(17.4) | 13.6/17.2(32.8) | 22.7/31.8(74.7) | 24.3/25.7(75.6) | 8260/21801 |
| Urban -single build. | 7.9/14.2(18.6) | 15.0/24.3(34.3) | 25.2/38.0(69.4) | 29.0/28.7(67.9) | 19143/82502 |
| Industry | 13.6/16.8(19.6) | 23.9/26.1(35.9) | 36.5/34.9(72.2) | 30.7/28.2(73.9) | 1464/14047 |
| Com.+transp. | 8.1/29.3(40.8) | 15.9/34.8(55.5) | 27.8/41.2(78.7) | 23.2/35.8(77.2) | 341/4875 |
| Airport | 6.6/17.5(34.4) | 9.9/34.4(41.3) | 14.7/35.2(60.8) | 11.3/29.3(130.2) | 137/738 |
| Parks | 5.5/11.9(14.8) | 10.0/15.7(25.7) | 16.3/31.8(56.6) | 15.2/25.7(71.9) | 2364/9598 |
| Leisure area | 2.3/17.1(29.5) | 9.4/28.8(36.7) | 19.5/33.8(67.7) | 30.8/29.0(77.0) | 623/3378 |
| Arable land | -/13.0(18.1) | -/26.2(29.5) | -/29.0(45.3) | -/23.9(46.8) | -/9488 |
| Pasture | -/13.3(24.7) | -/25.1(29.7) | -/31.5(53.9) | -/25.7(68.2) | -/938 |
| Nat. grassl. | -/13.4(14.2) | -/24.7(27.4) | -/28.5(30.2) | -/21.1(27.0) | -/362 |
| Dec. forest | 2.8/14.4(19.1) | 5.9/21.0(38.0) | 10.4/43.7(71.4) | 8.9/29.1(58.2) | 2096/8874 |
| Con. forest | 3.2/12.4(17.8) | 7.1/21.9(38.3) | 12.6/47.7(70.9) | 12.7/30.3(52.7) | 4141/7078 |
| mix. forest | 3.4/13.1(15.8) | 7.8/18.7(32.7) | 13.5/45.0(65.9) | 13.8/27.2(53.6) | 694/1820 |





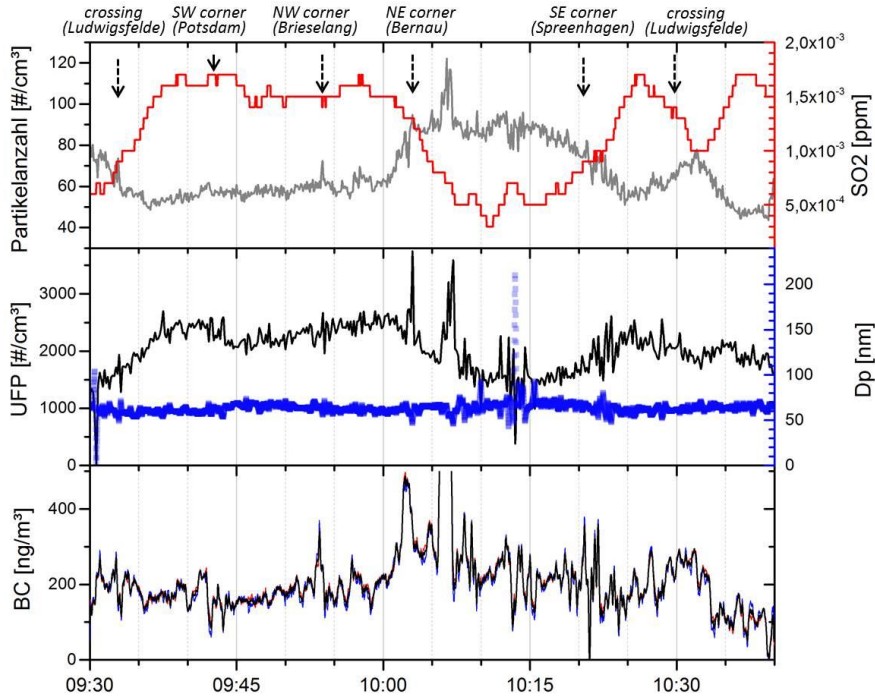

2 **Figure C1.** Particle parameters measured during the Oct. 10 flight around Berlin. Particle number concentration

3 and sulphur dioxide ($SO_2$), UFP and UFP-diameter, Black Carbon (from top position towards bottom) as

4 measured at a constant altitude of around 500 m (1700 ft).





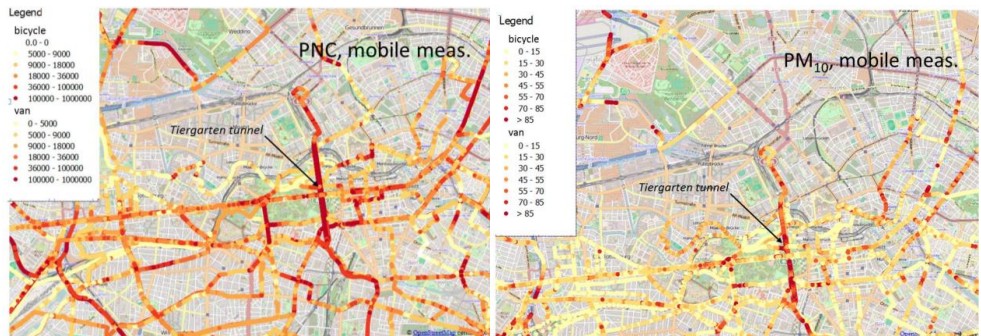

2 **Figure C2.** Zoomed heterogeneity of particle number (left) and mass (PM$_{10}$, right) concentrations in the center

3 of Berlin displayed in absolute measured values. This figure is an extension of Fig. 8.



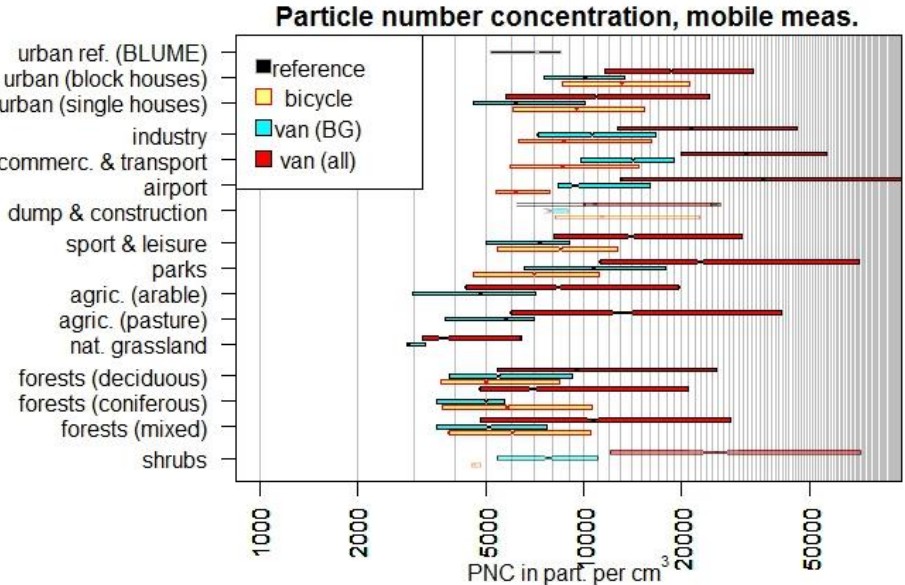

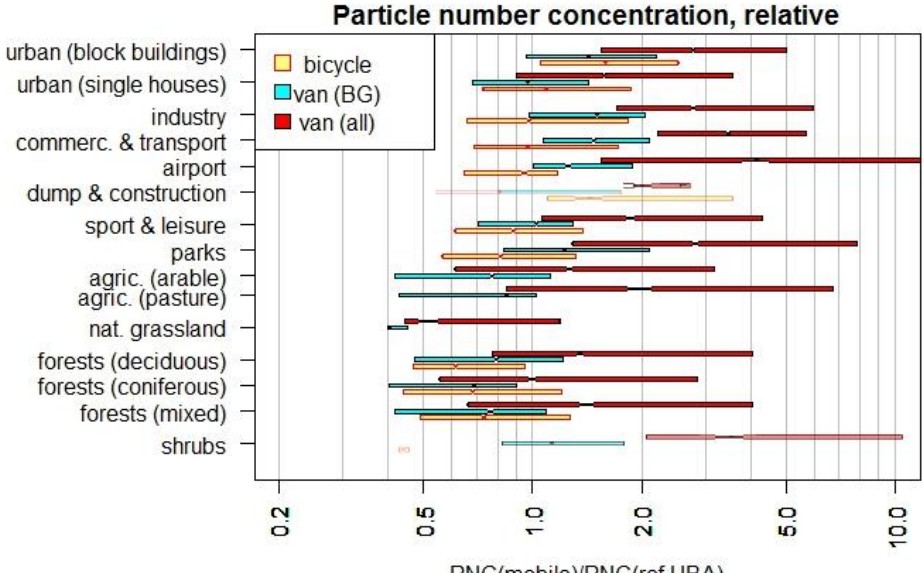

2 **Figure C3.** Boxplots of particle number concentration ratios for different land surface types (CORINE) and

3 different observation platforms compared to the measurements in Berlin-Neukölln. The boxplots range from the

4 25th to the 75th percentile with notches from the 45th to the 55th percentile centered on the median.





**D.  Information about further results – temperature**

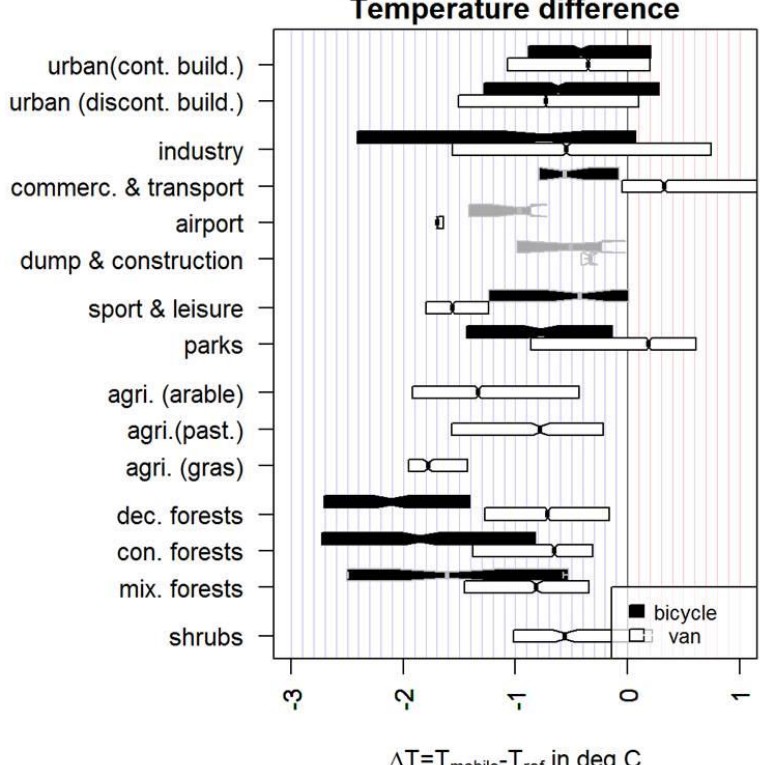

**Figure D1.** Boxplot of temperature differences for different land use types and different observation platforms
compared to the measurements in Berlin-Neukölln. Displayed is the range between the 25th and the 75th
percentile with a notch from 45th to 55th percentile. Grey borderlines of bars represent categories with
insufficient data values, missing bars no data by the corresponding method.





**Supporting online information:**
Further tables and graphs on frequency distributions of gases and particle properties are available in supporting
online information.
**Acknowledgements**
The authors thank all the cyclists at the institute for their high motivation and enthusiasm even during poor
weather conditions. Thanks go to Alfred Wiedensohler, Wolfram Birmili, Kay Weinhold and colleagues for
calibration of the particle instruments and further support regarding the measurements. Numerous colleagues at
the IASS in Potsdam provided various types of support. Thank you all. Without any of those the present study
would not have been possible. The same gratitude applies to the colleagues at the Leibnitz institute for
tropospheric research in Leipzig (Germany) for continuing support and discussion.

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





1   **Table 1.** European Union (EU) and U.S. (EPA) legislation on selected pollutant concentrations. *valid from 1st

2   of January 2015 onward.

| Pollutant | EU | | EPA | |
|---|---|---|---|---|
| | *daily* | *annual* | *daily* | *annual* |
| Ozone (EU: target value EPA: limit value) | 8h-mean: $\leq$ 120 $\mu g/m^3$ ($\approx$60 $ppb_v$) not to be exceeded more than 18 times a year | - | 8h-mean: 75 $ppb_v$ | - |
| Nitrogen oxides ($NO_2$) | 1h-mean: 200 $\mu g/m^3$ ($\approx$100 $ppb_v$) not to be exceeded more than 18 times a year | Mean: 40 $\mu g/m^3$ ($\approx$20 $ppb_v$) | 1h-mean: 100 $ppb_v$ | 53 $ppb_v$ |
| Benzene, toluene, xylenes (BTX) compounds | 1h-mean: 5 $\mu g/m^3$ ($\approx$1.9 $ppb_v$) | - | - | - |
| Particulate matter (PM) | 24h-mean: $PM_{10} \leq 50$ $\mu g/m^3$ not to be exceeded more than 35 times a year | Mean: $PM_{10} \leq 40$ $\mu g/m^3$ $PM_{2.5} \leq 25$ $\mu g/m^{3*}$ | 24h-mean: $PM_{100} \leq 150$ $\mu g/m^3$ $PM_{2.5} \leq 35$ $\mu g/m^3$ | Mean of 3 years: $PM_{2.5,prim.} \leq 12$ $\mu g/m^3$ $PM_{2.5,sec.} \leq 15$ $\mu g/m^3$ |
| Carbon monoxide (CO) | 8h-mean: 10 $mg/m^3$ ($\approx$10.3 $ppm_v$) | - | 8h-mean: 9 $ppm_v$ 1h-mean: 35 $ppm_v$ both not to be exceeded more than once a year | - |



1   **Table 2.** Contribution of different surface types to the total surface area of Berlin.

| Surface type | Area covered [ha] | Fraction of total [%] |
|---|---|---|
| Built-up areas, streets (19%) | 49 975 | 56.1 |
| Green areas: | 29 275 | 32.8 |
| *Forests* | *16 349* | *18.3* |
| *Public green areas* | *12 926* | *14.5* |
| Agricultural areas | 3 953 | 4.4 |
| Lakes, rivers | 5 953 | 6.7 |
| *Total* | *89 157* | *100* |



1    **Table 3.** List of applied mobile measurement platforms, parameters quantified and time scales.

| Mobile meas. platform | Parameters measured | Time scale |
|---|---|---|
| bicycle | T, particle number concentration, $PM_{10}$, $PM_{2.5}$, $PM_1$, PM(health), LDSA | June, $10^{th}$ – September, $5^{th}$ |
| van | T, rH, $O_3$, NO, $NO_2$, CO, $CO_2$, $CH_4$, particle number conc., particle surface area, PM and canister samples (VOCs) | July, $31^{th}$ – August, $6^{th}$ |
| ultralight aircraft | T, dew point, $O_3$, particle number conc., particle size distribution | June, $12^{th}$ |
| Air plane (DA 42) | T, dew point, $SO_2$, particle number, particle size distribution, soot | October, $10^{th}$ |



1  **Table 4.** Land use types based on the CORINE classification. For number of measurement values (n) for each

2  surface type for each instrument/parameter, see the SI.

| No. | Surface type name | Surface character type |
|---|---|---|
| 1 | Urban (contin. build.) | Residential areas, block houses with several floors |
| 2 | Urban (discount. build.) | Residential areas, single houses, less dense setting |
| 3 | Industry | Industrial area |
| 4 | Commercial and transport | Commercial areas, streets, railways, motorway |
| 5 | Airport | Runways, airport related areas |
| 6 | Dump & construction | Dump and construction sites (2006) |
| 7 | Sport & leisure facilities | Vegetated areas linked with sporting facilities |
| 8 | Parks | Parks |
| 9 | Agriculture (arable land) | Arable land, used for food production |
| 10 | Agriculture (pasture) | Areas for livestock feeding |
| 11 | Agriculture (nat. grassland) | Areas with natural grassland |
| 12 | Forests (deciduous) | Deciduous forests |
| 13 | Forests (coniferous) | Coniferous forests |
| 14 | Forests (mixed) | Mixed forests |
| 15 | Shrubs | Areas with sparse smaller plants, bushes etc. |



**Table 5.** Canister samples analysed for VOC compositions. An ozone scrubber was applied in front of the inlet
to prevent sampling losses and artefacts. All values are provided as mean volume mixing ratios in pptv. The
different environments are grouped and the number of available samples is provided for each case. The third
column represents urban background measurement conditions at Nansenstraße is considered (urban background
standard). Elevated anthropogenic compounds with respect to vegetated background area concentration
(>average+2STD of the two smaller mixing ratios of vegetated areas) are marked in bold. Underlined numbers
mark biogenic compounds exceeding the average of the two smaller mixing ratios for anthropogenic dominated
areas + 2 standard deviations. Marked numbers represent the compounds substantially affecting the area with no
predominant emission of those. "b.d." abbreviates *below detection limit*.

| Compound | *Locations dominated by engine related emissions* | | *Urban background* | *Locations dominated by biogenic emissions* | | |
|---|---|---|---|---|---|---|
| | *Motorway, traffic jam* | *Tiergarten tunnel* | *Nansen-straße* | *Pfauen-insel* | *Treptower Park* | *Grunewald* |
| | *2 samples* | *10 samples* | *14 samples* | *1 sample* | *11 samples* | *1 sample* |
| Ethene | **16973±1262** | **5113±1257** | **465±263** | 197±39 | 247±96 | **442±88** |
| Ethyne | **4981±627** | **2023±985** | 286±239 | 103±21 | 236±55 | 331±66 |
| Ethane | **3585±1018** | **1655±366** | **1686±1514** | 866±173 | **2978±1473** | 771±154 |
| Propene | **5119±758** | **1588±448** | 251±64 | 187±37 | 228±55 | 256±51 |
| Propane | **4723±3622** | **1533±779** | **825±613** | 504±101 | **1007±476** | 257±51 |
| Propyne | **681±38** | **351±182** | 73±28 | b.d. | 66±19 | b.d. |
| Acetaldehyde | **3067±2355** | **591±181** | **336±139** | 91±18 | **382±112** | b.d. |
| 2-methylpropane | **2666±1878** | **660±542** | **504±441** | 70±14 | **255±134** | 77±15 |
| Methanol | **7275±4012** | **6631±2646** | **4996±3082** | 4192±838 | 2608±612 | 2564±513 |
| 1-butene/ i-butene | **2482±304** | **740±297** | **300±412** | 100±20 | 111±21 | **156±31** |
| 1,3-butadiene | **731±73** | **249±109** | 43±11 | b.d. | 26±15 | b.d. |
| n-butane | **6140±3760** | **1626±938** | b.d. | 555±111 | 623±676 | 220±44 |
| trans-2-butene | **814±314** | **123±30** | 16±3 | **61±12** | 25±10 | 10±2 |
| cis-2-butene | **784±301** | **130±39** | **74±38** | 24±5 | 21±12 | **81±16** |
| 1,2-butadiene | **181±181** | b.d. | **33±7** | b.d. | b.d. | b.d. |
| Ethanol | **17622±8707** | **10462±7825** | 333±189 | 229±46 | 312±93 | 113±23 |



| | | | | | | |
|---|---|---|---|---|---|---|
| 3-methyl-1-butene | **224±112** | **99±37** | 52±6 | b.d. | 16±33 | b.d. |
| 2-methylbutane | **30906±10821** | **3913±1668** | 465±178 | b.d. | 306±90 | **656±131** |
| Acetone | **12328±7453** | **6827±5420** | **10721± 24004** | **37040±7408** | 3798±1856 | 2703±541 |
| 1-pentene | **605±220** | **86±39** | 35±8 | b.d. | 29±16 | 26±5 |
| 2-propanol | **612±612** | **420±357** | 44±14 | b.d. | 42±17 | **81±16** |
| 2-methyl-1-butene | **1014±173** | **71±108** | b.d. | b.d. | b.d. | b.d. |
| n-pentane | **7886±2785** | **1121±521** | 242±106 | 57±11 | 165±52 | 241±48 |
| Isoprene | b.d. | 157±93 | 266±159 | 1414±283 | 1320±363 | 776±155 |
| trans-2-pentene | **1421±173** | **214±91** | **28±13** | b.d. | b.d. | 14±3 |
| cis-2-pentene | **959±270** | **161±50** | 22±9 | 15±3 | b.d. | 11±2 |
| Propanal | **1251±1251** | **737±1120** | 54±24 | b.d. | 58±79 | 76±15 |
| 2-methyl-2-butene | **40±40** | **36±66** | 11±8 | b.d. | b.d. | b.d. |
| Acetic acid methylic ester | b.d. | b.d. | b.d. | b.d. | b.d. | b.d. |
| 1,3-pentadiene | b.d. | **47±117** | 14±4 | b.d. | b.d. | b.d. |
| Cyclopentadiene | b.d. | b.d. | 35±14 | b.d. | 45±22 | b.d. |
| 2,2-dimethylbutane | **6385±1992** | **875±364** | 117±111 | 67±13 | 112±110 | **175±35** |
| 2-butanol | b.d. | **3103±8097** | **117±156** | b.d. | 59±23 | 102±20 |
| 1-propanol | **502±502** | **418±259** | **342±377** | 94±19 | b.d. | b.d. |
| Cyclopentene | **335±335** | **27±75** | **39±11** | b.d. | b.d. | b.d. |
| Methacrolein | b.d. | b.d. | 80±37 | 287±57 | 147±49 | 200±40 |
| Cyclopentane / 2,3-dimethylbutane | **2646±792** | **6075±15604** | **275±316** | **277±55** | 88±27 | 139±28 |
| 2-methylpentane | **4772±2172** | **1274±500** | 232±112 | 45±9 | 160±100 | **291±58** |
| Methylvinylketone | b.d. | b.d. | 102± | 389±78 | 171±38 | 194±39 |
| Butanal | **1319±877** | 253±190 | 133±56 | b.d. | 126±99 | b.d. |
| 1-hexene | **47±47** | 20±58 | 113±68 | 129±26 | 40±54 | 38±8 |
| 3-methylpentane | **2259±557** | **572±250** | **73±40** | 42±8 | 54±19 | **123±25** |





| | | | | | |
|---|---|---|---|---|---|
| 2-methyl-1-pentene | **243±85** | **54±55** | 14±3 | b.d. | b.d. | b.d. |
| n-hexane | **1848±516** | **484±204** | **127±99** | 80±16 | 95±58 | 60±12 |
| trans-2-hexene | **190±46** | **59±21** | **110±53** | 15±3 | **26±14** | 11±2 |
| cis-2-hexene | **111±38** | **65±41** | **107±21** | 11±2 | b.d. | b.d. |
| 1,3-hexadiene (trans) | **85±85** | 27±52 | **53±10** | 34±7 | b.d. | b.d. |
| Methylcyclopentane | b.d. | **36±103** | **49±13** | 22±4 | b.d. | b.d. |
| 2,4-dimethylpentane | **1490±410** | **361±180** | 54±28 | 14±3 | 43±14 | **111±22** |
| Methylcyclopentene | **333±79** | **54±98** | 14±5 | b.d. | b.d. | b.d. |
| Benzene | **2281±796** | 1383±349 | 303±238 | 155±31 | 199±35 | 224±45 |
| 1-butanol | b.d. | **145±359** | 28±14 | b.d. | 39±19 | b.d. |
| Cyclohexane | **743±213** | **198±77** | 39±23 | 18±4 | 33±14 | **46±9** |
| 2-methylhexane | **708±132** | **256±144** | 36±14 | 23±5 | 34±24 | 35±7 |
| 2,3-dimethylpentane | **684±300** | **114±41** | 23±14 | **36±7** | 17±16 | 18±4 |
| 3-methylhexane | **894±138** | **268±84** | 82±34 | 54±11 | 109±33 | 110±22 |
| Pentanal | **102±14** | 12±22 | 11±2 | b.d. | b.d. | b.d. |
| Cyclohexene | b.d. | b.d. | 18±4 | b.d. | b.d. | b.d. |
| 1,3-dimethylcyclo-pentan (cis) | **287±2** | **74±40** | 11±5 | 11±2 | b.d. | 19±4 |
| 1-heptene | **138±42** | 25±31 | 17±10 | b.d. | b.d. | 13±3 |
| 2,2,4-trimethylpentane | **545±10** | **188±55** | 28±15 | b.d. | 24±10 | 34±7 |
| Heptane | **467±35** | **146±71** | 32±11 | 18±4 | 29±9 | 37±7 |
| 2,3-dimethyl-2-pentene | b.d. | 27±61 | b.d. | b.d. | b.d. | b.d. |
| Octene | 28±28 | b.d. | b.d. | b.d. | b.d. | b.d. |





| | | | | | | |
|---|---|---|---|---|---|---|
| Methylcyclohexane | **146±78** | **122±46** | 27±15 | b.d. | 18±15 | 14±3 |
| 2,3,4-trimethylpentane | **327±73** | **120±46** | 20±14 | 24±5 | 19±5 | 10±2 |
| Toluene | **8553±1675** | **2679±1012** | **407±237** | 299±60 | 276±133 | 212±42 |
| 2-methylheptane | **253±110** | **114±63** | 25±17 | b.d. | 17±12 | 10±2 |
| 4-methylheptane | **254±110** | **85±43** | 14±9 | b.d. | 11±10 | b.d. |
| 3-methylheptane | **121+67** | **82±45** | 17±13 | 68±14 | b.d. | 26±5 |
| Hexanal | 108±108 | **52±86** | **72±46** | b.d. | **129±69** | 12±2 |
| Acetic acid butylic ester | b.d. | b.d. | b.d. | b.d. | b.d. | b.d. |
| n-octane | **208±45** | **107±93** | 28±23 | 23±5 | 24±11 | **34±7** |
| Dimethylcyclo-hexane isomer | b.d. | b.d. | b.d. | b.d. | b.d. | b.d. |
| Ethylbenzene | **1285±200** | **485±207** | 76±40 | 21±4 | 55±31 | **127±25** |
| m/p-xylene | **3301±568** | **1853±2411** | 151±97 | 31±6 | 109±68 | **263±53** |
| Heptanal | b,d. | b.d. | 22±14 | b.d. | 93±62 | b.d. |
| Styrene | **277±67** | **117±21** | 57±40 | b.d. | 41±7 | 35±7 |
| 1-nonene | b.d. | b.d. | b.d. | b.d. | b.d. | b.d. |
| o-xylene | **1344±150** | **408±149** | 64±38 | 13±3 | 49±28 | **106±21** |
| n-nonane | **221±65** | **91±22** | 21±4 | 12±2 | 20±6 | 19±4 |
| i-propylbenzene | **92±36** | **50±15** | **30±70** | 15±3 | 11±8 | b.d. |
| α-pinene | b.d. | b.d. | 31±26 | 30±6 | 176±370 | 81±16 |
| n-propylbenzene | **271±48** | 94±43 | 20±13 | 66±13 | 12±6 | 88±18 |
| m-ethyltoluene | **832±136** | **214±131** | 31±26 | b.d. | 25±15 | **63±13** |
| p-ethyltoluene | **331±37** | **201±85** | 24±14 | b.d. | 18±8 | 20±4 |
| 1,3,5-trimethylben-zene (1,3,5-TMB) | **278±77** | **210±122** | **46±55** | 41±8 | 35±32 | 45±9 |
| Sabinene | b.d. | b.d. | b.d. | b.d. | b.d. | b.d. |
| o-ethyltoluene | **336±45** | **159±64** | 36±24 | b.d. | **67±30** | 30±6 |





| Octanal | b.d. | b.d. | 13±5 | b.d. | b.d. | b.d. |
|---|---|---|---|---|---|---|
| β-pinene | b.d. | b.d. | 15±8 | b.d. | 18±10 | 36±7 |
| 1,2,4-trimethylbenzene/ t-butylbenzene | **1514+292** | **462±127** | **63±37** | **172±34** | 43±19 | 45±9 |
| n-decane | **305±159** | **92±49** | 22±8 | **101±20** | 17±9 | 29±6 |
| 1,2,3-trimethyl-benzene (1,2,3-TMB) | **632±350** | **108±51** | **120±296** | **511±102** | 27±20 | 49±10 |
| limonene | b.d. | b.d. | b.d. | b.d. | b.d. | b.d. |
| eucalyptol | b.d. | b.d. | b.d. | 57±11 | b.d. | 24±5 |
| indane | 71±71 | b.d. | b.d. | 49±10 | b.d. | b.d. |
| 1,3-diethylbenzene | **187±6** | **57±40** | 13±11 | b.d. | b.d. | 17±3 |
| 1,4-diethylbenzene | **252±71** | **52±34** | **522±1380** | b.d. | b.d. | 11±2 |
| butylbenzene | **232±70** | **60±34** | b.d.l. | b.d. | b.d. | b.d. |
| n-undecane | **45±7** | 16±6 | 10±13 | b.d. | 22±10 | b.d. |
| n-dodecane | 24±13 | b.d. | 26±24 | b.d. | b.d. | b.d. |
| n-tridecane | b.d. | b.d. | b.d. | b.d. | b.d. | 10±2 |





1 **Table 6.** Particle number concentrations (bicycle/ van (background) measurements) for different land use types

2 in particles per cm$^3$. "-" indicates areas, which have not been tested by the method. This table provides the 25$^{th}$,

3 50$^{th}$ and 75$^{th}$ percentiles as well as the mean and the number of available data points.

| surface type | 25th | median | 75th | mean | no. of data |
|---|---|---|---|---|---|
| Urban - block build. | 8589/7555 | 13050/10110 | 21160/32915 | 25860/13390 | 55132/21646 |
| Urban -single build. | 6021/4550 | 9490/6181 | 15400/10080 | 17040/8861 | 139597/81293 |
| Industry | 6269/7201 | 8624/10614 | 16220/16710 | 16990/14488 | 9966/13784 |
| Com.+transp. | 5918/9807 | 8553/14240 | 14810/19040 | 14390/16281 | 4367/4856 |
| Airport | 5364/8308 | 6146/9424 | 7855/15930 | 7214/21970 | 968/781 |
| Parks | 4561/6555 | 7053/10680 | 11160/17820 | 12770/16736 | 14493/10287 |
| Arable land | -/2973 | -/4817 | -/7125 | -/7388 | -/9271 |
| Pasture | -/3733 | -/5733 | -/7050 | -/6343 | -/934 |
| Nat. grassl. | -/2878 | -/2878 | -/3233 | -/3586 | -/371 |
| Dec. forest | 3646/3846 | 5802/5467 | 10620/9169 | 12190/11865 | 28726/8806 |
| Con. forest | 3613/3501 | 4991/4993 | 8394/5658 | 8657/14630 | 38485/7020 |
| mix. forest | 3828/3501 | 6059/5093 | 10520/7685 | 11690/11865 | 7215/1810 |

