# Peer review of "BAERLIN2014 - The influence of land surface types on and the horizontal heterogeneity of air pollutant levels in Berlin"

_Atmospheric Chemistry and Physics, 2016_

## Referee Comment (RC1) · Anonymous Referee #1 · 25 Mar 2016

General comment

The paper discusses the spatial heterogeneity of atmospheric pollutants concentrations in the Berlin area measured in specific campaigns using mobile measuring platforms. Correlation with land surface types is also discussed. Some interpretations about pollution sources and pollutants transformation processes are given even if often discussed qualitatively. The paper is based on a large quantity of data, it is suitable for the journal and treats an important aspect of research. In conclusion, I believe that it could be published after a minor revision that takes into account my specific comments.

Specific comments

[Figure]

• I found a little confused the organization of the paper. It includes 4 appendices and a supplementary material. Given that there is not much difference between appendix and supplementary it would be likely better for the readers to have all this additional material in the supplement.

• Table A1 should mention that bicycles have a position sensor like for the van.

• Page 6 (line 4). It is indicated a temporal resolution of 6s but in Table A1 it is reported 12s.

• In table A1 it should be indicated the presence of a position sensor also for bicycles.

• Page 7 (line 7). Please change air-borne with airborne.

• Page 8 (lines 23-32). The instruments used in the background are the same as those used in mobile monitoring? It is important, especially for number particle concentrations, to have the same size range of the measuring instruments other wise the comparison could have a relevant bias.

• Page 9 (line 30). Table S2 is mentioned before table S1. Probably it should be better to change the order of sections in supplementary material.

• Page 17 (line 19). Table 7 is missing in the manuscript file.

• Page 17 (line 26). "found negligible" is not very quantitative. It would be possible to give a percentage?

• Page 19 (lines 1-12). The aspect regarding the difference in the size range of the instruments used is important in urban areas. References regarding urban size distribution should be included. Further, a more quantitative discussion of the expected differences should be included given that instruments starting to measure at 3 nm and 30 nm are used and this could generate differences larger than measured concentrations.

---

## Referee Comment (RC2) · Anonymous Referee #2 · 27 Mar 2016

The Discussion Paper by Bonn et al. provides a very comprehensive assessment of a wide range of measurements conducted during the BAERLIN2014 campaign. It is very timely and relevant as it combines measurements using different mobile platforms and instruments to address key issues related to the spatial variability of urban air pollution.

In its current form, however, I see some shortcomings with regard to structure and presentation which I feel need to be addressed to make the paper accessible and relevant to a wider scientific audience. I hope that the following general and specific comments support this process, as I would like to see this paper eventually published due to its undisputed contribution to the current scientific discussion.

Structure The paper covers a lot of ground: 3 different mobile measurement platforms

(bike, van, aircraft) plus canister measurements; VOC speciation, particle number counts, atmospheric trace gases, both secondary and primary pollutants ... This is reflected by the structure and - not to forget - the length of the paper and is one challenge the authors need to address. Primarily in Section 4, it becomes obvious that a clear, concise structure to relate these different topics in an accessible way is difficult, which adds to the whole section being quite hard to read: structured by trace gases, then individual gases in 4.1.x, then Land use type influences (4.1.5), while in 4.2 addressing particulate pollution, the structure is done by type of measurement platform. There are different approaches conceivable, including splitting the paper up into two, either by method, pollutant or into approach/methods and application/results. Ultimately, the Results and Discussion section over 14 pages of length needs to be revised structurally to present a more clear pathway for the reader and to better highlight the important aspects. In addition, a suggestion would be to consider omitting the land use type classification and temperature sections, which could be moved to Suppl. Mat or dropped (see below).

Objectives The paper identifies the following objectives in Section 2, which it sets out to address: (1) What is the spatial and temporal heterogeneity of pollutants in the BBMA area with a focus on Berlin and Potsdam? (2) How do different vegetation types influence the levels of ozone, NOx and VOCs in Berlin? (3) What is the impact of different types of vegetated areas on urban environmental conditions i.e. temperature, humidity and particulate pollutants (number and mass)? (4) And finally what is the contribution of anthropogenic and biogenic organic compounds to secondary organic aerosol and the total particulate mass in the Berlin and Potsdam area affecting health, both directly and indirectly through ozone production?

In my opinion, these should be revised and are currently a bit too ambitious, as in the current version, the paper addresses (1) partially (to assess temporal heterogeneity, the duration of the measurements is too short in time and not sufficient in my view to adequately capture temporal variation in both space and across seasons, time of day,

day of week etc.). For (2), the use of the 100 m 2006 CORINE landcover maps are not sufficiently detailed, so I would suggest to either address this (see below), or omit this objective. (3) is addressed in the discussions , but does suffer a bit from the spatial resolution of the landcover mapping. (4) finally is well addressed and in itself a vital question to cover, as well as possible to discuss with the measurements undertaken to a large extent. Following from the objectives, the authors qualify these further, which I feel is not necessary once the objectives are clearly laid out and revised. One note, the "aim of this study is to identify hotspots of pollution" (P4L36), this is not really included in the objectives and given the obvious limitations of the spatial coverage, I would suggest to remove this. It is without doubt a valuable thing to do, but the value of this study is not in comprehensive spatial coverage of the city (or parts of it), but the addressing the variability of the pollution fields with different methods and measurement platforms.

Landcover As indicated above, my concern with using a 100m landcover map to derive robust classifications for urban land use/cover and how it affects local pollution levels at a very high resolution (both van and bike measurements allow for a very high temporal and thus spatial resolution of pollution variabiltiy) is not adequate. Street canyons and street vegetation, as well as local parks and green spaces in Berlin will likely affect the microclimate and pollutant dispersion at a spatial scale well below 100 m, so the uncertainties introduced by using this dataset have to be expected to be significant. Furthermore, the question in how far the 2006 LCM reflects the 2016 situation needs to be addressed, as the 2006 maps will be based on imagery that might be even older? To remedy this, I would advise to either drop this part from the paper (it is already substantive enough to stand alone without this analysis), or use a different, more up-to-date and spatially resolved resource to analyse the landcover in the area under investigation. Aerial photography and alternative land cover information (e.g. Open Street Map) on the one hand will likely be freely available through GIS resources, so this should not be a major issue. On the other hand, the use of the camera on the bikes would offer a more immediate resource to classify the immediate surroundings,

including density of buildings, street-canyon situations and other influencing factors not immediately available from land cover maps (construction sites, local changes in layout or buildings etc.). I appreciate that this would need more work, thus suggesting that the influence of land cover on the variability of pollution perhaps needing its own paper to be adequately addressed.

Uncertainties The paper addresses a range of measurements using different instruments, which is inevitable, but will require a discussion of uncertainties arising from the differences in instruments and measurement techniques. One aspect, which is not addressed currently, for instance is the issue of deriving (indicative?) particle mass concentrations from optical instruments (or has gravimetric analysis been done on the GRIMM filters, which would then obviously cover longer periods, not allowing easily for a detailed temporal allocation of particle mass?). In addition, the potential contribution from long-range vs local sources has been raised in the paper, with the conclusions seeming to contradict recent literature (e.g. Kiesewetter et al. 2015, doi:10.5194/acp-15-1539-2015; Vieno et al. 2016, doi:10.5194/acp-16-265-2016) identifying a substantial contribution of long-range transported ammonium nitrates and -sulphates for large parts of European PM2.5 and PM10 concentrations, so a discussion of chemical speciation of the aerosols measured would need to be included to justify and support this claim.

I hope these general comments support the revisions, together with the following specific issues: Abstract: -P1L19 ozone and particulate matter are specifically mentioned, but not other trace gases, in particular NOx, why? -P1L23 "between the June 2nd" - remove "the" -P1L26 "compounds and particulates and..." - suggest using a comma first, then 'and' -P1L33 "reduction of temperature" - specifically which, max, mean? -P1L36 "pointwise" unclear what you mean here, specify please -P2L1/2 "on the scale of one hundred metres" this is rather unclear, please elaborate your spatial reference -P2L3 "mass concentrations being local" - see comment under 'Uncertainties', not sure this claim is so far well supported by the paper as it stands. -P2L7 "facilities for sports and

leisure" how do these influence concentrations specifically? are you referring to open fields, or swimming pools and sports centres, which would likely have very different influence

Introduction P2L17 "are already causing" - looking at emissions of air pollutants in the Western developed countries, the peak of NOx and other pollutants was in the 1980-2000 period, so air pollution has been causing health effects for a while. Suggest to drop "already". P2L27-33 This is one long and complex sentence which I suggest to split into 2 or 3 parts. could you briefly introduce "oxygen capacity" for a more general audience P2L37 "As held by the ..." this formulation is a bit awkward, could you rephrase the sentence, best switch it around to start with "Establishing such air quality programs ..." P3L3 space missing before "As a..." P3L6 "... can sue for an adjustment..." is rather German, perhaps better "can take legal action" or similar? P3L9 "In consequence ..." not a good start to a sentence, suggest to drop or reformulate P3L11 "respective" delete, not necessary here P3L11 "limit values continues to" - remove plural s from continues P3L12 "contained herein" reference is not clear, suggest to reformulate the phrase P3L15-18 another long sentence, suggest to break it up P3L18 "has been claimed to" ... by whom? where? only one 2007 reference is provided, but long-range transport contributions to PM10/2.5 have been subject to a lot of most recent literature, which should be referenced and acknowledged P3L23 "Due to their provision..." this sentence does not logically follow from the previous, I suggest to introduce a new paragraph here, or link it better P3L36 "the presented study tries to support city authorities" - does this refer to the paper, then it is yet another objective not introduced before, but if it refers to BAERLIN2014, this needs to be clarified P3L36-37 "supporting authorities" is mentioned twice, so trying to support authorities by supporting authorities? check and revise, please P4L3 "and a hub for major transport routes" better "a major European transport hub" P4L7-9 "impact on pollution levels ... and thereby on pollution levels" please check, this seems a circular reference here P4L9 "generally meet the EU limit values" - how does this relate to the adverse health effects outlined in the introduction before? I do not challenge the fact, but it would better be explained

a bit more to the audience, as a reader could feel that if limit values are widely attained, why is there a problem to investigate? P4L12 "and transport of" better qualify this as "atmospheric transport" or "long-range transport" to distinguish from road transport activities P4L16-19 why is this text set in italics? is this a quote, then by which source, or is this a key statement, then it is not founded anywhere in the current text. Suggest to remove, put in a box and explain, or add further reference. P4L23-30: I take it these are the objectives of BAERLIN, but it is somewhat confusing, so I would suggest to make these rather explicit and refer them to the overall study objectives of BAERLIN, which could e.g. be put in suppl. mat., otherwise it may confuse the reader quite a bit. P4L38 the reference to identifying dominant VOC sources to support action plans for the Senate seems to be a bit unrelated to the overall paper, with the exception of the canister studies, so wondering if this needs to be here, or should rather be in the conclusions as one potential area that the results of this paper could be used for? P5L8 "aircrafts" - remove 's' P5L14-19 I would suggest not to use 'mesoscale' here, which in my view is not quite right with the scales addressed by the different studies? Or explain what you explicitly mean by the terms in this context? P5L28 "that cars cannot" reads a bit awkward, could you rephrase e.g. as "areas that cars cannot enter"? P5L32 "particulate values" here and subsequently, could you make sure to be very precise what 'values' you are referring to, as both PNC, PM mass and other parameters are used in the study? P6L2 "Applied as well was ..." not a good start to the sentence, try to activate as much as possible, e.g. "The optical particle counter GRIMM 1.108 () was applied for ..." P6L5 can you elaborate on the setup here, if the instrument was covered in a backpack or pannier, how was uninhibited constant airflow guaranteed? perhaps add a picture of the instrument setups in the suppl. mat? P6L8 "Please find the detail ..." I would skip this sentence, not needed P6L11 introduce IASS at first use P6L17 "while the sampling frequency ... was relatively high" how did you match time scales/steps for all the measurements and the GPS? This should be introduced somewhere early on as it will be rather variable across instruments and methods. P6L27 "Location data was collected via GPS" and camera, this could be a means to derive contextual information, in addition to a time-activity diary? Was this considered? P6L29/30 how was aerosol mass measured in real time, can you elaborate on this here, as it is rather crucial for the interpretation of the results, and not trivial to achieve. P6L33 "a specific track was carried out", suggest to reformulate, e.g. "a pre-set route was followed" or similar P6L35-P7L5 the quantitative information would be better displayed in a table or graph than in the text here P8L3 "Method of relative parameters" not quite clear, suggest to rephrase e.g. "Method for deriving relative concentration parameters" or suchlike P8L29-32 again, time steps are mentioned here, but it is not clear how temporal resolution of the measurements has been harmonised/addressed, suggest to add a paragraph earlier on to address this. P9L1-14 as indicated above, I am not convinced that at 100 m x 100 m the land use types can provide a meaningful basis for the analysis. My suggestion would be to remove section 3.5 entirely P9L15ff As indicated in the general comments, suggest to revise the structure of Section 4 overall. P9L29 the part on the leaf blower seems to be marginal and not related to the objective to derive more general insights into the spatial variability. Could you explain better why this is important, or remove that part? It does seem to be a rather specific issue. P11L35-37 first sentence on CO is giving a generic statement about similar patterns for all gases, I would suggest to carefully check the paper and remove these, as they are repetitious and generic. Furthermore, in the results and discussion, I would not go into as much detail to explain the general sources of CO and its formation in urban environments, as done here, it just adds more text distracting from the valuable findings of this study. P12L15 "BLUME station" may have missed this earlier, but could not find another reference to this station name, so best introduce earlier P13L1 "if and only if" please avoid such phrasing, it is not needed here to emphasise P13L14 "diesel driven" ... "diesel consuming" remove driven/consuming, just "diesel passenger cars/LDVs" is sufficient. P13L16 "to the measured nitrogen dioxide mixing ratios" do you mean direct emissions of NO2 from diesel oxidation catalysts? I would then make this more direct and clear, it is a bit back-to-front else. P14L1ff as indicated above, I am not convinced by the results based on the coarse land use type resolution, and suggest to drop this.

P15L32-38 again, a rather generic basic introduction to particulate matter, which I suggest to skip as it is not really necessary here, perhaps add one reference in a short sentence to introduce this? P16L3 referring to 'small scale variation' here, which I think is fine and relates to my comments on micro/mesoscale wording earlier P16L12 "mixed layer height" do you mean mixing layer? P16L23 "applied for the" applied to? was applicable to? P17L6 "Particle mass concentrations ..." see comment above, could you elaborate somewhere how mass was measured, in the context of optical instruments being used P17L15 "on the regional and local scale" see above, please use a consistent spatial reference for the different scales addressed P18L6 "particulate masses" please be more accurate and specific in referring to parameters, particulate matter mass concentrations (of PM10? 2.5?) or PNC? P18L28 "Please take into account ..." I would suggest to drop such formulations, they just add words and no substance, rephrase to "The measurements with the van were conducted by following ..." P20L9ff see above, suggest to drop the section 4.2.4 P21L20ff "For most of the land use type classifications the differences between the van and bicycle measurements agree within the associated uncertainty" what is the associated uncertainty and how is it derived? Due to the short term of the measurement campaigns, the conclusions on the heat island effect would likely need more supporting work. Not sure if a discussion of the heat island effect here in this paper is necessary, and the caveats are outlined already in the text following on the same page. Consider shortening or removing? For a more thorough comparison, looking at the share and distribution of green space areas in different cities would be essential, in my opinion. P22L8 "characterise air quality on multiple scales" I think this is a bit of a leap, the study very well demonstrated the capability of mobile measurement platforms to quantify specific air pollutant concentrations, in a one-off campaign based mode, so perhaps better stick to this in the formulation? P22L9 "large geographical area" related to earlier comments on consistent reference to spatial scales, 'large' is relative and best quantify here "an area covering X square km"? P22L17/18 "elevated air pollutant concentrations found in Berlin were most likely produced in the vicinity of the observation and originated from local pollutant sources" -

while I accept this for NOx and CO, I am yet to be convinced by the findings presented here that this is the case for PM2.5, which would require a discussion of the chemical composition of the PM observed and a look at the regional-scale atmospheric transport processes; previous material presented by the Senate of Berlin indicated a substantial amount of PM originating from long-range transport, and recent literature has shown this for Europe in general and several European cities, so the role of ammonia and secondary inorganic aerosols should be more thoroughly assessed before this claim can be substantiated. P22L27-32 could you quantify the 'significant' influence of vegetation on pollutant concentrations? Was it statistically significant? Contribution of isoprenes and terpenes to local ozone formation, was this quantified in the study, or is this derived from models/previous knowledge? The second half of this statement on urban vegetation is more robust and accepted. P22L33-37 again, the resolution of the land use maps considered makes this statement harder to justify, in addition, urban airflow patterns and complex terrain influences on wind and dispersion would need to be taken into account adequately, which is not within the scope of this study for good reason. Perhaps this section needs to be qualified a bit to reflect these caveats.

Figures: - general point, consider making the background maps slightly less vibrant to better bring out the colours of the measurements, in particular the orange and yellow shades are hard to see. - Fig 6: add more legible legends to the graph - Fig 8: what are the units for the upper graphs, please add to the legend

Tables: Table A1: formatting of the table makes it a bit hard to read, i.e. alignment and space between columns; time resolution is variable for the instruments, relating to the comments made above on time-synchronisation

ANNEX: A2L12 "All particle instruments except the instrument were ..." which 'instrument' are you referring to? Figure B1: add the unit to the legend in both cases Table C1: This table is rather dense and could considered to be more accessible as a bar chart? Figure C2: map zoom and focus is different, making a direct comparison between PNC and mass concentrations difficult, for no reason? suggest to make sure

that both maps show the same area

REFERENCES For a paper of this substance, some of the recent literature in particular with regard to urban PM seems to be missing, e.g. from the CLEARFLO project (http://www.clearflo.ac.uk/outreach/papers/), as well as those on long-range transport contributions and composition of urban PM (see specific comments).

―――――――――――――――――

---

## Author Comment (AC1) · 21 Apr 2016

We would like to thank the anonymous reviewer very much for the detailed and constructive comments with respect to our study. In the following we will respond to the specific comments given to show the improvements suggested by the necessary changes where feasible and to indicate our motivation in the way of presentation (appendix).

1) Structure of appendix and supplementary material: Reviewer 1 asked for shifting the current appendix part to the supporting online information (SOI) as the difference between both parts was not clearly resolved. We agree on this and would only prefer to keep the additional information about the measurements (Table A1) included as

appendix in the main document as measurement methods, major results and conclusions should be incorporated in a single file for download later on. The other parts of the appendix will be shifted to the corresponding part in the SOI.

2) Including the position sensor in Table A1 (mentioned twice): Excellent remark and immediately done.

3) Temporal resolution of particular mass measurements via bicycle platform: Correct remark. The time resolution of the GRIMM1.108 instrument got mixed with the temperature sensor linked to. Will be corrected to 6 s.

4) p. 7, "air-borne": Is a typo and is corrected. Thanks.

5) Page 8 (lines 23-32), instruments at reference site and mobile ones: The comment is very important and our approach will be explained: For PM measurements the instruments used were of identical type (GRIMM 1.108). But the types differed for particle number concentration measurements. This is partly caused by the need for different techniques for mobile measurements with an unsteady surface. The usage of a condensing agent to make particles grow to detectable sizes is impossible. The mobile technique is novel and was available only once. Because of this we applied regular parallel measurement phases when visiting the reference site and checking the stationary instrument status. Based on these repeated measurements of about every second day for 30 min we made scatter plots of stationary and mobile measurements results and derived a calibration factors for the individual parameters. These factors staid pretty constant throughout the three months and are listed with standard deviations in Table S1.1.

6) Order of S1 and S2 in the supplementary material: Good point. Will be done.

7) The lack of Table 7, PM10 effects: Thanks. It was not missing but shifted to the appendix without correcting the reference number. As we moved the tables and figures to the SOI, we move back Table C1 to Table 7 to keep it in the document.

8) P. 17 l. 26, negligibility of larger particle sizes with respect to particle number concentrations: Nice suggestion. We will add the median percentage of the particle number concentration (diameter Dp >300 nm) to total particle number concentration 0.63% (mean: 1.2 ± 1.4 %, non-Gaussian distribution) to the text next to "negligible" and place figures to the SOI displaying the cumulative particle number concentrations for the individual types (stationary in Berlin-Neukölln during more than 2 months and mobile via van for 6 days of measurement). The figures can be seen as Fig. R1 below. The median values display 16% of the particles to be larger than 100 nm and 63% of the particles were found between particle diameters of 20 and 100 nm. Below 15 nm only 12% were found by the van measurements. Hopefully these numbers elucidate our remark "negligible" somewhat. Please note that these relative contributions refer to particle numbers but not to mass distributions. For particulate masses about a third (38%, stationary background) or a half (median: 46%, mean: 56%, mobile) is larger than 1 micron, about half (52%, stationary) or 75% (median, mobile)- 81% (mean, mobile) is larger than 500 nm and 90% (stationary) or nearly 100% (mobile) is larger than 100 nm in particle diameter. In here stationary measurements show a much better agreement of mean and median than the mobile observations most likely due to on-site production and rapid aerosol dynamic processes beyond equilibrium.

9) Page 19 (lines 1-12), effect of particle size detection ranges in urban areas with respect to number concentration: It is certainly important to consider the slightly different size ranges detected by the stationary TSI 3776 and the mobile DiSCmini (see Table A1). While generally different detection ranges in the lowermost range would be expected to result in different results especially in areas of intense new particle formation (Seinfeld and Pandis, 2006; Kulmala et al., Nat. Protoc., 2012; Ma and Birmili, Sci. Total Environ., 2015), the results are hardly different below 10 nm in particle diameter. The cause remains unresolved but speculations deal for example with (a) the extensive growth rate of freshly formed particles about 2 to 5 times larger than for instance in remote areas [Dal Maso et al., Boreal Env. Res., 2005; Ahlm et al., J. Geophys. Res., 2012; Ma and Birmili, Sci. Total Environ., 2015; Yu et al., Atmos. Chem. Phys., 2016]

and (b) unstable partially organic particles that crack and vanish during detection (substances splitter and escape their detection) [Bonn et al., Atmos. Chem. Phys. Diss., 2007].
* * *
[Figure]

**Contributions particle number concentration**

Cumulative contribution to total particle number concentration above 10 nm in particle

| | |
|---|---|
| —— | mean +/- std (stationary) |
| - - - | median (stationary) |
| —— | mean +/- std (van) |
| - - - | median (van) |

(y-axis) cumulative contribution in %

(x-axis) $D_p$ in μm

**Fig. 1.** Cumulative contribution to total particle number concentration above 10 nm in particle diameter starting at the largest size of 20 micron. Displayed are two datasets, i.e. (a) based on the stationary

---

## Author Comment (AC2) · 4 May 2016

We thank reviewer 2 for the time spent and the encompassing and supportive comments to improve the presentation a lot. As done for the other review, the reviewer's comments will be discussed below and our changes to address the comments in the manuscript are described below.

[Figure]

**General**

*The reviewer points out that the length of the study and its presentation structure should be modified, i.e. shortened and condensed to make it relevant for a larger community and number of readers. This is certainly true.*

**Structure**

*The general comments are reflected in this paragraph. I agree with the reviewer that results and discussion section of 14 pages may be too long and should be shortened. So we'll do that. The structure of "Results" shall be focussed on and condensed to (individual) gases, particles (number concentration and mass) and land use types. Different measurement platforms won't be separated anymore and will be part of "individual gases" and "particle properties". Because this article is supposed to be on overview presentation on the "mobile" campaign part, which was designed to elucidate the effect of urban vegetation on pollution levels, the individual sections can and will be cut down to focus the text, concentrating on the major findings. However, we find that the key features, i.e. the influence of different land usage types on pollution and the horizontal heterogeneity, should be kept. In relation to later comments regarding the land use classifications and how they are applied, we will lump certain land surface usage types to more general classes.*

**Objectives**

*There is probably a misunderstanding in the word "objectives" as the four aspects listed were named as parts of the objectives not as all-encompassing. No project can*

do so. *"Objectives" is meant more in terms of big picture research questions. To avoid this confusion, we suggest reducing the present list to three points, making their focus more specific and add one specific that was requested in the comments later on:*

1. *"Heterogeneity of **particle number and mass concentrations throughout the city characterized by different sources and sinks including green areas.***

   *This will be extended by a further study dealing with the bicycle measurements and their classification by the camera, which is in progress since some months due to the large amount of video material acquired.*

2. Influence of green spaces/areas on urban pollutants ($NO_x$, VOCs, ozone and particles) levels;

3. *Contribution of anthropogenic and biogenic organic compounds on particulate levels and on ambient ozone;*
   *and*

4. **Provide support for the city authorities for future action plan development to improve air quality.**"

   ***The present study is one of two overview articles on the BAERLIN2014 campaign addressing the mobile observations and analysis, while the second (von Schneidemesser et al., in prep.) will focus on the stationary measurements and source apportionment. The investigation of the link between $NO_x$, different VOCs and SOA was split off to a box model study and will be described in a further article.***"

*Hopefully this would make things easier to follow and better tracked by the current and potential future papers planned from the study. In order to do so point (1) will be*

*addressed exemplarily by selected single bicycle and van measurement tracks with a reference to the future bicycle study. This was the aimed at so far previously but will be concretized focused to make it more clearly. Point (2) is being covered in the section about VOC-canister samples at distinct characteristic sites as well as by mobile van measurements of NO$_x$ and ozone with consideration of bicycle, van and air-borne measured particle concentrations. Point (3) is discussed as noted by the reviewer in the "Results and discussion" section already. Although this could be more extensive, the further analysis details are expected to be presented in a further particle focused study. A further change because of the newly shaped foci of this study will be the structural change (see above comment) (a) observations of pollutants and (b) discussion not separated but combined. Based on our three different objectives the observations will be presented and discussed: (1) Differences of pollutants in green and non-green areas/spaces including the heterogeneity, (2) Influence of green areas/spaces on particulate pollutants and its heterogeneity in space and time as well as (3) a conclusion the contribution of anthropogenic and natural sources to particulate levels.*

**Landcover**

*The comment on skipping or improving the landcover analysis is a critical issue that has been discussed among the co-authors significantly, including the investigation of a variety of alternatives for more than a year and is worth discussion. While the resolution is fairly low (100m x 100m), which is critical for more detailed analysis, it is the only data directly accessible, reliability checked and including the information needed for analysis of the datasets obtained. In addition, it is used in other applications (Statopoulou and Cartalis, 2007; Janssen et al., 2008; Tomaselli et al., 2013). Geographical street maps (e.g. www.openstreetmap.org) were accessed but were lacking much of the information on green spaces that was available in CORINE, i.e. the one used, despite the lower resolution and year. The format (polygons) was found incompatible with the*

*database of GPS coordinates created. As the land cover refers to the major point of this study, i.e. the influence of green spaces, which change more slowly than buildings, we chose this map however with notable care not to interpret aspects beyond the range of significance and discuss where evidently overlapping effects occur and affect the results. The significance of the effect of individual land cover types would certainly increase the higher the resolution and the better the distinction between different types. But so far this is not possible. While the land usage type classification was done for 2006, i.e. 8 years before the project start, some changes will have occurred (buildings, building areas etc.) but much less so in green spaces, and even less so in large green spaces, such as Grunewald, a large forest area where many of the bike routes passed through. Those require substantially more time and small effects will become more obvious in the analysis of the bicycle data and its video files recorded. For this reason, the CORINE map classification was determined to be sufficient for an initial analysis.*

**1   Uncertainties**

*Uncertainties are always a key aspect of the reliability of data. Again we would like to stress that we (i) listed all the available measurements with instruments and their corresponding range of uncertainty and that we (b) have performed different calibration actions to exactly elaborate this aspect. As mentioned in the text on p. 37, l.12-13 all particle instruments except the DiSCmini were calibrated a priori at the Leipzig Institute for Tropospheric Research, i.e. the World calibration centre for aerosol measurement calibration. The DiSCmini instrument was checked and calibrated at UBA in Langen. This latter "instrument" mentioned in the text without DiSCmini will be corrected and "**DiSCmini**" pasted in as it got accidentally dropped as the text was rearranged during the writing. In line 12 all particle instruments will we named in brackets: "**(GRIMM 1.108 (2x), GRIMM 5.403, GRIMM 5.416 and TSI 3550 NSAM)**". Some measurement techniques and instruments were calibrated by chemical standards (PTR-MS, canister*

*sampling with GC-MS analysis) and cross-checked with the local continuous measurement of the BLUME network at Berlin-Neukölln (benzene, toluene). The DiSCmini, which was the only particle instrument not calibrated in Leipzig but in Langen, and the GRIMM 1.108 were compared regularly during the stops at the station in Neukölln with stationary measurements by GRIMM 5.416 and the stationary GRIMM 1.108 for about 20 min. The later comparison was operated by two calibrated instruments i.e. the stationary GRIMM 1.108 and the mobile GRIMM 1.108 (bicycle) that were calibrated a priori in Leipzig and checked thereafter, which no significant difference. Furthermore van and bicycle measurements were cross-checked in two joined tracks i.e. cyclist ahead and van following. Comparison figures (temporal lines and scatter plots with identical averaging time slots) will be provided in the supporting online information (SOI). We agree that different particle measurement techniques (gravimetric and optical) for particulate mass provide different results is based on the assumptions that a certain aerosol particle composition, used for calibration and applied for anywhere else in future usage, implies. We will insert a sentence on this in the "methods" and "results and discussion" sections each, when dealing with the particle instruments as results of both techniques are used in here. With respect to the source speciation of the aerosol particle mass we agree to include a discussion about the results from Kiesewetter et al. (2015), which are not contradicting completely but focus on a broader approach with similar assumptions for entire Europe and on the entire seasonality. Please note that we focussed on experimental studies at a single location and not on model challenges in here. We mentioned that a substantial particulate mass contribution is deriving from local sources for the time of study (June to August 2014), while the Kiesewetter et al. (2015) study investigated entire Europe on the seasonal scale and averaged. Other studies such as Kerschbaumer (2007) indicated remarkably contributions of PM from the industry of Southern Poland during winter time. This was probably not found because of a different meteorology during summer (westerlies) compared to winter time (easterlies). An accompanying study (von Schneidemesser et al., in prep) on the stationary measurements will deal with a principal component analysis on sources even*

*further, for which a reference will be set in this study in order not to overstretch the length.*

**Specific points**

Abstract, p.1, l.19:
*Ozone and particulate matter were referred to because of the aim of this project, which was focused on the role of green spaces in ozone production. $NO_x$ is certainly important especially in areas affected by traffic, industry and burning processes, and was measured but not the main focus of the study.*

p.1, l.23  30: *OK.*

p.1, l.33: *We are not really sure about the comments meaning, i.e. "max, mean", which were not used there. The temperature and particulate levels measured by bicycle platforms displayed significantly lower i.e. reduced values in vegetated areas. This can be classified by mean+/-standard deviation as well as median, which was done in the boxplots provided later on. We will reformulate this particular sentence to rule out misunderstanding.*

p.1, l.36: *'pointwise' addresses the measurement method by canister samples and analysis later on in the lab. As this particular method could not be applied all the time representative points were selected and during the tracks one or several samples were taken.*

p. 2, l.1/2: *'scale of one hundred meters':*

*The individual measurements were compared to other measurements in the surrounding area for a radius of 100 m. Therefore, for any data point sampled the observations in a spatial area of 100 m distance and time within ±1 min were extracted individually and correlated. We'll change the current form to 'For example, **moving average concentrations of the traffic related chemical species CO and NO varied by more than ±20% and 60% over the distance of one hundred meters around any measurement location**, respectively.'.*

p.2, l.3: *Regarding the particulate mass observations and uncertainty of observations we refer to two identical observations, i.e. the calibrated GRIMM 1.108 as stationary and mobile instruments. Both applied spectroscopy as measurement method, no gravimetric methods. While the different methods can certainly cause differences in absolute numbers, this does not apply for relative measures as given in this sentence with two identical measurement instruments.*

p.2, l.7: *'facilities for sports and leisure' refers to the CORINE classification. Of course this includes a variety of different surfaces with different impacts. Thanks for the remark. Any classification type, i.e. based on meteorology, biology or productivity will cause a shortcoming in one of the other areas. As noted above we suggest lumping the individual types to more general types with less information but a larger set of data. Furthermore, to address issues with the method, we are removing this category as it has fewer data points and is less robust than some of the other categories.*

Introduction, p.2, l.17: *Skipping 'already' is OK.*
p.2, l.27-33: *Yes we can and will. The sentence got too long and will be split into two. An additional sentence, explaining 'oxygen capacity' as '**the maximum quantity of oxygen that will combine chemically with the hemoglobin in a unit volume of blood [free medical dictionary, accessed April 25th 2016] and that can be used***

**by the body for brain and physical working**' will also be added.

p.2, l.37, ' 'As held by the ...' this formulation is a bit awkward, could you rephrase the sentence, best switch it around to start with "Establishing such air quality programs ...':
*Will be changed to 'Establishing such air quality programs is a subjective right of any person directly concerned and can thus be claimed by citizens in court (Janecek v. Bayern, ECJ, 2008).'.*

p.3, l.3: *Missing space will be added. Thanks.*

p.3, l.6: 'Correct, the expression derives from German to English translation.':
*This will be corrected.*

p.3, l.9, "In consequence ...' not a good start to a sentence':
*The expression 'in consequence' will be dropped and the sentence shall start with 'Berlin, like every. . .'.*

p.3, l.11, 'suggest to drop or reformulate 'respective' delete, not necessary here':
*Will be changed to 'The Senate of Berlin **thereto** adopted a **clean air** program for 2011-2017 (Berlin Senate, 2013b).'*

p.3, l.11, "limit values continues to' - remove plural s from continues':
*Will be done*

p.3, l.12, "contained herein' reference is not clear, suggest to reformulate the phrase':
*Will be changed to '. . ., it is questionable whether the **intended** measures are sufficient*

*to enable Berlin to comply with this obligation.'*

p.3, l.15-18, 'another long sentence, suggest to break it up':
*Changed to 'This measure was **intended** to lower traffic related emissions and the* **annual** *number of critical threshold exceedances according to EU law for $NO_x$ and PM (see Table 1) in Berlin. **It** resulted in an emission reduction by 20% for $NO_x$ and 58% for soot by diesel engines (Berlin Senate, 2011).'*

p.3, l.18, "has been claimed to' ... by whom? where? only one 2007 reference is provided, but long range transport contributions to PM10/2.5 have been subject to a lot of most recent literature, which should be referenced and acknowledged': *Agree. But this study was focussed on the Berlin-Brandenburg area, for which only this particular study was available so far. Oher studies using coarser resolved models may not be easily transferable. However, we will reformulate the particular sentence and add the following information to that: 'The study by Kerschbaumer (2007) has claimed a substantial contribution to $NO_x$ and particulate matter (PM) by long range transport from Polish industrialized areas. Several studies (Kiesewetter et al., 2015; Amato et al., 2016) conducted elsewhere supported this claim, while others (Petit et al., 2014; Mancilla et al., 2016) contradicted and identified local sources to be dominant.'*

p.3, l.23, "Due to their provision...' this sentence does not logically follow from the previous, I suggest to introduce a new paragraph here, or link it better':
*Will be done as follows: We will start a new paragraph and improve the link of both sentences. 'These vegetative areas are supposed to have notable effects on temperature and air quality. Therefore, increasing green areas such as parks and forests are often considered as measures to counteract urban heat island effects (Fallmann et al., 2014; Grewe et al., 2013; Schubert and Grossman-Clake, 2013) and air pollution problems (Irga et al., 2015; Janhäll et al., 2015).'*

p.3, l.36, "the presented study tries to support city authorities' - does this refer to the paper, then it is yet another objective not introduced before, but if it refers to BAERLIN2014, this needs to be clarified':

*Thanks. The support is provided as expertise to the collaborating partner 'Senate of Berlin' and is basically included in the foci mentioned. In order to clarify this we will be named in the list of foci of this study. But please note this study in one of two overview articles on the BAERLIN2014 project, with the second one (von Schneidemesser et al., in prep.) dealing with the stationary results. This latter study will be of most interest for the Senate as it was bound to a monitoring station. The present one will provide information about the differences across the urban area, effects of green areas/spaces and sources.*

p.3, l.36-37, "supporting authorities' is mentioned twice, so trying to support authorities by supporting authorities?':

*? We'll clarify that by reformulating the sentence: 'The aim of this study was to identify hotspots of pollution, the variability of basic air pollution trace gases, to quantify the impact of green areas and to exemplarily identify dominant VOC sources to support future development of action plans by the Berlin Senate with improved success.'*

p.4, l.3 'and a hub for major transport routes' better 'a major European transport hub':
*Will be done.*

p.4, l.7-9 'impact on pollution levels ... and thereby on pollution levels' please check, this seems a circular reference here':
*Indeed. Corrected. Thanks.*

p.4, l.9 "generally meet the EU limit values' - how does this relate to the adverse health effects outlined in the introduction before? I do not challenge the fact, but it would better be explained a bit more to the audience, as a reader could feel that if limit values are widely attained, why is there a problem to investigate?':
*Will be done in more detail.*

p.4, l.12 "and transport of' better qualify this as 'atmospheric transport" or "long-range transport' to distinguish from road transport activities':
*Will be changed to 'regional and long-range atmospheric transport of'.*

p.4, l.16-19 'why is this text set in italics? is this a quote, then by which source, or is this a key statement, then it is not founded anywhere in the current text. Suggest to remove, put in a box and explain, or add further reference.':
*This was used to emphasize. We will remove the italics.*

p.4, l.23-30: 'I take it these are the objectives of BAERLIN, but it is somewhat confusing, so I would suggest to make these rather explicit and refer them to the overall study objectives of BAERLIN, which could e.g. be put in suppl. mat., otherwise it may confuse the reader quite a bit.':
*Yes, they are. See as suggested above in the 'Objective' part.*

p.4, l.38 'the reference to identifying dominant VOC sources to support action plans for the Senate seems to be a bit unrelated to the overall paper, with the exception of the canister studies, so wondering if this needs to be here, or should rather be in the conclusions as one potential area that the results of this paper could be used for¿:
*We will add a sentence to the conclusion section as this VOC speciation affects ozone, PNC and PM production. The relevance of certain species with respect to those*

*aspects will be named in a short sentence.*

p.5, l.8: 'aircrafts' - remove 's': *OK.*

p.5 l.14-19 'I would suggest not to use 'mesoscale' here, which in my view is not quite right with the scales addressed by the different studies? Or explain what you explicitly mean by the terms in this context?':
*We will explicitly describe the resolution of observations expressed by the different scales and the major focus for using that particular platform. Thus when 'mesoscale' is used so far a suburb or city area is meant (resolution of hundreds of meters to tenth of kilometres), while microscale expresses street canyon and finer resolution (meters to tenths of meters). This will be named accordingly in the future version of the text.*

p.5, l.28 "that cars cannot' reads a bit awkward, could you rephrase e.g. as 'areas that cars cannot enter'?':
*Done.*

p.5, l.32 "particulate values' here and subsequently, could you make sure to be very precise what 'values' you are referring to, as both PNC, PM mass and other parameters are used in the study?':
*Sure. Two brackets including the individual parameters for the bicycle measurements have been added to the text: "...quantifying meteorological **(temperature, relative humidity)** and particulate values **(number, mass and lung deposable surface area concentrations)** are listed...:*

p.6., l.2 "Applied as well was ...' not a good start to the sentence, try to activate as much as possible, e.g. 'The optical particle counter GRIMM 1.108 () was applied for

...':
*Changed to 'We deployed the optical particle counter . . ..'.*

p.6, l.5 'can you elaborate on the setup here, if the instrument was covered in a backpack or pannier, how was uninhibited constant airflow guaranteed? Perhaps add a picture of the instrument setups in the suppl. mat?':
*Yes, both will be done. 'Backpack or pannier' expressed that instruments were provided to cyclists with different bicycles and they got the setup for either storing it inside the backpack (provided too) or in a pannier of the cyclists own equipment. An introduction how to set-up the equipment and which aspects to take special attention for was given to each participant. This will be noted in the supplementary material and added in a single sentence to the text: '. . . were transported in a backpack or pannier. **The inlets of the instruments were kept as short as possible (50cm each) and were mounted non-flexed at the top of the backpack or pannier, for which an explicit loss correction factor was derived before the start of the campaign.'**

p.6, l.8, "Please find the detail ...' I would skip this sentence, not needed':
*Those details are essential and we would prefer including the details of the instruments because of their methods and time resolution used for the results presented and discussed.*

p.6, l.11, 'introduce IASS at first use':
*This has been changed.*

p.6, l.17, "while the sampling frequency ... was relatively high' how did you match time scales/steps for all the measurements and the GPS? This should be introduced somewhere early on as it will be rather variable across instruments and methods.':

*Excellent remark. Will be done. The instrument with the highest time resolution was the GPS sensor (Garmin camera) and so an easy match was accessible for the range of individual measurement points. For comparing different instrument results the instrument with the higher time resolution was averaged for the same period as the time resolution of the coarser resolution.*

p.6, l.27, "Location data was collected via GPS' and camera, this could be a means to derive contextual information, in addition to a time-activity diary? Was this considered?':

*Yes. Especially for classifying the environment tested this needed to be taken into account. Time was always provided for the end of each measuring interval. However, for the match of GPS (1s) and instruments (≥1s) this didn't matter much for the bicycle observations.*

p.6, l.29/30, 'how was aerosol mass measured in real time, can you elaborate on this here, as it is rather crucial for the interpretation of the results, and not trivial to achieve.':

*For real time measurements of the particle size distribution and the particle mass concentration an electrical low pressure impactor (ELPI, Decati Ltd., Finland) was used. A corona charger charges the particles which are then classified in a 12 stage low pressure impactor. The particle mass is then calculated for the different size bins (Keskinen et al. ,1992). To calculate the local background concentrations a 5% percentile filter with a time constant of 180 s was used (Bukowiecki et al., 2002; Pirjola et al. 2006; Urban, 2010; Ehlers, 2014)*

p.6, l.33, "a specific track was carried out', suggest to reformulate, e.g. 'a pre-set route was followed' or similar':
*Done.*

p.6, l.35-p.7, l.5, 'the quantitative information would be better displayed in a table or graph than in the text here':
*OK. Shifted to Table 4 and referenced in the text. Other table numbers renamed accordingly.*

p.8, l.3, "Method of relative parameters' not quite clear, suggest to rephrase e.g. 'Method for deriving relative concentration parameters' or suchlike":
*Will be rephrased to '**Method for deriving comparable** relative **concentrations'**.*

p.8, l.29-32, 'again, time steps are mentioned here, but it is not clear how temporal resolution of the measurements has been harmonised/addressed, suggest to add a paragraph earlier on to address this.':
*In order to prevent further misunderstanding a small paragraph will be added explaining how the different temporal resolutions are harmonized according to the reviewers comment: '. . . at the corresponding time. **In order to harmonize the different time resolutions of stationary and mobile measurements the urban background measurements (reference) were averaged for 30 min intervals to exclude short term local effects. The corresponding stationary data point was selected in that way that the mobile time was assorted to the data point, in which 30 min time interval the mobile data point was included**. . ..'*

p.9, l.1-14, 'as indicated above, I am not convinced that at 100 m × 100 m the land use types can provide a meaningful basis for the analysis. My suggestion would be to remove section 3.5 entirely':
*We partially agree on that and would have appreciated a better surface resolution map with the information needed. But contrary to the reviewers suggestions we want to have this part included and will reduce the number of surface usage types in order to investigate the difference between urban green spaces and areas covered sealed*

*surfaces (buildings, streets). Please see our feedback to some of the earlier comments regarding changes and adjustments we have made to revise the application of the method.*

p.9, l.15ff: 'As indicated in the general comments, suggest to revise the structure of Section 4 overall.':
*Correct. As noted above, we will condense this part and split results and discussion into two sections. This will be structured around the three aspects highlightened in the 'Focus' section.*

p.9, l.29, 'the part on the leaf blower seems to be marginal and not related to the objective to derive more general insights into the spatial variability. Could you explain better why this is important, or remove that part? It does seem to be a rather specific issue.':
*Indeed we have discussed about this aspect for long. Leaf blowers are meanwhile used in Berlin rather frequently and the results from measurements in its vicinity during running were such that a substantial influence of local air quality is to be expected. As those instruments were and are used in the context of removing biogenic material i.e. leaves etc. those are related to the indirect influence of the biosphere on air quality and pollutants although driven by mankind. Because of that we shifted it to the supporting online information document and will discuss it briefly in the discussion part to unravel potential origins of certain VOCs and PM, which sometimes display a strange behaviour. This seems likely to be caused by these kind of instruments.*

p.11, L.35-37, 'first sentence on CO is giving a generic statement about similar patterns for all gases, I would suggest to carefully check the paper and remove these, as they are repetitious and generic. Furthermore, in the results and discussion, I would not go into as much detail to explain the general sources of CO and its formation in

urban environments, as done here, it just adds more text distracting from the valuable findings of this study.':
*Will be done.*

p.12, l.15, "BLUME station' may have missed this earlier, but could not find another reference to this station name, so best introduce earlier':
*Thanks. Will be introduced at first notice. It refers to the 'Berliner Luftgütemessnetz' in English "Berlin Air Quality measurement network". An overview can be found at http://www.stadtentwicklung.berlin.de/umwelt/luftqualitaet/de/messnetz/index.shtml unfortunately only in German.*
p.13, l.1, "if and only if' please avoid such phrasing, it is not needed here to emphasise':
*OK. Will be changed to 'if'.*

p.13, l.14, "diesel driven' ... 'diesel consuming' remove driven/consuming, just 'diesel passenger cars/LDVs' is sufficient.':
*Done.*

p.13, l.16, "to the measured nitrogen dioxide mixing ratios' do you mean direct emissions of $NO_2$ from diesel oxidation catalysts? I would then make this more direct and clear, it is a bit back-to-front else.':
*At least from diesel cars, yes. The study referred to was done 13 years ago and the techniques changed meanwhile so that the results published 2003 cannot be taken as quantitative but qualitative.*

p.14, l.1ff, 'as indicated above, I am not convinced by the results based on the coarse land use type resolution, and suggest to drop this.':
*As mentioned above, we would prefer keeping this in but reduce the number of surface*

*types and the length of the subsection. Again, see responses to previous comments.*

p.15, l.32-38, 'again, a rather generic basic introduction to particulate matter, which I suggest to skip as it is not really necessary here, perhaps add one reference in a short sentence to introduce this?':
*OK.*

p.16, l.3, 'referring to 'small scale variation' here, which I think is fine and relates to my comments on micro/mesoscale wording earlier':
*We will change the scale variations to physical dimensions 'from several meters to kilometres in spatial resolution'.*

p.16, l.12, "mixed layer height' do you mean mixing layer? ':
*Yes, actually 'mixing layer height' was measured in Berlin-Neukölln and the text will be changed in the text accordingly.*

p.16, l.23, "applied for the' applied to? was applicable to?':
*Excellent suggestion. Thanks.*

p.17, l.6, "Particle mass concentrations ...' see comment above, could you elaborate somewhere how mass was measured, in the context of optical instruments being used':
*OK. The text will be shortened to 'The observed PM10 and PM2.5 **at flight level** were identical to concentrations observed at the city boundaries **at the surface** in Grunewald (west) and in Friedrichshagen (southeast) with concentrations between 9 and 10 μg/m3 (BLUME, von Stülpnagel et al., 2015).'*

p.17, l.15, "on the regional and local scale' see above, please use a consistent spatial reference for the different scales addressed':

*Will be changed to distances: 'Next we focus on surface bound measurements **by van in Berlin and its surrounding area (radius of about 65-70 km)**.'*

p.18, l.6, "particulate masses' please be more accurate and specific in referring to parameters, particulate matter mass concentrations (of PM10? 2.5?) or PNC?':

*In this case this was used on purpose. PNC was not meant as it expresses particle number but not mass concentration. The observations applied not only to a certain PM group but were made for the different groups in a similar way. PM1, PM2.5 and PM10 displayed similar behaviour but the magnitude changed. We will change this to "Similar patterns but much more moderate increases have been seen for the different particulate masses (PM1, PM2.5 and PM10). As remarkable fractions of the PM10 particle mass are of secondary organic origin. . .".*

p.18, l.28, "Please take into account ...' I would suggest to drop such formulations, they just add words and no substance, rephrase to 'The measurements with the van were conducted by following ..':

*Good point. Will be done.*

p.20, l.9ff, 'see above, suggest to drop the section 4.2.4':

*See comments made with respect to the land use types earlier. We agree on the challenge of 100m × 100m resolution. But there is currently no trustworthy usable digital map available to use and even that resolution allows some important findings, however to be applied carefully. An improved map is an important aspect for future investigations.*

p.21, l.20ff, "For most of the land use type classifications the differences between the van and bicycle measurements agree within the associated uncertainty' what is the

associated uncertainty and how is it derived?':

*The uncertainty is treated as statistically uncertainty expressed in the boxplots by the notches made with R. Those represent the range of ±1.58 the interquartile range divided by the square root of the number of observations. They match approximately the 95% confidence interval (Chambers et al., 1983) and are independent of the underlying distribution (normal, Poisson etc.). No overlap of two boxplots within the range of the corresponding notches represents a significant difference between both measurements. We will add three sentences to the methods section 3.5 about this to make it clear: 'A significant difference in the medians of two different categories is determined at confidence interval of 95% using the approach by Chambers et al. (1983) of ±1.58\*IQR/ïČŰn. IQR is the interquartile range and n stands for the number of data points considered. This formulation is independent of the underlying statistical distribution and is provided in the figures as notches.'.*

'Due to the short term of the measurement campaigns, the conclusions on the heat island effect would likely need more supporting work. Not sure if a discussion of the heat island effect here in this paper is necessary, and the caveats are outlined already in the text following on the same page. Consider shortening or removing? For a more thorough comparison, looking at the share and distribution of green space areas in different cities would be essential, in my opinion.':

*We will provide a plot in the supplementary material but will skip it from the article itself.*

p.22, l.8, "characterise air quality on multiple scales' I think this is a bit of a leap, the study very well demonstrated the capability of mobile measurement platforms to quantify specific air pollutant concentrations, in a one-off campaign based mode, so perhaps better stick to this in the formulation?':

OK. Let's take that: 'The mobile measurements with bicycle, van and air plane/glider as part of the BAERLIN2014 measurement campaign has demonstrated the ability of

integrated measurement platforms to characterize air quality in the presented one-off campaign mode.'

p.22, l.9, "large geographical area' related to earlier comments on consistent reference to spatial scales, 'large' is relative and best quantify here 'an area covering X square km'?':
*Yes. Will be changed to 'Van-based measurements were used to cover a circular area of about 65 km in radius in and around the city of Berlin, while. . .'.*

p.22, l.17/18, "elevated air pollutant concentrations found in Berlin were most likely produced in the vicinity of the observation and originated from local pollutant sources" – while I accept this for NOx and CO, I am yet to be convinced by the findings presented here that this is the case for PM2.5, which would require a discussion of the chemical composition of the PM observed and a look at the regional-scale atmospheric transport processes; previous material presented by the Senate of Berlin indicated a substantial amount of PM originating from long-range transport, and recent literature has shown this for Europe in general and several European cities, so the role of ammonia and secondary inorganic aerosols should be more thoroughly assessed before this claim can be substantiated.':
*Disagree to a certain extent. If particulate mass values increase from 8 $\mu$g/m$^3$ at the city boundaries to 17.2 or 31.8 $\mu$g/m$^3$ for the van measurements as smoothed baseline or all values during the passage throughout the city, we can draw the conclusion of elevated pollution levels to be caused locally during the time of observation. The latter should and will be emphasized. The chemical composition is nice to know but not necessarily needed for general conclusion. Detailed investigations would be favoured but were out of the scope of the presented project. We would be very keen on learning more about this in a future study. Anyway, we will add the additional phase 'during the period of observations elevated air pollution levels were found to be very likely*

*originating from local sources.'.*
* * *

---

## Author Comment (AC3) · 4 May 2016

p.22, l.27-32, 'could you quantify the 'significant' influence of vegetation on pollutant concentrations? Was it statistically significant? Contribution of isoprenes and terpenes to local ozone formation, was this quantified in the study, or is this derived from models/previous knowledge? The second half of this statement on urban vegetation is more robust and accepted.':

*Thanks for pointing this out. We added the definition of statistically significant to subsection 3.5 (see above comments) and will add a table (Table 9) with the corresponding numbers to this important statement. Those are: Compared to the urban background station in Berlin-Neukölln ozone is found to be significantly reduced in*

[Figure]

*parks (-38.7±2.3%) and forests (-14.7±2.5%), while it is significantly enhanced in agricultural areas (+30.7±3.4%). In this context coniferous forest types caused more reduction (-21.7±3.1%) than deciduous forests (-11.3±2.9%). Carbon monoxide: Significant reduction near vegetated areas, i.e. parks -28.4±0.1%, forests -37.0±0.1% and agricultural areas -26.6±0.1%, compared to Berlin Neukölln. Nitrogen oxides: Except for agricultural areas $NO_x$ was significantly enhanced, i.e. near parks +660±140% and near forests +31.3±17.3% (not significant for coniferous types). However summing $NO_x$ and $O_3$ to Ox there is no significant change except for areas close to parks (+32.5±13.5%) and agricultural areas (+14.3±2.9%). The cause is different, as the enhanced $NO_x$ increases Ox near parks, while the already produced ozone by $NO_2$ photolysis enhances Ox in agricultural areas. Ox nearby mixed forest behave similar as nearby parks. But the number of observations is much smaller (136) than for other surface types (>500). Particle number concentrations (PNC): Observations (bicycle) in vegetated areas display significantly reduced PNC compared to the background station in Neukölln, i.e. in parks (-8.5±2.5%), in forests (-29.1±1.8%) and in agricultural areas (-33.2±1.8%). Using the 'Urban' classifications for bicycle based measurements these reductions increase further. Particulate mass (PM 1 and PM10): Particulate mass reduced in any vegetated area significantly. The reductions for the individual land usage types were found identical for PM1 and PM10 within the range of uncertainty: in parks (PM1: -34.5±3.0%, PM10: -38.1±2.2%) and in forests (PM1: -61.8±1.6%, PM10: -58.1±1.5%). For agricultural areas van measurements were available only, which were done at the street. Any PNC and PM measurements were found significantly higher than at the urban background site in Berlin-Neukölln indicating the vicinity of sources. Therefore we stuck to bicycle measurements for the above calculations.*

p.22, l.33-37, 'again, the resolution of the land use maps considered makes this statement harder to justify, in addition, urban airflow patterns and complex terrain influences on wind and dispersion would need to be taken into account adequately,

which is not within the scope of this study for good reason. Perhaps this section needs to be qualified a bit to reflect these caveats.':

*Agree. We will reformulate the sentences to 'The **general** vegetation effect **described above tends** to dependent on the spatial extent of vegetated areas. Urban parks with a much smaller size compared to urban forested areas were shown to not have significantly lower but rather elevated NO or NO2 concentrations than the urban background station in Neukölln (NO: >+45% and NO2: >100%). **This was affected by the street based observations and was most likely influenced the present wind direction on site (no record), which may explain the significantly enhanced NOx levels nearby areas classified as parks. A future study definitely needs to acquire a higher resolved land surface type map usable for investigating the effect of parks**.'*

Figures: 'general point, consider making the background maps slightly less vibrant to better bring out the colours of the measurements, in particular the orange and yellow shades are hard to see. –':

*Good point. Will be done.*

Fig 6: 'add more legible legends to the graph –':

*OK. We can add the longitude and latitude and the mean diameter on the red y-axis. The y-axis description on the lower right plot doesn't improve by any other colour.*

Fig 8: 'what are the units for the upper graphs, please add to the legend':

*Thanks. The typical units for PNC ($cm^{-3}$) and for PM ($\mu g/m^3$) got lost. Will be done!*

Tables: 'Table A1: formatting of the table makes it a bit hard to read, i.e. alignment and space between columns; time resolution is variable for the instruments, relating to the

comments made above on time-synchronisation':
*The table may be turned by 90 degrees but no clear improvement was found. Concerning the time resolution: Multiple colleagues with a variety of instruments of different possible time resolutions contributed to the dataset analysed. We aimed to gain not an identical time resolved by best spatially resolved information. For all comparisons made the datasets acquired in parallel were averaged to the coarsest time resolution but not with respect to surface classification. This will become important for the video analysis currently conducted.*

**ANNEX**

A2L12 "All particle instruments except the instrument were ...' which 'instrument' are you referring to?':
*Thanks. The 'DiSCmini' got dropped and will be inserted. However, as agreed with the first reviewer, this part will be shifted to the supplementary material.*

Figure B1: 'add the unit to the legend in both cases':
*Will be done by reediting the jpg as the software (QGIS) tool doesn't allow additional modifications.*

Table C1: 'This table is rather dense and could considered to be more accessible as a bar chart?':
*The collaborators discussed about this for long since a bar chart was included in an earlier version. But the information details got lost within. Will be shifted to SOI too.*

Figure C2: 'map zoom and focus is different, making a direct comparison between

PNC and mass concentrations difficult, for no reason? suggest to make sure that both maps show the same area':
*Will be tried with the software. Thanks.*

**References**

'For a paper of this substance, some of the recent literature in particular with regard to urban PM seems to be missing, e.g. from the CLEARFLO project (http://www.clearflo.ac.uk/outreach/papers/), as well as those on long-range transport contributions and composition of urban PM (see specific comments).':
*We will add those references and hope to have improved that important topic.*
*Alves, C.A., Gomes, J., Nunes, T., Duarte, M., Calvo, A., Custódioa, D., Pio, C., Karanasiou, A., and Querol, X. (2014). Size-segregated particulate matter and gaseous emissions from motor vehicles in a road tunnel. Atmos. Res., 153, 134-144. vspace0.2cm*
*Bohnenstengel, S.I., Belcher, S.E., Aiken, A., Allan, J.D., Allen, G., Bacak, A., Bannan, T.J., Barlow, J.F., Beddows, D.C.S, Bloss, W.J., Booth, A.M., Chemel, C., Coceal, O., Di Marco, C.F., Dubey, M.K., Faloon, K.H., Fleming, Z.L., Furger, M., Gietl, J.K., Graves, R.R., Green, D.C., Grimmond, C.S.B., Halios, C.H., Hamilton, J.F., Harrison, R.M., Heal, M.R., Heard, D.E., Helfter, C., Herndon, S.C., Holmes, R.E., Hopkins, J.R., Jones, A.M., Kelly, F.J., Kotthaus, S., Langford, B., Lee, J.D., Leigh, R.J., Lewis, A.C., Lidster, R.T., Lopez-Hilfiker, F.D., McQuaid, J.B., Mohr, C., Monks, P.S., Nemitz, E., Ng, N.L., Percival, C.J., Prévôt, A.S.H., Ricketts, H.M.A., Sokhi, R., Stone, D., Thornton, J.A., Tremper, A.H., Valach, A.C., Visser, S., Whalley, L.K., Williams, L.R., Xu, L., Young, D.E., Zotter, P. (2015). Meteorology, air quality, and health in London: The ClearfLo project. Bull. Amer. Meteor. Soc., 96 (5), 779-804.*

[Figure]

Manes, F. Marando, F., Capotorti, G., Blasi, C., Salvatori, E., Fusaro, L., Ciancarella, L., Mircea, M., Marchetti, M., Chirici, G., and Munafò, M. (2016). Regulating ecosystem services of forests in ten Italian metropolitan cities: Air quality improvement by PM10 and O3 removal. Ecol. Indic., 67, 425-440.

Silli V., SalVatori E., and Manes, F. (2015). Removal of airborne particulate matter by vegetation in an urban park in the city of Rome (Italy): An ecosystem perspective. Ann. Bot. (Roma), 5, 69-78.

Valach, A.C., Langford, B., Nemitz, E., Mackenzie, A.R., and Hewitt, C.N. (2015). Seasonal and diurnal trends in concentrations and fluxes of volatile organic compounds in central London. Atmos. Chem. Phys., 15, 7777–7796.

**Literature cited in this response**

Amato, F., Alastuey, A., Karanasiou, A., Lucarelli, F., Nava, S., Calzolai, G., Severi, M., Becagli, S., Gianelle, V. L., Colombi, C., Alves, C., Custódio, D., Nunes, T., Cerqueira, M., Pio, C., Eleftheriadis, K., Diapouli, E., Reche, C., Cruz Minguillón, M., Manousakas, M.-I., Maggos, T., Vratolis, S., Harrison, R.M., and Querol, X. (2016). AIRUSE-LIFEC: a harmonized PM speciation and source apportionment in five southern European cities, Atmos. Chem. Phys., 16, 3289–3309.

Bukowiecki, N., Dommen, J., Prévôt, A. S. H., Richter, R., Weingartner, E., und Baltensperger, U. (2002). A mobile pollutant measurement laboratory—measuring gas phase and aerosol ambient concentrations with high spatial and temporal resolution, Atmos. Environ., 36, 5569‐5579.

Chambers, J.M., Cleveland, W.S., Kleiner, B., and Tukey, P.A. (1983). Graphical methods for data analysis. Statistics/probability series, Wadsworth  Brooks/Cole, Belmont,

US.

Ehlers, C. (2014). Mobile Messungen - Messung und Bewertung von Verkehrsemissionen Schriften des Forschungszentrums Jülich. Reihe Energie  Umwelt / Energy Environment 229 VII, 136 p.., 2014, ISBN 978-3-89336-989-8.

Janssen, S., Dumont, G., Fierens, F., and Mensinka, C. (2008).Spatial interpolation of air pollution measurements using CORINE land cover data, Atmos. Environ., 42, 4884–4903.

Keskinen, J., Pietarinen, K. and Lehtimäki, M. (1992). Electrical Low Pressure Impactor, J. Aerosol Sci. 23, 353-360.

Kiesewetter, G., Borken-Kleefeld, J., Schöpp, W., Heyes, C., Thunis, P., Bessagnet, B., Terrenoire, E., Fagerli, H., Nyiri, A., and Amann, M. (2015). Modelling street level PM10 concentrations across Europe: source apportionment and possible futures, Atmos. Chem. Phys., 15, 1539–1553.

Mancilla, Y., Mendoza, A., Fraser, M. P., and Herckes, P. (2016). Organic composition and source apportionment of fine aerosol at Monterrey, Mexico, based on organic markers, Atmos. Chem. Phys., 16, 953–970.

Petit, J.-E., Favez, O., Sciare, J., Canonaco, F., Croteau, P., MoËǦcnik, G., Jayne, J., Worsnop, D., and Leoz-Garziandia, E. (2014). Submicron aerosol source apportionment of wintertime pollution in Paris, France by double positive matrix factorization (PMF2) using an aerosol chemical speciation monitor (ACSM) and a multi-wavelength Aethalometer, Atmos. Chem. Phys., 14, 13773–13787.

Pirjola, L., Paasonen, P., Pfeiffer, D., Hussein, T., Hämeri, K., Koskentalo, T., Virtanen, A., Rönkkö, T., Keskinen, J., Pakkanen, T.A., and Hillamo, R.E. (2006). Dispersion of particles and trace gases nearby a city highway: Mobile laboratory measurements in Finland, Atmos. Environ., 40, 867‐879.

Stathopoulou, M., and Cartalis, C. (2007). Daytime urban heat islands from Landsat

ETM+ and Corine land cover data: an application to major cities in Greece. Sol. Energy, 81, 358–368.

Tomaselli, V., Dimopoulos, P., Marangi, C., Kallimanis, A.S., Adamo, M., Tarantino, C., Panitsa, M., Terzi, M., Veronico, G., Lovergine, F., Nagendra, H., Lucas, R., Mairota, P., Mücher, S., and Blonda, P. (2013). Translating land cover/land use classifications to habitat taxonomies for landscape monitoring: a mediterranean assessment, Landscape Ecol., 28, 905–930

Urban, S. (2010). Charakterisierung der Quellverteilung von Feinstaub und Stickoxiden in ländlichem und städtischem Gebiet, Forschungszentrum Jülich GmbH Zentralbibliothek, Verlag, Jülich.

p.22, l.33-37, 'again, the resolution of the land use maps considered makes this statement harder to justify, in addition, urban airflow patterns and complex terrain influences on wind and dispersion would need to be taken into account adequately, which is not within the scope of this study for good reason. Perhaps this section needs to be qualified a bit to reflect these caveats.':
*Agree. We will reformulate the sentences to 'The general vegetation effect described above tends to dependent on the spatial extent of vegetated areas. Urban parks with a much smaller size compared to urban forested areas were shown to not have significantly lower but rather elevated NO or $NO_2$ concentrations than the urban background station in Neukölln (NO: >+45% and $NO_2$: >100%). This was affected by the street based observations and was most likely influenced the present wind direction on site (no record), which may explain the significantly enhanced $NO_x$ levels nearby areas classified as parks. A future study definitely needs to acquire a higher resolved land surface type map usable for investigating the effect of parks.'*

---

## Author Response (AR1)

Dear Prof. Dr. Facchini, dear anonymous reviewers,

Please find the revised version of our manuscript acp-2016-57 attached. Any of the substantial changes included is based on one of both reviewers' excellent suggestions for improvement and readability of the study. Since it was impossible to send the file with all the changes marked at the particular website, this marked version will be send separately by email. Regarding the individual changes you will find our detailed response in both answers (acp-2016-57-AC1 with respect to reviewer #1 as well as acp-2016-57-AC2 and acp-2016-57-AC3 for reviewer #2) to the individual reviewer's comments (acp-2016-57-RC1 and acp-2016-57-RC2).

Overall, the focus of the study was reformulated, the lengthy results and discussion section was drastically reduced and oriented more clearly according to the desired foci and the conclusion adopted. Individual aspects were added as far as this fitted with the aims of this study (see authors responses) and figure colours changed for a better visualization. Because of lacking time and changed access to IT facilities at different institutes the colour scale of Figure 5 (horizontal pattern of particle number concentration and particulate mass) will be done next week. Every detail will be identical except the colour scale that will be in line with Figs. 3 (CO, NO and $NO_2$) and 7 (ozone). The appendix was shifted to the supporting online material except one important table describing instrumental properties for merging different platform based measurements.

Finally one additional table (Table 9) was added summarizing the different air pollutant concentrations for vegetated and non-vegetated sites in order to give clear support for the conclusions drawn.

In this way we hope to have fulfilled the anticipated changes to allow the study to be published in Atmospheric Chemistry and Physics. In case of any questions please feel free to contact either the first or the contact author.

Kind regards

Boris Bonn

[revised manuscript text omitted]